# GOttack: Universal Adversarial Attacks on Graph Neural Networks via Graph Orbits Learning

**Zulfikar Alom[1], Tran Gia Bao Ngo[1], Murat Kantarcioglu[2], Cuneyt Gurcan Akcora[3]**

[1]Department of Computer Science, University of Manitoba, Canada
[2]Department of Computer Science, Virginia Tech, USA
[3]AI Initiative - University of Central Florida, USA

`md.alom@umanitoba.ca, ngot1@myumanitoba.ca, muratk@vt.edu`
`cuneyt.akcora@ucf.edu`

## Abstract

Graph Neural Networks (GNNs) have demonstrated superior performance in node classification tasks across diverse applications. However, their vulnerability to adversarial attacks, where minor perturbations can mislead model predictions, poses significant challenges. This study introduces GOttack, a novel adversarial attack framework that exploits the topological structure of graphs to undermine the integrity of GNN predictions systematically.

By defining a topology-aware method to manipulate graph orbits, our approach generates adversarial modifications that are both subtle and effective, posing a severe test to the robustness of GNNs. We evaluate the efficacy of GOttack across multiple prominent GNN architectures using standard benchmark datasets. Our results show that GOttack outperforms existing state-of-the-art adversarial techniques and completes training in approximately $55\%$ of the time required by the fastest competing model, achieving the highest average misclassification rate in $155$ tasks. This work not only sheds light on the susceptibility of GNNs to structured adversarial attacks but also shows that certain topological patterns may play a significant role in the underlying robustness of the GNNs. Our Python implementation is shared at `https://github.com/cakcora/GOttack`.

## 1 Introduction

Recent advances in Graph Neural Networks (GNNs) have brought significant progress in node classification tasks, utilizing the power of graph topology and node features to generate insightful inferences across various application domains such as social networks (Fan et al., 2020), bioinformatics (Zhang et al., 2021) and communication systems (He et al., 2021). Despite their effectiveness, GNNs exhibit inherent vulnerabilities to adversarial attacks; a minor yet strategically designed perturbation in the graph structure or nodal features can deceive the model into erroneous predictions. This susceptibility not only undermines the reliability of GNNs but also poses a grave security risk in critical applications.

Existing approaches predominantly rely on direct node feature manipulation or edge modifications without considering their topological impact. We address this limitation by designing a novel adversarial attack framework that systematically alters the graph topology to induce misclassification errors. Distinct from existing methods, we leverage node connectivity patterns (i.e., orbits) in local graph structures to maximize adversarial efficacy.

Our studies of graph topology yield a surprising result; we have uncovered a universal attack strategy commonly employed by several well-known gradient-based adversarial models. This strategy uses two graph orbits to delineate the resilience of GNNs to adversarial manipulations. Following this discovery, we introduce the GOttack algorithm, an advanced method that identifies and exploits these vulnerable graph orbits. GOttack not only enhances attack misclassification rates but also operates

with greater efficiency, reducing the complexity of the attack search associated with such adversarial interventions.

Through rigorous experiments across three GNN node classification backbones, four state-of-the-art adversarial models, five benchmark datasets, and four defense models, we demonstrate that GOttack achieves higher misclassification rates, maintains a lower computational overhead, and proves effective against defense models.

Our contributions can be summarized as follows:

- We determine a key topological equivalence group among graph nodes, revealing its frequent use in the selection process of gradient-based adversarial models.

- Our findings present a new vulnerability in GNNs related to graph topology. Our experiments demonstrate that such attacks are highly effective against state-of-the-art defense models.

- Our proposed attack strategy, GOttack, achieves the highest misclassification rate and represents a scalable attack model suitable for large graphs.

## 2 NOTATION AND PRELIMINARIES

We consider the task of node classification in a graph, denoted as $\mathcal{G} = (\mathcal{V}, \mathcal{E}, \mathbf{X})$, where $\mathcal{V}$ represents the set of nodes, $\mathcal{E} \subseteq \{(v, w) \mid v, w \in \mathcal{V}\}$ is the set of edges, and $\mathbf{X} = \{x_0, x_1, \ldots, x_{n-1}\}$ comprises feature vectors such that $x_i \in \mathbb{R}^M$ is the $M$-dimensional feature vector of node $i$. The adjacency matrix $\mathbf{A} \in \{0, 1\}^{N \times N}$ for the graph $\mathcal{G}$ has elements $\mathbf{A}_{vw} = 1$ if there is an edge $e_{vw}$ connecting nodes $v$ and $w$, and 0 otherwise.

A subset of nodes $\mathcal{V}_L \subseteq \mathcal{V}$ is labeled, each associated with class labels from the set $\mathcal{C} = \{1, \ldots, c\}$, where $y_v$ denotes the true label of node $v$ in $\mathcal{V}_L$. This setup facilitates examining how shared labels influence edge formation between nodes, an essential aspect of understanding neighbor influence on node classification. Homophily (Zhu et al., 2020) in graphs is traditionally characterized by the similarity between connected node pairs, where nodes are considered similar if they share identical labels. The homophily ratio is constructed based on this premise as follows:

**Definition 1** (Homophily Ratio). *Let $\mathcal{G}$ denote the aforementioned graph and $\mathbf{y}$ represent the vector of node labels. The homophily ratio is defined as the proportion of edges connecting nodes with the same labels, formally given by: $h(\mathcal{G}, \{y_v; v \in \mathcal{V}\}) = \frac{1}{|\mathcal{E}|} \sum_{(v,w) \in \mathcal{E}} \mathbb{1}(y_v = y_w)$, where $\mathbb{1}(\cdot)$ denotes the indicator function.*

A graph is considered highly homophilous if the homophily ratio $h(\cdot)$ is large, typically within the range $0.5 \leq h(\cdot) \leq 1$. Conversely, a low homophily ratio indicates a heterophilous graph.

**Node Classification.** The goal of node classification is to infer a function $g : \mathcal{V} \to \mathbb{P}(\mathcal{C})$, that assigns a probability distribution over the class set $\mathcal{C}$ to each unlabeled node $v$, where $\hat{y}_v$ is the predicted class for node $v$, identified as the class with the highest probability in $g(v)$. This setup, characterized as transductive learning, implies that the model predictions are based on instances both seen and used during training.

The Graph Convolutional Network (GCN), as introduced by Kipf & Welling (2017), provides a foundational model for understanding and analyzing the vulnerabilities exposed by our proposed attack model. GCN employs a message-passing technique that utilizes the features of neighboring nodes, making it susceptible to adversarial manipulations that can alter node connections and lead to misclassifications. As a result, this section will define our attack using the GCN operations, which are detailed in Appendix B.1.

**Adversarial Attack.** Adversarial attacks on graph data aim to subtly perturb graph structures or node features, causing GCN to misclassify specific nodes. This entails creating a new graph $\mathcal{G}' = (\mathbf{A}', \mathbf{X}')$ from the original $\mathcal{G} = (\mathbf{A}, \mathbf{X})$, with changes to $\mathbf{A}$ (i.e., structural attacks) or $\mathbf{X}$ (i.e., feature attacks). In an attack, a subset of nodes $v \in \mathcal{V}_T \subseteq \mathcal{V}$ are targeted to have the GCN misclassify their labels $\hat{y}_v \neq y_v$ within a specified budget $\Delta$ as $\sum_u \sum_f \left| \mathbf{X}_{uf} - \mathbf{X}'_{uf} \right| + \sum_{u<w} |\mathbf{A}_{uw} - \mathbf{A}'_{uw}| \leq \Delta$.

The node $v$ may be directly affected (i.e., $u = v$), or influence attacks can impact any other node within the graph (i.e., $u \neq v$). The attacks take various forms and occur during different phases. i)

Poisoning attacks occur during training time, aiming to compromise the model by manipulating the training dataset. Evasion attacks occur at test time, attempting to generate deceptive samples that evade detection by a trained model. ii) In targeted attacks, the objective is to misclassify specific target nodes $\mathcal{V}_T$, while in non-targeted attacks, the goal is to reduce the overall accuracy of the model.

## 3 RELATED WORK

Recent interest has grown in adversarial attacks on graph neural networks Günnemann (2022). These attacks demonstrate how minor changes to input features or graph structure can alter network outputs, often causing incorrect classifications. We begin with an overview of gradient-based attacks and then discuss non-gradient-based methods (refer to App. Table 56 for a complete classification).

**Gradient-based attacks.** Gradient-based attacks on graph neural networks exploit the gradients of the model to perturb node features or graph structure, aiming to mislead the network into making incorrect predictions. Zügner et al. (2018) proposed Nettack, which marked the inaugural exploration of gradient attacks on attributed graphs, revealing significant accuracy declines even with minor alterations. Another novel approach by Xu et al. significantly reduced classification performance by causing a small number of edge perturbations (Xu et al., 2019a). Similarly, Li et al. (2023) introduced the SGA framework to target nodes by using a smaller subgraph around the target node and leveraging gradient information for attack optimization. Fast Gradient Attack (FGA) used gradient information from GCNs and outperformed baseline methods by efficiently disturbing network embedding with minimal link rewiring (Chen et al., 2018). To describe the robustness of deep learning models for graph-based tasks, Zügner & Günnemann (2019) introduced training-time attacks using meta-gradients to perturb graph structures, which effectively renders the model near-useless for production use, all without direct access to the target classifier. Our approach also uses a gradient-based attack; however, we reduce the amount of costly gradient computations.

**Non-gradient-based attacks.** Sun et al. (2020) introduced a novel node injection poisoning attack that used hierarchical Q-learning to optimize the injection process. Likewise, Chang et al. (2022) proposed the GF-Attack for conducting black-box adversarial attacks on graph embedding models without access to labels. Hussain et al. (2021) proposed Structack, which uses structural centrality and similarity insights to lower GNN costs efficiently. Similarly, Zou et al. (2021) proposed the topological defective graph injection attack, where adversaries inject adversarial nodes into existing graphs rather than modifying links or node attributes. Zhang et al. (2023) proposed membership inference attacks targeting edges, also known as link-stealing attacks, which used customized attacks by introducing a group-based attack paradigm that is suited to various groups of edges. Mu et al. (2021) proposed the attack as an optimization problem to minimize perturbations to the graph structure, with a particular emphasis on the difficult hard-label black-box attack scenario.

The approach we propose in this paper differs from all the above-mentioned proposals in that none have attempted to identify equivalence groups for graph nodes based on graph orbits to minimize the search space in discrete optimization. Furthermore, our proposed solution introduces a highly effective and efficient algorithm for universal attacks on node classification models, demonstrating significantly faster performance compared to existing methods (e.g., attack training in approximately 55% of the time required by the fastest competing model).

## 4 METHODOLOGY

We propose the GOttack framework to execute adversarial attacks on node classification GNNs while minimizing changes to the graph's structure. Figure 1 illustrates the overview of the GOttack research workflow.

**Challenge.** A significant attack challenge in our task is the substantial time complexity, as structural attacks on the underlying graph data might require up to $O(2^{|\mathcal{V}| \times |\mathcal{V}|})$ steps for finding the optimal set of edges to remove or add. This scale of complexity necessitates the development of efficient methods.

**GOttack Philosophy.** GOttack draws inspiration from the Mapper philosophy of Topological Data Analysis (TDA), as described by Singh et al. (2007). This philosophy posits that data has shape, and finding the shape may allow us to build better models. In our case, we focus on identifying and

Figure 1: Complete research workflow diagram of GOttack.

grouping nodes on the graph so that graph topology can help us select which nodes and edges to attack in GNNs.

**GOttack Solution.** GOttack utilizes group theory (Bogopolskij, 2008) to identify node equivalence classes (Alon, 2007) on a graph, which guide our decisions on which edges to add or remove. This approach strategically groups nodes according to their positions on the graph in potential attack strategies, thereby enhancing both the precision and time-efficiency of our adversarial interventions.

## 4.1 ATTACK MODEL

We aim to target a specific node $v \in \mathcal{V}$ in a **targeted, structural, direct poisoning attack** to alter its predicted class. As the prediction of $v$ depends not only on its individual attributes but also on the characteristics of neighbouring nodes $\mathcal{N}(v)$ within the graph, we are focused on perturbations to the initial graph $\mathcal{G}$ with the condition: $\exists w, \mathcal{A}'_{vw} \neq \mathcal{A}_{vw}$ where $w \in \mathcal{V}$. However, to prevent the attacker from completely modifying the graph, we impose a constraint on the total number of allowable changes, controlled by a specified budget $\Delta$: $\sum_{w \in \mathcal{V}} |\mathcal{A}_{vw} - \mathcal{A}'_{vw}| \leq \Delta$.

**Problem Statement.** Consider an initial graph $\mathcal{G} = (\mathcal{A}, \mathcal{X})$, a target node $v$, and its true class $y_v$. Our goal is to modify the graph's structure such that the classification of $v$ changes from $y_v$ to $y_{v'}$, where $y_v \neq y_{v'}$, thereby maximizing the difference from its original classification. The proposed attacks can be mathematically formulated as a bi-level optimization problem:

$$\arg \max_{(\mathcal{A}', \mathcal{X}) \in \mathcal{G}'} \max_{y_{v'} \neq y_v} \ln Z^*_{v, y_{v'}} - \ln Z^*_{v, y_v} \tag{1}$$

where $\mathbf{Z}^* = f_{\theta^*}(\mathcal{A}', \mathcal{X})$ and $\theta^* = \arg \min_\theta \mathcal{L}(\theta; \mathcal{A}', \mathcal{X})$ subject to the budget constraint. Specifically, we aim to find a modified graph $\mathcal{G}' = (\mathcal{A}', \mathcal{X})$ in which the target node $v$ is assigned a label $y_{v'}$ that maximizes the difference from its original label $y_v$ in terms of probability scores.

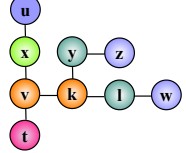

## 4.2 EQUIVALENCE CLASSES IN STRUCTURAL ATTACKS

We utilize an attack strategy based on node equivalence groups to guide our decisions regarding the addition and deletion of edges. This section begins by defining graphlets, which are instrumental in identifying node equivalence groups. We then discuss orbits, representing the specific positions a node can assume within a graphlet to facilitate effective grouping. As a last step, we compute Graph Orbit Vectors from all graphlets to develop a multi-orbit based attack strategy.

Figure 2: A toy graph where shared node colors imply similar orbit counts. Nodes $u$, $z$, and $w$ have 15 and 18 orbits, respectively.

**Definition 2** (Graphlet (Kloks et al., 2000)). *A graphlet $\mathcal{G}_{gp}$ within a larger graph $\mathcal{G} = (\mathcal{V}, \mathcal{E}, \mathcal{X})$ is a connected induced subgraph $\mathcal{G}s' = (\mathcal{V}', \mathcal{E}', \mathcal{X}')$, where $\mathcal{V}' \subseteq \mathcal{V}$, and $\mathcal{E}'$ includes all edges $e_{uv} \in \mathcal{E}$ with both $u$ and $v$ in $\mathcal{V}'$, and $|\mathcal{V}'|$ typically equals 5 (as defined in Appendix Figure 6).*

There are 30 distinct graphlets of 5-nodes (see Appendix Figure 6 for the shapes). For instance, consider the nodes $\{u, x, v, k, y\}$ in Figure 2, which forms a graphlet with $|\mathcal{V}'| = 5$.

Orbits are defined by automorphisms of the graphlet; an automorphism $\sigma$ of a graphlet $\mathcal{G}_{gp}$ satisfies $\sigma \cdot \mathcal{G}_{gp} = \mathcal{G}_{gp}$. Nodes $v$ and $w$ in $\mathcal{V}$ are similar if there exists an automorphism $\sigma$ such that $\sigma(v) = w$.

The orbit of a node $v$, denoted by $\text{Orb}(\mathcal{G}_{gp}, v)$, is the set of all nodes $w \in \mathcal{V}$ that can be mapped onto $v$ by some automorphism of the graphlet:

**Definition 3** (Orbit (Alon, 2007)). $Orb(\mathcal{G}_{gp}, v) = \{w \in \mathcal{V} \mid \sigma \in Aut(\mathcal{G}_{gp}) : \sigma(v) = w\}$.

where $\text{Aut}(\mathcal{G}_{gp})$ is the group of automorphisms of $\mathcal{G}_{gp}$. Each orbit is denoted by $\text{Orb}_j$, where $j$ is a unique identifier for each orbit within a specific graphlet. A node $v$ *touches* an orbit $\text{Orb}_j$ if $v$ is part of an induced subgraph in the graph and $v$ belongs to $\text{Orb}_j$. By extension, a node may appear in multiple graphlets and, hence, occupy multiple orbits. For example, in Figure 2, node $x$ appears in graphlets comprising of nodesets $\{x, v, k, l, w\}$ and $\{x, v, k, y, l\}$ and so on. Overall, the 30 distinct graphlets create **73 distinct orbits** (see Appendix Figure 6 for the orbit positions). The number of graphlets and orbits are all uniquely determined by the choice of using 5-node graphlets.

**Definition 4** (Graph Orbit Vector). *We propose a Graph Orbit Vector (GOV) as a numerical representation of a node's participation across the **73 orbits** in a graph. Consequently, $\mathbf{GOV_v}$ of a node $v$ is an $n = 73$-dimensional vector, where each dimension corresponds to the count of a specific orbit touched by node $v$. This dimensionality reflects the full set of topological positions a node can occupy across all $5$-node graphlets, making it a comprehensive description of a node's structural embedding.*

We initialize the vector element for $\text{Orb}_j$ of node $v$ at zero and increase it by one each time $v$ appears in orbit $\text{Orb}_j$ of any graphlet, where $\text{Orb}_j$ is an induced subgraph containing $v$. This count reflects the number of graphlets in which $v$ participates through orbit $\text{Orb}_j$. As a result, the Graph Orbit Vector provides a profile of a node's topological embedding within the graph. Notably, orbit discovery is performed for the entire graph as a pre-processing step, eliminating the need for recomputation for each target node.

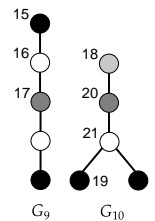

### 4.3 GOTTACK: GRAPH STRUCTURE POISONING VIA ORBIT LEARNING

In our exploration of graph topology, we have discovered a distinctive feature shaped by orbits 15 and 18: nodes touching these orbits appear in peninsula-like subgraphs (such as the one formed in Figure 2 by nodes $\{u, x, v, k, t\}$). These orbits (see Figure 3) are characteristic of being three or four edges away

Figure 3: Graphlets where orbits 15 and 18 are defined.

from another endpoint in the graphlet and serve as critical indicators of topological peripheries within a graph. This geometric arrangement lends a substantial foundation to our subsequent analyses and adversarial strategies on graph data, as discussed in the following hypotheses and experimental sections.

We start by stating our topological observation as influenced by the Mapper philosophy: 1) **Orbit Proxy.** Under the homophily assumption, node classification posits that graph neighbors are similar to a node in the label. Consequently, nodes in more distant positions, as can be identified by orbits, are less similar. 2) **Periphery Orbits.** Orbits 15 and 18 indicate the topological periphery in a graph and provide a useful proxy for identifying distant nodes that differ in labels.

We claim that the path distance within the graph encodes a notion of remoteness, which in turn yields minimal information about a node's label. This indicates that physical proximity within the network strongly influences the predictive accuracy regarding node labels. In our experiments (Section 5), we also provide empirical evidence demonstrating the utility of the Orbit Proxy in heterophilic graph cases, albeit in a weaker form.

In Theorem 1, we formally state that nodes located in orbits 15 and 18, due to their peripheral placement, are particularly effective for establishing paths to remote parts of the graph. This has significant implications for designing network protocols and algorithms that rely on efficient data traversal and retrieval mechanisms.

**Theorem 1** (Remote Connection Candidates). *Let $H(v, w)$ denote the expected random walk hitting time from node $v$ to node $w$ in $\mathcal{G}$. For any target node $v \in V$, nodes in orbits $15$ and $18$ are the most effective candidates of $w$ for establishing paths to the most remote parts of $\mathcal{G}$, due to their longer expected hitting times $H(v, w)$ compared to other nodes not in these orbits.*

Due to space limitations, the proof is given in Appendix A.

The Periphery Orbits observation forms the backbone of our attack strategy; we identify nodes of periphery orbits (i.e., 15 and 18) and affect the adjacency matrix (i.e., add or remove edges to these

nodes) accordingly to confuse a GNN to misclassify a target. Consider the target node $v$ in Figure 2. Our hypothesis posits that creating an edge from $v$ (or any other node) to either of the (orbit 15 and 18) nodes $u$, $z$ or $w$ will yield the highest misclassification error in GCN. The selection among periphery nodes uses a gradient-based method, as we detail below in surrogate loss.

**Orbits** $(15, 18)$. We define two ordered orbit categories based on the highest and second-highest orbit counts. The highest is given by $Orb_{\max}^v = \arg\max(\text{GOV}_v)$. If multiple orbits share this count, one is chosen arbitrarily. The second-highest is $Orb_{\sec}^v = \arg\max(\{j \in \text{GOV}_v \mid j \neq Orb_{\max}^v\})$. Each node is then assigned an orbit category $Orb^v = Orb_{\max}\|Orb_{\sec}$, where order does not matter.

**Example 1.** *Consider a node $v$ in graph $\mathcal{G}$. Although GOttack works with $k = 5$-node graphlets, for simplicity, this example employs three-node graphlets ($k = 3$) for orbit counting, yielding a 4-dimensional vector to store all possible orbits. Suppose the orbit count vector for node $v$ is $GOV_v = [4, 15, 11, 12]$. The task is to identify the largest and second-largest orbit count values in $GOV_v$. The largest orbit count value is $15$, denoted by $Orb_{max}^v = 01$, and the second-largest orbit count value is $12$, denoted by $Orb_{sec}^v = 03$. Thus, node $v$ is categorized into the orbit category $Orb^v = 01|03$ or $0103$ in short.*

**Surrogate loss.** Our objective is to maximize the discrepancy in log probabilities for the target node $v$ within a specified budget $\Delta$. The log-probabilities can be simplified to $\hat{A}^2 \mathcal{X} \mathcal{W}$. We linearize the model by replacing the nonlinearity $\sigma(.)$ with a simple linear activation function. Therefore, from Eq. 3, $Z' = \text{softmax}(\hat{A}\hat{A}XW^1W^2) = \text{softmax}(\hat{A}^2 XW)$. Therefore, the surrogate loss function, $\mathcal{L}_s(\mathcal{A}, \mathcal{X}; \mathcal{W}, v)$, is designed to optimize the following objective: $\underset{(A', X) \in \mathcal{G}'}{\arg\max} \mathcal{L}_s(\mathcal{A}', \mathcal{X}; \mathcal{W}, v)$, where,

the surrogate loss function $\mathcal{L}_s$ is defined as $\mathcal{L}_s(\mathcal{A}', \mathcal{X}; \mathcal{W}, v) = \max_{w \neq z}[\hat{A}^2 XW]_{v,z} - [\hat{A}^2 XW]_{v,w}$. This function aims to solve the maximum loss over a set of permissible changes in $\mathcal{A}$ of $\mathcal{G}$.

**Structure poisoning.** We compute a candidate node set called the orbit category, denoted as $Orb_{cat}$, consisting only of allowable elements $(v, u)$ where the edge changes from 0 to 1 (i.e., adding an edge) or vice versa. Specifically, for a given target node $v$, we create a candidate set such that $u \in \mathcal{V}$ where $Orb_{\max}^u = 1518$. Among the candidate edge changes, we select the one that yields the highest surrogate loss. However, to compute the surrogate loss score, we first need to determine the class prediction of the target node $v$ after adding or removing an edge $(u, v)$. Here, we are optimizing the loss score with respect to $\mathcal{A}$; the term $\mathcal{X}\mathcal{W}$ is constant. The log-probabilities of node $v$ are then given by $g(v) = [\hat{A}^2]_v \cdot C$, where $[\hat{A}^2]_v$ denotes a row vector and $C$ is the constant term ($\mathcal{X}\mathcal{W}$). Thus, we only need to inspect how this row vector changes to determine the optimal edge manipulation. Following the insight developed by Zügner & Günnemann (2019), we can derive an incremental update, so there is no need to recompute the updated $[\hat{A}^2]_v$ from scratch.

We assume access to the complete graph and its training labels, allowing adversaries to exploit its structure but not the target model, focusing instead on transferable attacks. This setting applies to domains like social and transaction networks, where graph structures are publicly visible (e.g., social media friendships, academic citations, and cryptocurrency transactions).

**Time Complexity.** Orbit discovery is the primary computational cost of GOttack. The time complexity for computing all orbits for all nodes is $O(|E| \times d + |V| \times d^4)$, where $O(|V| \times d^4)$ corresponds to the time required to enumerate all five-node graphlets, and $d$ denotes the maximum degree in the graph. For instance, orbit discovery on the CORA dataset takes only 0.17 seconds. All time costs for orbit discovery are provided in Appendix Table 52.

## 5 EXPERIMENTS

This section presents the experimental evaluation we carried out to show the effectiveness of the proposed approach. We answer the following questions: i) *How effective is the GOttack approach in terms of misclassification rate compared with existing state-of-the-art approaches?* ii) *How efficient is the proposed model in terms of computation time compared with the existing models?* iii) *How easy is it to defend against GOttack compared to existing models?*

**Datasets.** We conduct experiments on five widely used node classification datasets, the statistics of which are provided in Table 1. Cora (Yang et al., 2016), Citeseer (Yang et al., 2016) and Pubmed

Table 2: Misclassification rate (in %) (↑) of target nodes in five datasets where three backbone GNNs (GCN, GIN and GraphSAGE) are attacked in node classification with budget $\Delta = 1$. See Appendix D.1 for stds.

| | Cora | | | Citeseer | | | Polblogs | | | BlogCatalog | | | Pubmed | | |
|---|---|---|---|---|---|---|---|---|---|---|---|---|---|---|---|
| Method | GSAGE | GCN | GIN | GSAGE | GCN | GIN | GSAGE | GCN | GIN | GSAGE | GCN | GIN | GSAGE | GCN | GIN |
| Random | 19.11 | 2.07 | 17.30 | 30.01 | 2.01 | 10.03 | 15.13 | 12.04 | 17.04 | 3.09 | 12.09 | 4.19 | 14.02 | 20.05 | 20.05 |
| Nettack | 58.04 | 34.06 | 46.10 | 66.09 | 46.04 | 57.04 | 29.02 | 38.04 | 13.02 | 50.11 | 20.02 | 65.07 | 52.01 | 50.04 | 47.02 |
| FGA | 54.08 | 32.05 | 40.09 | 60.11 | 31.10 | 44.10 | 22.08 | 31.09 | 14.02 | 46.15 | 10.04 | 61.09 | 42.03 | 32.02 | 52.00 |
| SGA | 61.06 | 41.05 | 57.05 | 60.06 | 41.06 | 57.06 | 35.05 | 37.07 | 35.08 | 51.45 | 24.03 | 61.02 | 30.00 | 57.04 | 47.01 |
| PRBCD | 35.02 | 41.06 | 36.10 | 35.04 | 46.04 | 42.02 | 8.01 | 42.07 | 33.06 | 33.05 | 33.05 | 5.09 | 38.02 | 52.03 | 43.06 |
| **GOttack (ours)** | 59.05 | 41.52 | 37.03 | 61.09 | 46.07 | 57.06 | 29.03 | 41.08 | 15.08 | 52.10 | 22.04 | 63.08 | 52.08 | 57.09 | 55.05 |

(Yang et al., 2016) datasets are citation networks with undirected edges and binary features where nodes are publications and edges are citation links.

In the Polblogs (Adamic & Glance, 2005) dataset, nodes are political blogs, and edges are links between them. In the BlogCatalog (Tang & Liu, 2009) dataset, nodes' attributes are constructed by keywords, which are generated by users as a short description of their blogs. We split the networks into labeled (20%) and unlabeled nodes (80%).

We further split the labeled nodes into equal parts *training* and *validation* sets to train the surrogate model. We have used the ORCA algorithm (Hocevar & Demsar, 2014) for the orbit discovery process on these datasets.

Table 1: Dataset statistics.

| Dataset | Hom. | Nodes | Edges | Features | Labels |
|---|---|---|---|---|---|
| Cora | 0.81 | 2,485 | 5,069 | 1,433 | 7 |
| Citeseer | 0.74 | 2,110 | 3,668 | 3,703 | 6 |
| Polblogs | 0.91 | 1,222 | 16,714 | 1,490 | 2 |
| Pubmed | 0.81 | 19,717 | 44,325 | 500 | 3 |
| BlogCatalog | 0.40 | 5,196 | 171,743 | 8,189 | 6 |

**Experimental Setup.** We have conducted the experiments under the transductive, semi-supervised learning setting. We have used both common node classifier GNNs, including GCN (Kipf & Welling, 2017), GIN (Xu et al., 2019b), and GraphSAGE (Hamilton et al., 2017) and defense models, including RobustGCN (Zhu et al., 2019), GCN-Jaccard (Wu et al., 2019), GCN-SVD (Entezari et al., 2020), and MedianGCN (Chen et al., 2021) as the backbone to evaluate our adversarial attacks.

We average over five different random initializations/splits, where for each, we follow these steps: Initially, we train the surrogate model on the labeled data. From the test set, among all nodes correctly classified, we adopt the popular practice of Nettack and select: (i) the 10 nodes with the highest margin of classification, indicating clear correctness, (ii) the 10 nodes with the lowest margin (still correctly classified), and (iii) 20 additional nodes randomly chosen. These selected nodes will be the targets for the attacks. All results presented here are computed by us in the same settings.

In our attack model, the budget $\Delta$ limits the number of allowed perturbations, such as edge additions or deletions. Its optimal value depends on the graph's structure, particularly node degrees: small $\Delta$ (e.g., 1 or 2) can significantly impact low-degree nodes, while higher values suit high-degree nodes. To ensure comparability, we adopt fixed budgets ($\Delta = 1, \ldots, 5$), following prior works like Nettack and FGA.

We compare GOttack against five state-of-the-art graph adversarial attack frameworks: Nettack (Zügner et al., 2018), FGAttack (FGA)(Chen et al., 2018), SGAttack (SGA) (Li et al., 2023) and PRBCD (Geisler et al., 2021) as well as a Random (dummy) baseline (see App. Section B.2 for descriptions). We use GCN as the surrogate model for the attack models except for SGA, which suggests SGC. We report the misclassification rate, which is the percentage of nodes that were incorrectly classified by the model in relation to the total number of nodes being classified.

In our experiments, we utilized the PyG (PyTorch Geometric) library, which employs PyTorch as the backend for implementing GNN models. Additionally, we used the PyTorch adversarial library DeepRobust (Li et al., 2020) for robustness evaluations. The experiments were conducted using Python (Version 3.8.19) and PyTorch Version 3.10.0. The computational environment was a Linux cluster equipped with an AMD Ryzen Threadripper 3960X 24-core processor, 2520GB RAM and NVIDIA RTX A6000 with 49GB RAM GPUs. Our Python implementation is shared at `https://github.com/cakcora/GOttack`.

**Parameters setting.** Our models were trained using the Adam (Kingma & Ba, 2015) optimization algorithm, employing a fixed learning rate of 0.01. Training sessions were conducted over a span of 200 epochs; we employed the *softmax* activation function.

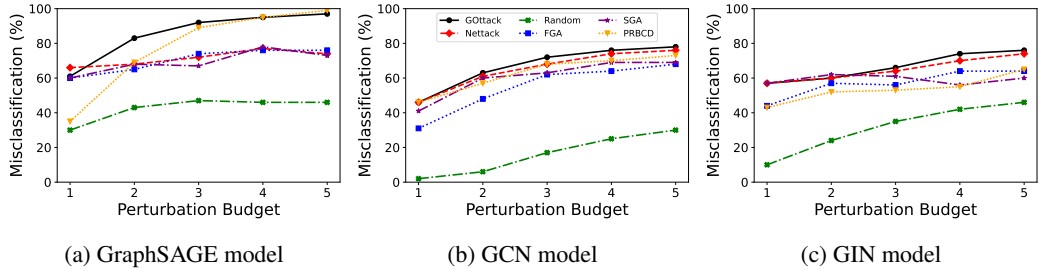

(a) GraphSAGE model    (b) GCN model    (c) GIN model

Figure 4: Budgeted attack results on the Citeseer dataset.

## 5.1 GOTTACK MISCLASSIFICATION RESULTS

Table 2 shows the misclassification rate achieved with a single edge perturbation (addition or deletion). GOttack yields the highest misclassification rates in 7 out of 15 attack settings and ranks second in 5 other settings. SGAttack yields 4 best attacks, while PRBCD and Nettack each perform best in 2 attack settings. PRBCD, a widely used method adopted by PyTorch, yields surprisingly low rates in two datasets for GSAGE and GIN. GOttack yields the highest overall rate of 52.08, whereas the second best Nettack yields 47.02.

Likewise, Table 3 shows the misclassification rates of the top-3 attack models on defense models with a single edge perturbation. GOttack yields the highest misclassification rates in 7 out of 16 attack settings and ranks second in 7 other settings, while SGA achieves the best performance in 7 attack settings. GOttack achieves the highest overall rate of 33.07, whereas SGA has a rate of 32.5.

In Figure 4, we gradually increase the attack budget and represent the corresponding misclassification rates on the Citeseer dataset (see Appendix D for all results). We observe that for GraphSAGE, GOttack yields better misclassification results, reaching a maximum misclassification score of 97% with 5 perturbations. Similarly, for the GCN and GIN models, the proposed approach attains misclassification scores of 77% and 76%, respectively, outperforming the SOTA models.

Extending the analysis of the relationship between increasing budgets ($\Delta = 1, ..., 5$) and misclassification rates to all datasets, we observe the following behavior (see Section D for complete results): **GOttack achieves the highest misclassification rate in** 28 **out of** 65 **tasks** for GCN, GSAGE and GIN models across all budgets. The numbers are 16 for PRBCD, 16 for SGAttack, and 3 for Nettack. **GOttack achieves the highest average rate of** 0.58 **over all budgets and datasets compared to the second best model of PRBCD and SGAttack with** 0.57 (Appendix Table 8).

Furthermore, as shown in Table 4, GOttack demonstrates superior computational efficiency and scalability, making it particularly well-suited for large-scale graph datasets. This efficiency stems from its topological approach to candidate node selection, which reduces the search space for potential attack points without sacrificing effectiveness. For example, in the BlogCatalog dataset, GOttack requires only about 55% of the time taken by Nettack to execute, significantly reducing computational overhead. Despite its speed, GOttack still maintains a high level of attack effectiveness, generating attack candidates in less than 10 minutes, even for datasets with a large number of nodes and edges. This balance between performance and computational cost highlights GOttack's practical applicability to real-world scenarios.

While both Nettack and GOttack are scalable for large graphs, their methodologies differ: Nettack selects candidates based on degree distribution in each iteration, whereas GOttack leverages orbit structures. Although orbit discovery incurs some overhead, it reduces the candidate set to about 23% of all nodes (see Appendix Figure 8), making the initial cost worthwhile. Among alternative methods,

Table 3: Misclassification rate (in %) (↑) of target nodes in different datasets against four defense models (RGCN, GCN-Jaccard, GCN-SVD and MedianGCN) are attacked in node classification with budget $\Delta = 1$. See Appendix D.2 for stds.

| Method | Cora | | | | Citeseer | | | | Polblogs | | | | BlogCatalog | | | |
|---|---|---|---|---|---|---|---|---|---|---|---|---|---|---|---|---|
| | RGCN | JAC | SVD | MDGCN | RGCN | JAC | SVD | MDGCN | RGCN | JAC | SVD | MDGCN | RGCN | JAC | SVD | MDGCN |
| SGA | 44.01 | 33.02 | 28.03 | 32.05 | **53.00** | 36.01 | 24.04 | **36.01** | **46.03** | 43.05 | **16.01** | **43.05** | 25.02 | **25.02** | 15.05 | **20.01** |
| Nettack | **48.08** | 38.01 | 24.01 | 32.05 | 46.04 | 42.02 | **28.04** | 27.05 | 38.05 | 46.03 | 12.04 | 33.02 | 35.04 | 19.03 | 19.02 | 17.02 |
| **GOttack (ours)** | 43.04 | **39.01** | **28.08** | 32.07 | 48.07 | **42.09** | 25.03 | 28.02 | 40.02 | **53.06** | 10.02 | 34.07 | **35.05** | 20.02 | **20.00** | 19.03 |

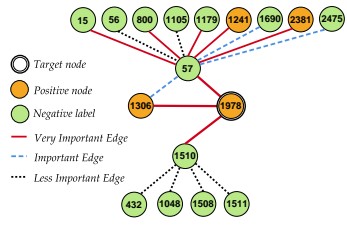
(a) Computation graph before attack

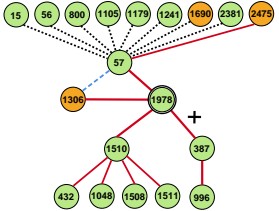
(b) Computation graph after attack

Figure 5: The computation graph for the targeted node $1978$ from the CORA datasets, as identified by GNNExplainer (Luo et al., 2020). The edge $(1978, 387)$ is added during the successful attack. Edge importances change considerably after the attack and negative class gains importance due to the newly added nodes.

only SGA is more scalable, as it focuses on local neighborhoods for candidate selection; however, its performance is worse than GOttack.

We compared GOttack to adaptive attacks, which consider specific model architectures during attack generation. Results (see Appendix Section D) reveal that while adaptive attacks outperform in single-model scenarios, GOttack's transferable perturbations exhibit broader applicability across diverse settings, reinforcing its utility in exploratory research on adversarial robustness.

## 5.2 INSIGHTS FROM GOTTACK RESULTS

GOttack strategically selects nodes using the graph topology. This selection criterion prompts several thoughts and questions, which we address below:

**The periphery definition.** A visual inspection of Figure 6 shows that more orbits, such as $19$, $39$, and $27$, could fit the periphery definition. The primary reason for not considering these orbits is their relative scarcity in all the graph datasets. For example, node orbits from the Cora dataset in Appendix Figure 8 show that most nodes predominantly feature orbits within the $1518$ and $1922$ groups. Furthermore, as shown in Appendix Section D, attacks based on $1819$, $1519$ and $1922$ yield less powerful attacks.

**Higher order orbits.** GOttack uses two orbits: $1518$. The decision against using $> 2$ orbits is influenced by the fact that nodes typically do not have more orbits. For the single orbit case, our experiments had larger time complexity, as a larger pool of nodes was considered, yet the efficacy of attacks remained similar.

**Gradient-based models target $1518$ nodes.** Our analysis reveals that **gradient-based models predominantly target $1518$ nodes**, as shown for Net-

Table 4: End-to-end time costs in seconds ($\downarrow$). See Appendix Table 51 for stds.

|  | Global | | | | Local |
|---|---|---|---|---|---|
|  | GOttack | Nettack | FGA | PRBCD | SGA |
| Cora | 48.27 | 59.57 | 55.63 | 242.37 | **32.24** |
| Citeseer | 42.02 | 46.12 | 42.43 | 210.55 | **41.02** |
| Polblogs | **139.68** | 175.23 | 165.37 | 210.87 | 164.07 |
| BlogCatalog | 512.04 | 916.91 | **223.02** | 259.38 | 335.91 |
| Pubmed | 246.96 | 243.78 | 533.55 | 240.23 | **68.23** |
| **Median** | 139.68 | 175.23 | 165.37 | 240.23 | 68.23 |

tack in Table 5 and for other models in Appendix Tables 6 and 7. For instance, Table 5 shows that in the highly homophilous Polblogs, 97.5% of initial attacks involve $1518$ nodes, despite these nodes comprising only 9.41% of the total. A similar trend appears in the heterophilous BlogCatalog dataset (2.5% and 22.5% in $\Delta = 1, 2$ attacks). This highlights the strategic role of orbit $1518$ nodes in network attacks.

**Node, Homophily, Distance and Subgraph-based Explanations for GOttack.** We conducted a comprehensive analysis to understand how GOttack causes node misclassification. Initially, we used GNN explainers (see Appendix Section E.1), which factor in the subgraph effects and node features in perturbations. Figure 5 shows an example where misclassifications are due to changes in the importance of existing edges, which offers evidence that explanations specific to nodes or edges alone are insufficient to adequately explain an attack, as shown in Figure 5. However, as widely noted in the literature (Li et al., 2024a; Li et al., 2024b), explainers often disagree on the exact interpretation (see Figure 9b).

Next, we focused on node-centric metrics to determine the positional changes of the target node within the graph following the attack. As detailed in Appendix Table 45, the target nodes' clustering coefficient, degree, betweenness, and closeness centralities did not show a significant difference compared to those influenced by other attacks. Secondly, we examined whether the attack brought in nodes with different labels into the computation graph of the target node (see Appendix Table 50). We found that after the attack, both similar and differently labeled nodes increased by 0.92 and 3.36 on average, respectively. However, these values are not as pronounced as those seen in other attacks, suggesting that the induced label diversity change by GOttack is not substantial. Next, we computed the shortest path distances from the target nodes to nodes with similar and different labels in the entire graph (see Appendix Table 48). This analysis provided empirical proof supporting our Theorem 1: the 1518 strategy more significantly reduces the distance to nodes of different labels ($-0.03$) than to those of similar labels ($-0.02$). This behavior, which indicates a targeted modification in network dynamics, was not observed with other orbits.

**Budget Selection and Structural Implications.** The choice of the attack budget $\Delta$ is important in determining the effectiveness and practicality of adversarial attacks. Fixed budgets, as employed in this study, provide consistency for benchmarking but may oversimplify real-world scenarios where the impact of a perturbation is context-dependent. For example, a budget of 5 for a node with a degree of 2 introduces a disproportionate perturbation compared to the same budget applied to a node with a degree of 2000. Future work could explore dynamic or adaptive budget strategies that account for degree distribution and other structural factors. Tables 42, 43 and, 44 present comparisons of GOttack with established budget settings, specifically considering the percentile of node degrees ($\epsilon$).

**Defenses and Availability.** GOttack can be defended against by attending to node orbit types in GNN aggregations and reducing the importance of neighbors in 1518 orbits. In that case, topology offers new orbits to attack, such as 1519 and 1922, as we study and report in Appendix D. We have created a graph-algebra-based orbit selection scheme in Appendix C.3 that can attack i) a graph of any size and ii) a graph without the periphery orbits of 1518. However, we note that in all the datasets used, nodes belonging to orbit 1518 were particularly common (see Figure 8).

**Limitations.** A notable limitation of our study is the assumption of access to complete graph and training labels, which may not always hold in real-world scenarios. Future work could relax these assumptions

Table 5: Comparison of orbit-based node selection in sequential Nettack phases.

| Dataset | Orbits | % of nodes | % in $1^{st}$ Attack | % in $2^{nd}$ Attack |
|---|---|---|---|---|
| Cora ($h = 0.81$) | 1518 | 24.00% | 77.00% | 71.10% |
| | 1519 | 14.41% | 5.70% | 14.29% |
| | 1819 | 11.59% | 10.00% | 12.50% |
| Citeseer ($h = 0.74$) | 1518 | 21.99% | 51.60% | 61.29% |
| | 1519 | 21.18% | 12.50% | 15.00% |
| | 1819 | 11.56% | 3.29% | 0.00% |
| Polblogs ($h = 0.91$) | 1518 | 9.41% | 97.50% | 60.00% |
| | 1519 | 2.29% | 0.00% | 0.00% |
| | 1819 | 12.93% | 2.50% | 27.50% |
| Pubmed ($h = 0.81$) | 1518 | 20.14% | 25.00% | 10.00% |
| | 1519 | 23.07% | 32.50% | 52.50% |
| | 1618 | 2.24% | 22.50% | 10.00% |
| BlogCatalog ($h = 0.40$) | 1518 | 3.25% | 2.50% | 22.50% |
| | 1519 | 61.57% | 62.50% | 37.50% |
| | 1922 | 19.77% | 35.00% | 40.00% |

to enhance practical applicability while retaining the theoretical contributions. GOttack optimizes candidate node selection for attacks using a topological approach. Our primary hypothesis is that this topological method offers both efficiency and effectiveness. While brute-force methods or gradient-based optimizations can theoretically achieve the same attack success by exhaustively exploring all possible options, these approaches come with significant computational costs. Thus, the tradeoff between attack efficiency and runtime must be carefully balanced. GOttack addresses this by providing a principled strategy based on topology, offering a more computationally feasible solution without sacrificing attack efficacy.

# 6   CONCLUSION

We have identified a key equivalence group for graph nodes based on their topological positions within the graph and demonstrated that gradient-based attack models frequently target this group in their attacks. Our approach, GOttack, introduces a seminal topological strategy in adversarial graph machine learning. By uncovering this previously unexplored vulnerability tied to graph topology, our work highlights the susceptibility of graph neural networks to topology-based attacks and paves the way for developing efficient attack models. GOttack not only enhances misclassification rates but does so in a scalable manner. For future work, we aim to study topological defenses against attack models.

ACKNOWLEDGMENTS

This work has received funding from Canadian NSERC Discovery Grant RGPIN-2020-05665: Data Science on Blockchain and Canadian Research Manitoba grant 324278-352400-2000: Bonafide: Decentralized Services for Sharing and Searching User Generated Data, and from the NSF awards DMS-2204795, OAC-2115094, CNS-2331424, ARL/Army Research Office awards W911NF-24-1-0202 and W911NF-24-2-0114 and NIH award 5RM1HG009034-08.

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
