CONTENTS

## A    RANDOM WALKS TO ORBITS 15 AND 18

We will start by defining hitting time as used in random walks over graphs (Burioni & Cassi, 2005). Our notation is listed in Table 55.

The expected hitting time of a random walk starting from node $v$ and reaching node $w$ is

$$H(v, w) = \mathbb{E}\left[\min\{t \in \mathbb{N} \setminus \{0\} : X_t = w\} \mid X_0 = v\right].$$

By definition, the first hitting time of a node on itself is typically defined as zero, i.e., $H(v, v) = 0$.

**Theorem 1** (Remote Connection Candidates). *Let $H(v, w)$ denote the expected hitting time from node $v$ to node $w$ in $\mathcal{G}$. For any node $v \in V$, nodes in orbits 15 and 18 are the most effective candidates for establishing paths to the most remote parts of $\mathcal{G}$, due to their longer expected hitting times $H(v, w)$ compared to other nodes not in these orbits.*

*Proof:*    It is known that the hitting time $H(v, w)$ is influenced by the structural configuration of the graph and the position of the nodes within it. Kahn et al. have proved that for any regular graph $\mathcal{G} = (\mathcal{V}, \mathcal{E})$, the maximum hitting time $H(v, w)$ is bounded by $O(n^2)$ (Kahn et al., 1989), where $n$ is the number of nodes. The expected time of the first hit from $v$ to $w$ is affected by the structural properties of the graph. Consider a node $w$ appearing in orbits 15, 18, or both. By the definition of graphlets $\mathcal{G}_9$ and $\mathcal{G}_{10}$, we have i) $\deg(w) = 1$ when connected to a single graphlet and ii) the first hitting time of $w$ is influenced by the configuration of nodes adjacent to $w$. As a result, $H(v, w)$ is greater than $H(v, z)$ for all $z \in N(w)$. Extending this, on the shortest path from $v$ to $w$, $H(v, w)$ is the maximum hitting time among all paths from $v$ to any node in $\mathcal{V}$.

Target node $v$ can be at the center or away from the center.

**Center case.** Due to their peripheral placement in graphlets, nodes in orbits 15 and 18 are further away from central nodes or densely connected regions of the graph; their hitting times are expected to approach the upper bound of $n^2$ due to their increased distance from other nodes in $\mathcal{G}$.

**Periphery case.** The target node $v$ may not necessarily be at the center of the graph. However, by the definition of orbits 15 and 18, there is at least one node in these orbits (perhaps the other end of the same graphlet) that has the longest distance to the target node $v$. This further increases the hitting time, as the random walk must navigate through central nodes and potentially longer paths to reach these peripheral nodes. $\square$

## B    GRAPH NEURAL NETWORK

Graph Neural Networks are pivotal in learning node embeddings by capturing node features and their local network neighbourhoods. These embeddings encapsulate the essential characteristics of the nodes into condensed representations by leveraging both the graph structure and feature information from neighbouring nodes. Such embeddings have practical applications across various domains, which are detailed further in the related work section 3.

### B.1    BACKBONE MODELS

In our evaluation of various models, we incorporated baseline models utilizing Graph Neural Networks (GNNs). In this section, we will elucidate the rationale behind each baseline model utilized in our study.

**The Graph Convolutional Network (GCN)**, introduced by  Kipf & Welling (2017), provides a foundational model for understanding and analyzing the vulnerabilities exposed by our proposed attack model. GCN employs a message-passing technique that utilizes the features of neighboring nodes, making it susceptible to adversarial manipulations that can alter node connections and lead to misclassifications. Here, we provide a detailed overview of the GCN architecture, particularly focusing on the structure of its graph convolutional layer (i.e., hidden layer $\mathbf{H}^{(l+1)}$):

$$\mathbf{H}^{(l+1)} = \sigma\left(\tilde{\mathbf{D}}^{-\frac{1}{2}} \tilde{\mathbf{A}} \tilde{\mathbf{D}}^{-\frac{1}{2}} \mathbf{H}^{(l)} \mathbf{W}^{(l)}\right) \qquad (2)$$

In this formulation, $\tilde{\mathbf{A}} = \mathbf{A} + \mathbf{I}_N$ represents the adjacency matrix of the undirected graph augmented with self-loops, and $\mathbf{I}_N$ is the identity matrix. The matrix $\mathbf{W}^{(l)}$ denotes the trainable weight matrix for layer $l$, optimized during backpropagation. $\tilde{\mathbf{D}}_{ii} = \sum_j \tilde{\mathbf{A}}_{ij}$ defines the degree matrix, and $\sigma$ represents a non-linear activation function.

Setting $\mathbf{H}^{(0)} = \mathbf{X}$ (i.e., the initial node features), the GCN model with $l$ layers computes node classifications as follows:

$$\mathbf{Z} = f(\mathbf{A}, \mathbf{X}) = \text{softmax}(\mathbf{A}\mathbf{X}\Theta) = \text{softmax}(\hat{\mathbf{A}}\sigma(\hat{\mathbf{A}}\mathbf{X}\mathbf{W}^1)\mathbf{W}^2) \tag{3}$$

Here, $\hat{\mathbf{A}} = \tilde{\mathbf{D}}^{-\frac{1}{2}}\tilde{\mathbf{A}}\tilde{\mathbf{D}}^{-\frac{1}{2}}$ acts as the renormalized adjacency matrix, $\mathbf{X}$ is the feature matrix, and $\Theta$ includes the set of parameters (e.g., $\mathbf{W}^1$, $\mathbf{W}^2$) to be learned. The matrix $\mathbf{Z} \in \mathbb{R}^{n \times c}$ represents the probabilities of $C = \{c_i\}$ for each node $v \in V$, with each row indicating the likelihood of each class label $c$ for a node.

GCNs operate under both inductive and transductive settings. We focus on transductive classification, where all node connections and features are accessible during the training phase. For such tasks, the softmax function normalizes the final output matrix $\mathbf{Z}$, and the cross-entropy loss $L = -\sum_{v \in \mathcal{V}_{\text{tr}}} \log \mathbf{Z}_{v,y_v}$ is calculated, comparing the predicted probabilities to the true labels, where $y_v$ represents the true class of node $v$. Weight updates are performed using gradient descent optimization algorithms, such as Adam (Kingma & Ba, 2015).

**The Graph Isomorphism Network(GIN)** is another type of GNN designed to respect graph isomorphisms. It produces the same embedding for isomorphic graphs. Although the learning process is similar to the GCN, it uses a different aggregation function. The following equation (Xu et al., 2019b) shows the calculation of the hidden layer $\mathbf{H}^{(l+1)}$. where, $\epsilon^{(l)}$ is a trainable parameter and $\mathbf{MLP}^{(l)}$ is a multi layer perception.

$$\mathbf{H}^{(l+1)} = \sigma\left((1 + \epsilon^{(l)}) \cdot \mathbf{MLP}^{(l)}(\mathbf{H}^{(l)})\right) \tag{4}$$

**GraphSAGE** is a type of GNN that also gathers information from the neighboring nodes like GCN but in a slightly different way. The following equation (Hamilton et al., 2017) shows the aggregation of node feature $\mathbf{X}$ using a sampling strategy, where $\mathbf{AGG}$ is an aggregation function such as mean or max pooling.

$$\mathbf{H}_v^{(l+1)} = \mathbf{AGG}\left(\{\mathbf{H}_u^{(l)}, \forall u \in \mathcal{N}(v)\}\right) \tag{5}$$

## B.2    BASELINE MODELS

In this subsection, we present a brief idea of the existing state-of-the-art adversarial techniques that we have considered as comparable baseline methods.

**Nettack (Zügner et al., 2018)** is a targeted attack method to enforce misclassification on the target nodes using edge and feature perturbations, which can handle both direct and influence attacks.

**FGA attack (Chen et al., 2018)** is a targeted attack method to enforce misclassification on the target nodes using graph perturbations by generating adversarial graph networks based on the gradient information of GCN.

**SGAttack (Li et al., 2023)** is a targeted attack method to enforce misclassification on the target nodes using features or edge perturbations through a multi-stage attack framework, which needs only a much smaller subgraph.

**PRBCD attack (Geisler et al., 2021)** is a sparsity-aware first-order optimization strategy that effectively targets GNNs by optimizing parameters in a manner that scales to large graphs.

**Random attack** randomly selects non-adjacent node pairs and introduces fake edges or removing existing edges between them.

### B.3 Defense Methods

We provide a brief overview of existing state-of-the-art adversarial defense techniques that effectively counteract adversarial attacks. These defense methods are considered to demonstrate the effectiveness of the GOttack approach.

**RobustGCN (Zhu et al., 2019)** improves GCN robustness by representing nodes as Gaussian distributions to absorb perturbations. Moreover, it uses a variance-based attention mechanism to assign weights to node neighborhoods.

**GCN-Jaccard (Wu et al., 2019)** enhances GCN robustness by removing edges between nodes with low Jaccard similarity.

**GCN-SVD (Entezari et al., 2020)** leverages low-rank approximations of graphs to mitigate the effects of adversarial attacks by focusing on high-rank singular components that are more susceptible to perturbations.

**MedianGCN (Chen et al., 2021)** enhances the robustness by replacing the traditional weighted mean aggregation scheme with a median-based approach.

## C Further analysis of graphlets and orbits

In this section, we present an overview of graphlets and their topological properties, followed by the orbit $1518$, which is the topological node embedding. Finally, we discuss the hierarchy of orbits based on relation algebra.

### C.1 Topological properties of graphlets and orbits

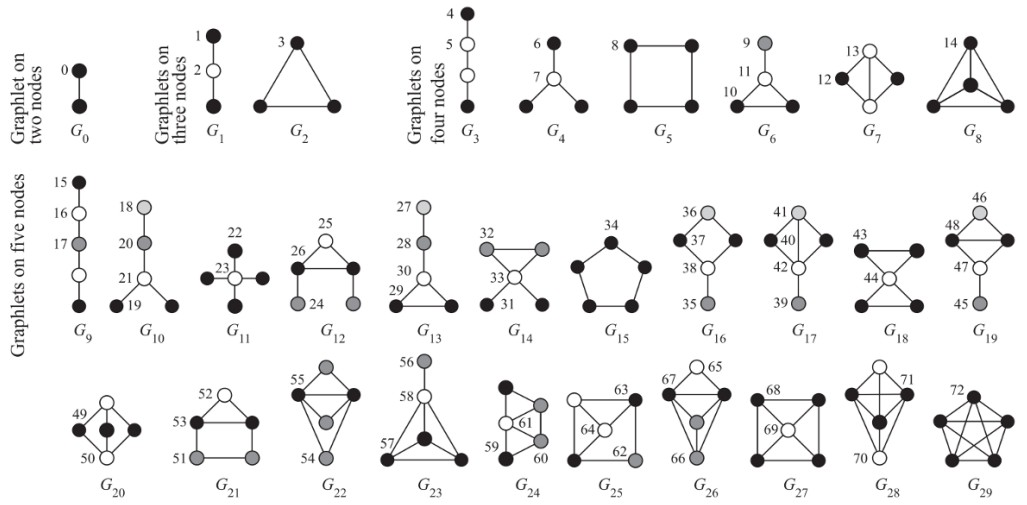

Figure 6: All 30 Graphlets with two to five nodes with the automorphism orbits of each graphlet (Hocevar & Demsar, 2014).

Graphlets are small, connected, non-isomorphic induced subgraphs that represent topological patterns of interconnection between $k$ nodes in a graph (Feng & Chen, 2020). Figure 6 illustrates all graphlets with two to five nodes, including 9 different graphlets with 2 to 4 nodes and up to 30 graphlets, ranging from $\mathcal{G}_0$ to $\mathcal{G}_{29}$, with 5 nodes. The 30 graphlets with 5 nodes and their corresponding 73 orbits. Each orbit represents a unique position within a graphlet as determined by its automorphism group. The total of 73 orbits corresponds to the feature dimensions of $\mathbf{GOV}_v$. However, the structural properties of a network can be represented by the frequency of graphlet appearances within the network. The orbits define the unique characteristics of the nodes within a graphlet, express different connection modes between nodes, and contain abundant high-order structural information (Feng & Chen, 2020).

Table 6: Comparison of orbit-based node selection in sequential FGA phases.

| Dataset | Orbits | % of nodes | % in $1^{st}$ Attack | % in $2^{nd}$ Attack |
|---------|--------|-----------|---------------------|---------------------|
| Cora | 1518 | 24.00% | 60.00% | 45.00% |
| | 1519 | 14.41% | 10.00 % | 20.00% |
| | 1819 | 11.59% | 15.00 % | 02.50% |
| Citeseer | 1518 | 21.99% | 20.00% | 15.00% |
| | 1519 | 21.18% | 32.50 % | 20.00% |
| | 1819 | 11.56% | 15.00 % | 05.00% |
| Polblogs | 1518 | 09.41% | 37.50% | 37.50% |
| | 1519 | 02.29% | 00.00% | 00.00% |
| | 1819 | 12.93% | 62.50 % | 00.00% |

Table 7: Comparison of orbit-based node selection in sequential SGA phases.

| Dataset | Orbits | % of nodes | % in $1^{st}$ Attack | % in $2^{nd}$ Attack |
|---------|--------|-----------|---------------------|---------------------|
| Cora | 1518 | 24.00% | 51.00% | 47.00% |
| | 1519 | 14.41% | 12.00 % | 13.00% |
| | 1819 | 11.59% | 6.00 % | 07.00% |
| Citeseer | 1518 | 21.99% | 37.00% | 32.00% |
| | 1519 | 21.18% | 27.00 % | 26.00% |
| | 1819 | 11.56% | 19.00 % | 23.00% |
| Polblogs | 1518 | 09.41% | 52.00% | 46.00% |
| | 1519 | 02.29% | 08.00% | 08.00% |
| | 1819 | 12.93% | 07.00 % | 10.00% |

For instance, consider the graphlet $\mathcal{G}_{11}$ from the set of five-node graphlets. It can be observed that $\mathcal{G}_{11}$ is a star graph where the node labeled with orbit 23 is the central node, while the remaining nodes, labeled with orbit 22, are leaf nodes. The topological context of a node can be determined by counting the orbits of that node.

Orbit counting is computationally expensive because the number of orbits in a graph grows exponentially with the size of the original graph. However, many advanced algorithms have been developed to mitigate the complexity of computing graphlets and orbits (Kloks et al., 2000; Hocevar & Demsar, 2014; Melckenbeeck et al., 2018; Kowaluk et al., 2013). Consequently, numerous studies leverage the topological insights provided by graphlet and orbit counting across various domains. For example, Feng & Chen (2020) used orbits counting as high-order structural features of nodes to learn efficient node representations, which were then utilized to enhance link prediction tasks.

### C.2    ADDITIONAL ANALYSIS OF 1518 ORBIT NODES

Connecting the 1518 orbit node with the target node $v \in \mathcal{V}_T \subseteq \mathcal{V}$ has been demonstrated to significantly impact the prediction accuracy of GNNs $f_\theta$ on the node $v$. This impact is also reflected in the effectiveness of various attack methods, such as FGA and SGA, which frequently select the 1518 node for graph manipulations (e.g., adding or removing edges with $v$). Table 6 shows that in the Cora dataset, up to 60% and 45% of the nodes manipulated by FGA in the first and second attacks, respectively. Likewise, the node labeled as 1518 is consistently the primary target of SGA in both the first and second attack scenarios across all datasets (ref. Table 7). For instance, in the Citeseer dataset, SGA manipulates nodes in the first and second attacks at rates of 37% and 32%, respectively. Similarly, in the Polblogs dataset, these percentages are 52% and 46% for the first and second attacks.

### C.3    ORBIT HIERARCHY

Based on the analysis of the impact of GOttack discussed in 5.1 and experimental results, we have concluded that the 1518 orbit is crucial for attacking the GNNs. This raises the question: *what happens if the* 1518 *orbit node does not exist in the network?* To address this, we have developed an orbit hierarchy (or orbit transition), as shown in Figure 7, based on relational algebra. From this hierarchy, we can see that if the 15 and 18 orbit nodes are absent, the attack model can instead select nodes from orbits 4 and 6, respectively. Similarly, if nodes from orbits 4 and 6 are not present, nodes from orbit 1 can be chosen.

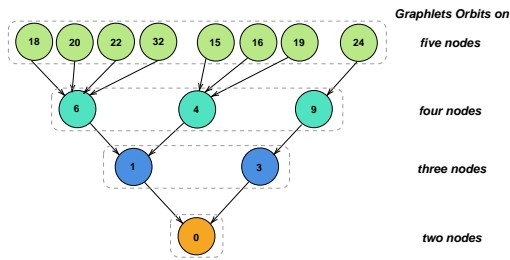

Figure 7: Orbit hierarchy.

To better understand how the orbit transition approach works, let us consider the following example.

**Example 2.** *Consider a* 5*-node graphlet* $\mathcal{G}_{10}$*, where orbits* 18*,* 19*,* 20 *and* 21 *are present (see Figure 6). If orbit* 18 *is removed from* $\mathcal{G}_{10}$*, then it becomes the* 4*-node graphlet* $\mathcal{G}_4$*. Suppose* $\mathcal{G}_{10} = (\mathcal{V}, \mathcal{E})$ *with*

$\mathcal{V} = \{v_1, v_2, v_3, v_4, v_5\}$ and edges $\mathcal{E}$, where node $v \in \mathcal{V}$ belongs to one of the orbits $Orb_{18}^v$, $Orb_{19}^v$, $Orb_{20}^v$ and $Orb_{21}^v$. If the removal operation $\rho$ removes orbit node $Orb_{18}^{v_1}$, resulting in $\mathcal{G}_4 = (\mathcal{V}', \mathcal{E}')$, where $\mathcal{V}' = \mathcal{V} - \{v_1\}$. Therefore, by using the projection operations ($\pi$) the transition of orbits can be written as: $\pi_{Orb_{21}^v}(\mathcal{G}_{10}) \rightarrow Orb_7^v$ and $\pi_{Orb_{19}^v, Orb_{20}^v}(\mathcal{G}_{10}) \rightarrow Orb_6^v$, indicating that orbit $Orb_{21}^v$ transitions to orbit $Orb_7^v$, and orbit $Orb_{19}^v$ and $Orb_{20}^v$ now belong to orbit $Orb_6^v$.

# D    EXPERIMENTAL RESULTS ON ALL DATASETS

In this section, we present the results of our experiments. In Table 8, we demonstrate the success of our GOttack strategy across five different datasets, with the results averaged over these datasets. Detailed results for each individual dataset are provided in the subsequent Tables 9 to 21. We present the results of various attack strategies, including GOttack and its variants such as 1819, 1519, and 1922. The results are presented as the mean and standard deviation computed from five independent runs.

Table 8: Summary of attack results averaged over five datasets.

| Budget | 1 | 2 | 3 | 4 | 5 |
|---|---|---|---|---|---|
| Random | 0.15 | 0.22 | 0.28 | 0.31 | 0.38 |
| Nettack | 0.43 | 0.51 | 0.56 | 0.61 | 0.61 |
| GOttack | 0.44 | 0.54 | **0.61** | **0.65** | **0.68** |
| 1819 orbit attack | 0.35 | 0.42 | 0.46 | 0.47 | 0.50 |
| 1519 orbit attack | 0.37 | 0.41 | 0.44 | 0.45 | 0.46 |
| 1922 orbit attack | 0.35 | 0.40 | 0.41 | 0.44 | 0.45 |
| FGA | 0.37 | 0.47 | 0.52 | 0.53 | 0.56 |
| SGAttack | **0.47** | **0.57** | 0.58 | 0.62 | 0.64 |
| PRBCD | 0.43 | 0.54 | 0.59 | 0.64 | 0.67 |

## D.1    NON-DEFENSE GNN BACKBONE

In this section, we demonstrate the effectiveness of the proposed attack model; we evaluate the performance of GOttack alongside various adversarial attack techniques on different non-defence GNN backbones as discussed further in Section B.1, namely GCN, GIN, and GraphSAGE. Based on the analysis of the results, as shown in from Table 9 to Table 21, we conclude that the proposed GOttack achieves the best performance with GSAGE in 13 out of 20 tasks while attaining the second-highest performance with GCN.

### D.1.1    ATTACK RESULTS ON GCN

We present the performance of various adversarial attack techniques on GCN. Tables 9 to 13 display the misclassification rates for various datasets with 1 to 5 perturbed edges, respectively. The proposed GOttack method achieved the highest performance in 8 out of 25 tasks, while PRBCD attained the highest performance in 10 tasks. Additionally, SGA ranked third in 5 tasks.

### D.1.2    ATTACK RESULTS ON GRAPHSAGE

The results of adversarial techniques on GraphSAGE are presented in Tables 14 to 17. In summary, the proposed GOttack outperforms all baseline adversarial techniques by achieving the highest misclassification rate in 13 out of 20 tasks. In addition, SGA performs the second highest performance in 5 out of 20 tasks. Likewise, both PRBCD and Nettack have the third highest performance, with scores of 1 in their respective tasks.

### D.1.3    ATTACK RESULTS ON GIN

In this subsection, we discuss the performance of the proposed GOttack along with various adversarial attack techniques on GIN, as shown in Tables 18 to 21. Results show that GOttack outperforms all

Table 9: Misclassification rate (↑) on Cora with budget $\Delta = 1$ to 5: GOttack achieves performance in 1 out of 5 tasks (GCN model).

| Budget → | 1 | 2 | 3 | 4 | 5 |
|---|---|---|---|---|---|
| Random | 0.02 ± 0.029 | 0.05 ± 0.000 | 0.15 ± 0.100 | 0.23 ± 0.104 | 0.27 ± 0.076 |
| Nettack | 0.34 ± 0.060 | 0.50 ± 0.068 | 0.58 ± 0.057 | 0.66 ± 0.029 | 0.70 ± 0.011 |
| GOttack | **0.41 ± 0.052** | 0.54 ± 0.049 | 0.62 ± 0.033 | 0.66 ± 0.042 | 0.71 ± 0.052 |
| 1819 orbit attack | 0.32 ± 0.063 | 0.45 ± 0.073 | 0.540 ± 0.076 | 0.60 ± 0.051 | 0.65 ± 0.025 |
| 1519 orbit attack | 0.37 ± 0.029 | 0.47 ± 0.080 | 0.57 ± 0.062 | 0.62 ± 0.074 | 0.64 ± 0.072 |
| 1922 orbit attack | 0.34 ± 0.078 | 0.43 ± 0.074 | 0.50 ± 0.075 | 0.57 ± 0.080 | 0.58 ± 0.069 |
| FGA | 0.32 ± 0.057 | 0.44 ± 0.080 | 0.53 ± 0.033 | 0.58 ± 0.048 | 0.61 ± 0.042 |
| SGA | 0.41 ± 0.058 | 0.59 ± 0.050 | 0.63 ± 0.070 | 0.69 ± 0.020 | 0.69 ± 0.060 |
| PRBCD | 0.41 ± 0.060 | **0.61 ± 0.072** | **0.64 ± 0.038** | **0.73 ± 0.014** | **0.76 ± 0.025** |

Table 10: Misclassification rate (↑) on Citeseer with budget $\Delta = 1$ to 5: GOttack achieves highest performance in 4 out of 5 tasks (GCN model).

| Budget → | 1 | 2 | 3 | 4 | 5 |
|---|---|---|---|---|---|
| Random | 0.02 ± 0.014 | 0.06 ± 0.038 | 0.17 ± 0.075 | 0.25 ± 0.066 | 0.30 ± 0.109 |
| Nettack | 0.46 ± 0.045 | 0.61 ± 0.038 | 0.68 ± 0.027 | 0.74 ± 0.021 | 0.76 ± 0.014 |
| GOttack | **0.46 ± 0.034** | **0.63 ± 0.037** | **0.72 ± 0.054** | **0.76 ± 0.063** | **0.78 ± 0.042** |
| 1819 orbit attack | 0.44 ± 0.058 | 0.56 ± 0.038 | 0.67 ± 0.045 | 0.71 ± 0.029 | 0.74 ± 0.033 |
| 1519 orbit attack | 0.45 ± 0.050 | **0.63 ± 0.031** | 0.68 ± 0.018 | 0.73 ± 0.037 | 0.76 ± 0.021 |
| 1922 orbit attack | 0.40 ± 0.033 | 0.55 ± 0.041 | 0.67 ± 0.033 | 0.70 ± 0.033 | 0.72 ± 0.0401 |
| FGA | 0.31 ± 0.107 | 0.48 ± 0.040 | 0.62 ± 0.068 | 0.64 ± 0.045 | 0.68 ± 0.069 |
| SGA | 0.41 ± 0.060 | 0.60 ± 0.050 | 0.63 ± 0.068 | 0.69 ± 0.022 | 0.69 ± 0.057 |
| PRBCD | 0.46 ± 0.044 | 0.57 ± 0.014 | 0.68 ± 0.038 | 0.70 ± 0.00 | 0.73 ± 0.014 |

Table 11: Misclassification rate (↑) on Polblogs with budget $\Delta = 1$ to 5: GOttack achieves performance in 0 out of 5 tasks (GCN model).

| Budget → | 1 | 2 | 3 | 4 | 5 |
|---|---|---|---|---|---|
| Random | 0.12 ± 0.043 | 0.09 ± 0.052 | 0.15 ± 0.075 | 0.19 ± 0.104 | 0.20 ± 0.101 |
| Nettack | 0.38 ± 0.040 | 0.43 ± 0.058 | 0.46 ± 0.063 | 0.50 ± 0.082 | 0.51 ± 0.072 |
| GOttack | 0.41 ± 0.086 | 0.46 ± 0.08 | 0.51 ± 0.089 | 0.52 ± 0.089 | 0.55 ± 0.077 |
| 1819 orbit attack | 0.26 ± 0.193 | 0.35 ± 0.100 | 0.36 ± 0.183 | 0.41 ± 0.152 | 0.46 ± 0.089 |
| 1519 orbit attack | 0.30 ± 0.198 | 0.30 ± 0.203 | 0.30 ± 0.164 | 0.32 ± 0.179 | 0.33 ± 0.178 |
| 1922 orbit attack | 0.22 ± 0.184 | 0.25 ± 0.149 | 0.25 ± 0.156 | 0.26 ± 0.080 | 0.26 ± 0.038 |
| FGA | 0.31 ± 0.098 | 0.40 ± 0.078 | 0.45 ± 0.097 | 0.45 ± 0.089 | 0.48 ± 0.087 |
| SGA | 0.37 ± 0.075 | **0.56 ± 0.051** | **0.56 ± 0.037** | **0.57 ± 0.068** | **0.65 ± 0.020** |
| PRBCD | **0.42 ± 0.072** | 0.48 ± 0.058 | 0.55 ± 0.043 | 0.55 ± 0.043 | 0.58 ± 0.052 |

Table 12: Misclassification rate (↑) on BlogCatalog with budget $\Delta = 1$ to 5: GOttack achieves performance in 0 out of 5 tasks (GCN model).

| Budget → | 1 | 2 | 3 | 4 | 5 |
|---|---|---|---|---|---|
| Random | 0.12 ± 0.092 | 0.17 ± 0.071 | 0.20 ± 0.097 | 0.25 ± 0.085 | 0.30 ± 0.082 |
| Nettack | 0.20 ± 0.027 | 0.25 ± 0.085 | 0.37 ± 0.072 | 0.40 ± 0.077 | 0.45 ± 0.065 |
| GOttack | 0.22 ± 0.040 | 0.25 ± 0.050 | 0.35 ± 0.062 | 0.37 ± 0.083 | 0.45 ± 0.075 |
| FGA | 0.10 ± 0.047 | 0.20 ± 0.083 | 0.27 ± 0.050 | 0.35 ± 0.031 | 0.37 ± 0.018 |
| SGA | 0.24 ± 0.030 | 0.28 ± 0.05 | 0.32 ± 0.050 | 0.37 ± 0.040 | 0.41 ± 0.070 |
| PRBCD | **0.33 ± 0.052** | **0.39 ± 0.072** | **0.41 ± 0.115** | **0.45 ± 0.146** | **0.50 ± 0.153** |

Table 13: Misclassification rate (↑) on Pubmed with budget $\Delta = 1$ to 5: GOttack achieves highest performance in 3 out of 5 tasks (GCN model).

| Budget → | 1 | 2 | 3 | 4 | 5 |
|---|---|---|---|---|---|
| Random | 0.20 ± 0.050 | 0.21 ± 0.014 | 0.27 ± 0.109 | 0.24 ± 0.072 | 0.25 ± 0.029 |
| Nettack | 0.50 ± 0.045 | 0.62 ± 0.037 | **0.67 ± 0.052** | 0.72 ± 0.032 | 0.75 ± 0.025 |
| GOttack | **0.57 ± 0.012** | 0.60 ± 0.037 | 0.65 ± 0.020 | **0.72 ± 0.012** | **0.75 ± 0.017** |
| FGA | 0.32 ± 0.025 | 0.48 ± 0.037 | 0.51 ± 0.012 | 0.50 ± 0.075 | 0.57 ± 0.050 |
| SGA | 0.57 ± 0.040 | **0.65 ± 0.067** | 0.65 ± 0.053 | 0.69 ± 0.021 | 0.72 ± 0.031 |
| PRBCD | 0.52 ± 0.038 | 0.63 ± 0.025 | 0.64 ± 0.043 | 0.70 ± 0.058 | 0.72 ± 0.038 |

Table 14: Misclassification rate (↑) on Cora with budget $\Delta = 1$ to 5: GOttack achieves highest performance in 4 out of 5 tasks (GraphSAGE model).

| Budget → | 1 | 2 | 3 | 4 | 5 |
|---|---|---|---|---|---|
| Random | 0.19 ± 0.115 | 0.30 ± 0.128 | 0.39 ± 0.052 | 0.39 ± 0.076 | 0.47 ± 0.100 |
| Nettack | 0.58 ± 0.045 | 0.66 ± 0.042 | 0.70 ± 0.067 | 0.74 ± 0.054 | 0.77 ± 0.027 |
| GOttack | 0.59 ± 0.055 | **0.78 ± 0.054** | **0.86 ± 0.014** | **0.88 ± 0.029** | **0.92 ± 0.033** |
| 1819 orbit attack | 0.5 ± 0.053 | 0.52 ± 0.080 | 0.56 ± 0.135 | 0.52 ± 0.083 | 0.53 ± 0.080 |
| 1519 orbit attack | 0.52 ± 0.031 | 0.54 ± 0.014 | 0.58 ± 0.037 | 0.59 ± 0.076 | 0.54 ± 0.065 |
| 1922 orbit attack | 0.52 ± 0.045 | 0.59 ± 0.049 | 0.55 ± 0.04 | 0.57 ± 0.085 | 0.59 ± 0.045 |
| FGA | 0.54 ± 0.089 | 0.57 ± 0.082 | 0.68 ± 0.072 | 0.70 ± 0.108 | 0.71 ± 0.029 |
| SGA | **0.61 ± 0.060** | 0.68 ± 0.060 | 0.67 ± 0.090 | 0.78 ± 0.040 | 0.73 ± 0.080 |
| PRBCD | 0.35 ± 0.025 | 0.54 ± 0.038 | 0.59 ± 0.095 | 0.76 ± 0.038 | 0.74 ± 0.113 |

Table 15: Misclassification rate (↑) on Citeseer with budget $\Delta = 1$ to 5: GOttack achieves highest performance in 3 out of 5 tasks (GraphSAGE model).

| Budget → | 1 | 2 | 3 | 4 | 5 |
|---|---|---|---|---|---|
| Random | 0.30 ± 0.014 | 0.43 ± 0.058 | 0.47 ± 0.025 | 0.46 ± 0.038 | 0.46 ± 0.063 |
| Nettack | **0.66 ± 0.091** | 0.68 ± 0.123 | 0.72 ± 0.069 | 0.77 ± 0.065 | 0.74 ± 0.082 |
| GOttack | 0.61 ± 0.093 | **0.83 ± 0.062** | **0.92 ± 0.029** | **0.95 ± 0.047** | 0.97 ± 0.041 |
| 1819 orbit attack | 0.52 ± 0.029 | 0.56 ± 0.052 | 0.62 ± 0.043 | 0.600 ± 0.057 | 0.65 ± 0.047 |
| 1519 orbit attack | 0.55 ± 0.085 | 0.55 ± 0.12 | 0.62 ± 0.107 | 0.59 ± 0.038 | 0.63 ± 0.057 |
| 1922 orbit attack | 0.50 ± 0.060 | 0.62 ± 0.060 | 0.54 ± 0.060 | 0.64 ± 0.042 | 0.60 ± 0.083 |
| FGA | 0.60 ± 0.113 | 0.65 ± 0.079 | 0.74 ± 0.072 | 0.76 ± 0.052 | 0.76 ± 0.082 |
| SGA | 0.60 ± 0.060 | 0.68 ± 0.060 | | 0.78 ± 0.037 | 0.73 ± 0.080 |
| PRBCD | 0.35 ± 0.043 | 0.69 ± 0.014 | 0.89 ± 0.138 | 0.95 ± 0.075 | **0.99 ± 0.132** |

Table 16: Misclassification rate (↑) on Polblogs with budget $\Delta = 1$ to 5: GOttack achieves performance in 1 out of 5 tasks (GraphSAGE model).

| Budget → | 1 | 2 | 3 | 4 | 5 |
|---|---|---|---|---|---|
| Random | 0.15 ± 0.132 | 0.15 ± 0.139 | 0.16 ± 0.118 | 0.17 ± 0.109 | 0.18 ± 0.151 |
| Nettack | 0.29 ± 0.029 | 0.34 ± 0.078 | 0.36 ± 0.074 | 0.39 ± 0.068 | 0.38 ± 0.115 |
| GOttack | 0.29 ± 0.038 | 0.36 ± 0.054 | 0.44 ± 0.072 | 0.49 ± 0.084 | **0.54 ± 0.099** |
| 1819 orbit attack | 0.21 ± 0.091 | 0.25 ± 0.077 | 0.29 ± 0.042 | 0.30 ± 0.113 | 0.30 ± 0.066 |
| 1519 orbit attack | 0.23 ± 0.112 | 0.24 ± 0.102 | 0.24 ± 0.08 | 0.22 ± 0.053 | 0.24 ± 0.070 |
| 1922 orbit attack | 0.20 ± 0.074 | 0.20 ± 0.074 | 0.21 ± 0.045 | 0.24 ± 0.08 | 0.25 ± 0.093 |
| FGA | 0.22 ± 0.081 | 0.28 ± 0.071 | 0.32 ± 0.084 | 0.33 ± 0.096 | 0.37 ± 0.110 |
| SGA | **0.35 ± 0.057** | **0.38 ± 0.065** | **0.51 ± 0.093** | **0.52 ± 0.079** | 0.52 ± 0.030 |
| PRBCD | 0.08 ± 0.014 | 0.10 ± 0.043 | 0.13 ± 0.014 | 0.13 ± 0.052 | 0.13 ± 0.029 |

Table 17: Misclassification rate (↑) on Pubmed with budget $\Delta = 1$ to 5: GOttack achieves highest performance in 5 out of 5 tasks (GraphSAGE model).

| Budget → | 1 | 2 | 3 | 4 | 5 |
|---|---|---|---|---|---|
| Random | 0.14 ± 0.029 | 0.20 ± 0.038 | 0.30 ± 0.090 | 0.30 ± 0.072 | 0.30 ± 0.095 |
| Nettack | 0.52 ± 0.090 | 0.60 ± 0.050 | 0.65 ± 0.07 | 0.77 ± 0.070 | 0.52 ± 0.060 |
| GOttack | **0.52 ± 0.080** | **0.67 ± 0.060** | **0.70 ± 0.08** | **0.77 ± 0.060** | **0.77 ± 0.050** |
| FGA | 0.42 ± 0.030 | 0.55 ± 0.050 | 0.60 ± 0.070 | 0.62 ± 0.060 | 0.70 ± 0.040 |
| SGA | 0.30 ± 0.000 | 0.47 ± 0.001 | 0.57 ± 0.000 | 0.55 ± 0.000 | 0.55 ± 0.000 |
| PRBCD | 0.38 ± 0.025 | 0.48 ± 0.095 | 0.54 ± 0.038 | 0.59 ± 0.052 | 0.59 ± 0.029 |

baseline adversarial techniques in 7 out of 20 tasks, while SGA, PRBCD and Nettack achieve the performance in 6, 5 and 1 tasks, respectively.

Table 18: Misclassification rate (↑) on Cora with budget $\Delta = 1$ to 5: GOttack achieves performance in 0 out of 5 tasks (GIN model).

| Budget → | 1 | 2 | 3 | 4 | 5 |
|---|---|---|---|---|---|
| Random | $0.17 \pm 0.303$ | $0.36 \pm 0.292$ | $0.57 \pm 0.066$ | $0.61 \pm 0.014$ | $0.65 \pm 0.063$ |
| Nettack | $\underline{0.46 \pm 0.108}$ | $\underline{0.55 \pm 0.116}$ | $\mathbf{0.62 \pm 0.084}$ | $\underline{0.64 \pm 0.091}$ | $\underline{0.66 \pm 0.049}$ |
| GOttack | $0.37 \pm 0.037$ | $0.48 \pm 0.076$ | $0.54 \pm 0.074$ | $0.59 \pm 0.101$ | $0.64 \pm 0.076$ |
| 1819 orbit attack | $0.34 \pm 0.089$ | $0.32 \pm 0.055$ | $0.38 \pm 0.063$ | $0.36 \pm 0.084$ | $0.40 \pm 0.112$ |
| 1519 orbit attack | $0.36 \pm 0.029$ | $0.36 \pm 0.048$ | $0.36 \pm 0.091$ | $0.37 \pm 0.041$ | $0.40 \pm 0.069$ |
| 1922 orbit attack | $0.35 \pm 0.085$ | $0.34 \pm 0.091$ | $0.36 \pm 0.065$ | $0.38 \pm 0.121$ | $0.38 \pm 0.096$ |
| FGA | $0.40 \pm 0.095$ | $0.46 \pm 0.058$ | $0.52 \pm 0.027$ | $0.57 \pm 0.027$ | $0.64 \pm 0.033$ |
| SGA | $\mathbf{0.57 \pm 0.060}$ | $\mathbf{0.63 \pm 0.060}$ | $\underline{0.61 \pm 0.065}$ | $0.57 \pm 0.090$ | $0.61 \pm 0.040$ |
| PRBCD | $0.36 \pm 0.104$ | $0.47 \pm 0.063$ | $0.59 \pm 0.095$ | $\mathbf{0.67 \pm 0.063}$ | $\mathbf{0.83 \pm 0.113}$ |

Table 19: Misclassification rate (↑) on Citeseer with budget $\Delta = 1$ to 5: GOttack achieves highest performance in 4 out of 5 tasks (GIN model).

| Budget → | 1 | 2 | 3 | 4 | 5 |
|---|---|---|---|---|---|
| Random | $0.10 \pm 0.038$ | $0.24 \pm 0.038$ | $0.35 \pm 0.043$ | $0.42 \pm 0.025$ | $0.46 \pm 0.063$ |
| Nettack | $\underline{0.57 \pm 0.049}$ | $0.60 \pm 0.084$ | $\underline{0.64 \pm 0.033}$ | $\underline{0.70 \pm 0.035}$ | $\underline{0.74 \pm 0.065}$ |
| GOttack | $\mathbf{0.57 \pm 0.040}$ | $\underline{0.60 \pm 0.077}$ | $\mathbf{0.66 \pm 0.099}$ | $\mathbf{0.74 \pm 0.104}$ | $\mathbf{0.76 \pm 0.074}$ |
| 1819 orbit attack | $0.42 \pm 0.136$ | $0.45 \pm 0.064$ | $0.47 \pm 0.094$ | $0.43 \pm 0.080$ | $0.50 \pm 0.033$ |
| 1519 orbit attack | $0.39 \pm 0.140$ | $0.44 \pm 0.119$ | $0.43 \pm 0.063$ | $0.48 \pm 0.055$ | $0.44 \pm 0.089$ |
| 1922 orbit attack | $0.45 \pm 0.066$ | $0.47 \pm 0.091$ | $0.39 \pm 0.101$ | $0.42 \pm 0.014$ | $0.46 \pm 0.114$ |
| FGA | $0.44 \pm 0.102$ | $0.57 \pm 0.104$ | $0.56 \pm 0.084$ | $0.64 \pm 0.108$ | $0.64 \pm 0.054$ |
| SGA | $0.57 \pm 0.060$ | $\mathbf{0.62 \pm 0.061}$ | $0.61 \pm 0.067$ | $0.56 \pm 0.090$ | $0.60 \pm 0.040$ |
| PRBCD | $0.43 \pm 0.025$ | $0.52 \pm 0.072$ | $0.53 \pm 0.025$ | $0.55 \pm 0.050$ | $0.65 \pm 0.050$ |

Table 20: Misclassification rate (↑) on Polblogs with budget $\Delta = 1$ to 5: GOttack achieves performance in 0 out of 5 tasks (GIN model).

| Budget → | 1 | 2 | 3 | 4 | 5 |
|---|---|---|---|---|---|
| Random | $0.17 \pm 0.043$ | $0.20 \pm 0.066$ | $0.20 \pm 0.115$ | $0.22 \pm 0.113$ | $0.24 \pm 0.080$ |
| Nettack | $0.13 \pm 0.029$ | $0.20 \pm 0.048$ | $0.29 \pm 0.055$ | $0.32 \pm 0.048$ | $0.38 \pm 0.061$ |
| GOttack | $0.15 \pm 0.086$ | $0.23 \pm 0.048$ | $0.28 \pm 0.011$ | $0.32 \pm 0.042$ | $0.34 \pm 0.029$ |
| 1819 orbit attack | $0.15 \pm 0.048$ | $0.16 \pm 0.022$ | $0.15 \pm 0.037$ | $0.14 \pm 0.049$ | $0.16 \pm 0.065$ |
| 1519 orbit attack | $0.12 \pm 0.053$ | $0.16 \pm 0.068$ | $0.18 \pm 0.056$ | $0.16 \pm 0.052$ | $0.18 \pm 0.045$ |
| 1922 orbit attack | $0.15 \pm 0.031$ | $0.12 \pm 0.066$ | $0.19 \pm 0.052$ | $0.17 \pm 0.048$ | $0.18 \pm 0.040$ |
| FGA | $0.14 \pm 0.029$ | $0.14 \pm 0.048$ | $0.15 \pm 0.031$ | $0.17 \pm 0.037$ | $0.20 \pm 0.040$ |
| SGA | $\mathbf{0.35 \pm 0.080}$ | $\mathbf{0.35 \pm 0.070}$ | $\underline{0.37 \pm 0.120}$ | $\underline{0.40 \pm 0.076}$ | $\mathbf{0.50 \pm 0.097}$ |
| PRBCD | $\underline{0.33 \pm 0.063}$ | $\mathbf{0.38 \pm 0.076}$ | $\mathbf{0.41 \pm 0.029}$ | $\mathbf{0.43 \pm 0.029}$ | $\underline{0.47 \pm 0.100}$ |

Table 21: Misclassification rate (↑) on Pubmed with budget $\Delta = 1$ to 5: GOttack achieves highest performance in 3 out of 5 tasks (GIN model).

| Budget → | 1 | 2 | 3 | 4 | 5 |
|---|---|---|---|---|---|
| Random | $0.20 \pm 0.052$ | $0.22 \pm 0.090$ | $0.30 \pm 0.058$ | $0.31 \pm 0.080$ | $0.32 \pm 0.066$ |
| Nettack | $0.47 \pm 0.020$ | $0.60 \pm 0.080$ | $0.60 \pm 0.070$ | $0.57 \pm 0.070$ | $0.52 \pm 0.060$ |
| GOttack | $\mathbf{0.55 \pm 0.050}$ | $\underline{0.60 \pm 0.040}$ | $\mathbf{0.67 \pm 0.060}$ | $\mathbf{0.67 \pm 0.080}$ | $\underline{0.67 \pm 0.070}$ |
| FGA | $\underline{0.52 \pm 0.001}$ | $\mathbf{0.67 \pm 0.060}$ | $0.62 \pm 0.030$ | $0.52 \pm 0.010$ | $0.52 \pm 0.020$ |
| SGA | $0.47 \pm 0.018$ | $0.60 \pm 0.011$ | $\underline{0.66 \pm 0.022}$ | $\underline{0.67 \pm 0.021}$ | $\mathbf{0.68 \pm 0.014}$ |
| PRBCD | $0.43 \pm 0.066$ | $0.55 \pm 0.090$ | $0.53 \pm 0.014$ | $0.56 \pm 0.014$ | $0.59 \pm 0.080$ |

Table 22: Misclassification rate (↑) on Cora with budget $\Delta = 1$ to 5: GOttack achieves performance in 2 out of 5 tasks (GCN-Jaccard model).

| Budget → | 1 | 2 | 3 | 4 | 5 |
|---|---|---|---|---|---|
| FGA | $0.24 \pm 0.011$ | $0.41 \pm 0.047$ | $0.47 \pm 0.065$ | $0.55 \pm 0.020$ | $0.56 \pm 0.062$ |
| SGA | $0.33 \pm 0.025$ | $\mathbf{0.50 \pm 0.038}$ | $0.54 \pm 0.050$ | $0.59 \pm 0.013$ | $\mathbf{0.65 \pm 0.025}$ |
| Nettack | $\underline{0.38 \pm 0.011}$ | $\underline{0.50 \pm 0.073}$ | $\underline{0.54 \pm 0.023}$ | $\mathbf{0.62 \pm 0.000}$ | $\underline{0.61 \pm 0.112}$ |
| GOttack | $\mathbf{0.39 \pm 0.011}$ | $0.46 \pm 0.062$ | $\mathbf{0.61 \pm 0.031}$ | $\underline{0.62 \pm 0.023}$ | $0.57 \pm 0.054$ |

Table 23: Misclassification rate (↑) on Citeseer with budget $\Delta = 1$ to 5: GOttack achieves highest performance in 1 out of 5 tasks (GCN-Jaccard model).

| Budget → | 1 | 2 | 3 | 4 | 5 |
|---|---|---|---|---|---|
| FGA | $0.35 \pm 0.035$ | $0.42 \pm 0.073$ | $0.46 \pm 0.031$ | $0.52 \pm 0.065$ | $0.57 \pm 0.112$ |
| SGA | $0.36 \pm 0.013$ | $\mathbf{0.60 \pm 0.050}$ | $\mathbf{0.63 \pm 0.025}$ | $\mathbf{0.68 \pm 0.000}$ | $0.70 \pm 0.025$ |
| Nettack | $\mathbf{0.42 \pm 0.020}$ | $0.52 \pm 0.035$ | $\underline{0.59 \pm 0.047}$ | $0.60 \pm 0.040$ | $\underline{0.72 \pm 0.040}$ |
| GOttack | $\underline{0.42 \pm 0.092}$ | $0.52 \pm 0.020$ | $0.57 \pm 0.020$ | $\underline{0.62 \pm 0.011}$ | $\mathbf{0.72 \pm 0.020}$ |

Table 24: Misclassification rate (↑) on Polblogs with budget $\Delta = 1$ to 5: GOttack achieves performance in 1 out of 5 tasks (GCN-Jaccard model).

| Budget → | 1 | 2 | 3 | 4 | 5 |
|---|---|---|---|---|---|
| FGA | $0.50 \pm 0.077$ | $0.42 \pm 0.061$ | $\mathbf{0.56 \pm 0.042}$ | $\mathbf{0.59 \pm 0.031}$ | $\underline{0.55 \pm 0.054}$ |
| SGA | $0.43 \pm 0.050$ | $\underline{0.43 \pm 0.000}$ | $\underline{0.56 \pm 0.088}$ | $0.43 \pm 0.025$ | $\mathbf{0.59 \pm 0.088}$ |
| Nettack | $\underline{0.46 \pm 0.031}$ | $\mathbf{0.47 \pm 0.073}$ | $0.44 \pm 0.031$ | $\underline{0.47 \pm 0.02}$ | $0.52 \pm 0.071$ |
| GOttack | $\mathbf{0.53 \pm 0.065}$ | $0.43 \pm 0.065$ | $0.49 \pm 0.071$ | $0.46 \pm 0.058$ | $0.54 \pm 0.077$ |

Table 25: Misclassification rate (↑) on BlogCatalog with budget $\Delta = 1$ to 5: GOttack achieves performance in 0 out of 5 tasks (GCN-Jaccard model).

| Budget → | 1 | 2 | 3 | 4 | 5 |
|---|---|---|---|---|---|
| FGA | $0.12 \pm 0.050$ | $0.19 \pm 0.011$ | $0.25 \pm 0.000$ | $0.325 \pm 0.035$ | $0.34 \pm 0.042$ |
| SGA | $\mathbf{0.25 \pm 0.025}$ | $0.28 \pm 0.000$ | $0.33 \pm 0.000$ | $0.35 \pm 0.050$ | $0.40 \pm 0.025$ |
| Nettack | $\underline{0.19 \pm 0.030}$ | $\mathbf{0.30 \pm 0.031}$ | $\mathbf{0.41 \pm 0.058}$ | $\mathbf{0.46 \pm 0.071}$ | $\mathbf{0.47 \pm 0.04}$ |
| GOttack | $0.20 \pm 0.020$ | $\underline{0.30 \pm 0.054}$ | $\underline{0.36 \pm 0.071}$ | $\underline{0.43 \pm 0.077}$ | $\underline{0.42 \pm 0.042}$ |

## D.2 DEFENSE GNN BACKBONE

To demonstrate the effectiveness of the proposed attack model, we compare the performance of GOttack with various adversarial attack techniques across different defense models, namely GCN-Jaccard, MedianGCN, RobustGCN, and GCN-SVD. Based on the analysis of the results, as shown in Tables 22 to 37, we can conclude that the proposed GOttack is more robust against MedianGCN, achieving the highest performance in 11 out of 20 tasks, whereas it attained the least performance in 3 out of 20 tasks.

### D.2.1 ATTACK RESULTS ON GCN-JACCARD

In this subsection, we discuss the performance of the proposed GOttack along with various adversarial attack techniques on GCN-Jaccard, as shown in Tables 22 to 25. The proposed GOttack method achieved the second highest performance in 4 out of 20 tasks, while Nettack and SGA attained the highest performance in 7 tasks.

Table 26: Misclassification rate (↑) on Cora with budget $\Delta = 1$ to 5: GOttack achieves performance in 1 out of 5 tasks (MedianGCN model).

| Budget → | 1 | 2 | 3 | 4 | 5 |
|---|---|---|---|---|---|
| FGA | $0.25 \pm 0.060$ | $0.32 \pm 0.040$ | $0.49 \pm 0.033$ | $0.54 \pm 0.065$ | $0.55 \pm 0.036$ |
| SGA | $\underline{0.32 \pm 0.050}$ | $\mathbf{0.52 \pm 0.075}$ | $\mathbf{0.60 \pm 0.057}$ | $\mathbf{0.65 \pm 0.06}$ | $\mathbf{0.75 \pm 0.031}$ |
| Nettack | $0.32 \pm 0.052$ | $\underline{0.45 \pm 0.044}$ | $\underline{0.59 \pm 0.058}$ | $0.61 \pm 0.033$ | $0.66 \pm 0.033$ |
| GOttack | $\mathbf{0.32 \pm 0.036}$ | $0.45 \pm 0.048$ | $0.58 \pm 0.053$ | $\underline{0.63 \pm 0.033}$ | $\underline{0.68 \pm 0.029}$ |

Table 27: Misclassification rate (↑) on Citeseer with budget $\Delta = 1$ to 5: GOttack achieves highest performance in 0 out of 5 tasks (MedianGCN model).

| Budget → | 1 | 2 | 3 | 4 | 5 |
|---|---|---|---|---|---|
| FGA | $0.18 \pm 0.018$ | $0.37 \pm 0.048$ | $0.46 \pm 0.037$ | $0.56 \pm 0.079$ | $0.62 \pm 0.052$ |
| SGA | $\mathbf{0.36 \pm 0.013}$ | $\mathbf{0.50 \pm 0.000}$ | $\mathbf{0.60 \pm 0.000}$ | $\mathbf{0.66 \pm 0.063}$ | $\mathbf{0.73 \pm 0.000}$ |
| Nettack | $0.27 \pm 0.050$ | $\underline{0.46 \pm 0.051}$ | $0.55 \pm 0.050$ | $0.65 \pm 0.024$ | $0.71 \pm 0.048$ |
| GOttack | $\underline{0.28 \pm 0.029}$ | $0.43 \pm 0.048$ | $\underline{0.57 \pm 0.048}$ | $\underline{0.65 \pm 0.022}$ | $\underline{0.72 \pm 0.053}$ |

Table 28: Misclassification rate (↑) on Polblogs with budget $\Delta = 1$ to 5: GOttack achieves performance in 1 out of 5 tasks (MedianGCN model).

| Budget → | 1 | 2 | 3 | 4 | 5 |
|---|---|---|---|---|---|
| FGA | $0.32 \pm 0.065$ | $\underline{0.46 \pm 0.117}$ | $0.49 \pm 0.044$ | $0.50 \pm 0.044$ | $\underline{0.53 \pm 0.029}$ |
| SGA | $\mathbf{0.43 \pm 0.050}$ | $\mathbf{0.50 \pm 0.075}$ | $\mathbf{0.55 \pm 0.025}$ | $\underline{0.51 \pm 0.038}$ | $\mathbf{0.60 \pm 0.025}$ |
| Nettack | $0.33 \pm 0.029$ | $0.40 \pm 0.060$ | $\underline{0.51 \pm 0.081}$ | $0.47 \pm 0.092$ | $0.47 \pm 0.082$ |
| GOttack | $\underline{0.34 \pm 0.079}$ | $0.37 \pm 0.057$ | $0.45 \pm 0.06$ | $\mathbf{0.53 \pm 0.078}$ | $0.51 \pm 0.060$ |

Table 29: Misclassification rate (↑) on BlogCatalog with budget $\Delta = 1$ to 5: GOttack achieves performance in 0 out of 5 tasks (MedianGCN model).

| Budget → | 1 | 2 | 3 | 4 | 5 |
|---|---|---|---|---|---|
| FGA | $\mathbf{0.25 \pm 0.025}$ | $0.19 \pm 0.062$ | $0.17 \pm 0.025$ | $0.25 \pm 0.00$ | $0.26 \pm 0.037$ |
| SGA | $\underline{0.20 \pm 0.018}$ | $\mathbf{0.25 \pm 0.048}$ | $\mathbf{0.32 \pm 0.085}$ | $\mathbf{0.32 \pm 0.082}$ | $\mathbf{0.35 \pm 0.062}$ |
| Nettack | $0.17 \pm 0.025$ | $\underline{0.24 \pm 0.037}$ | $0.26 \pm 0.037$ | $\underline{0.26 \pm 0.037}$ | $0.31 \pm 0.037$ |
| GOttack | $0.19 \pm 0.037$ | $0.21 \pm 0.037$ | $\underline{0.31 \pm 0.125}$ | $0.25 \pm 0.049$ | $\underline{0.34 \pm 0.037}$ |

Table 30: Misclassification rate (↑) on Cora with budget $\Delta = 1$ to 5: GOttack achieves performance in 1 out of 5 tasks (RobustGCN model).

| Budget → | 1 | 2 | 3 | 4 | 5 |
|---|---|---|---|---|---|
| FGA | $0.34 \pm 0.060$ | $0.46 \pm 0.068$ | $0.63 \pm 0.048$ | $0.62 \pm 0.048$ | $0.65 \pm 0.044$ |
| SGA | $\underline{0.44 \pm 0.013}$ | $0.55 \pm 0.025$ | $0.60 \pm 0.025$ | $0.69 \pm 0.088$ | $0.71 \pm 0.013$ |
| Nettack | $\mathbf{0.48 \pm 0.080}$ | $\mathbf{0.61 \pm 0.046}$ | $\mathbf{0.72 \pm 0.029}$ | $\mathbf{0.70 \pm 0.033}$ | $\mathbf{0.73 \pm 0.025}$ |
| GOttack | $0.43 \pm 0.048$ | $\underline{0.60 \pm 0.036}$ | $\underline{0.70 \pm 0.027}$ | $\underline{0.69 \pm 0.033}$ | $\mathbf{0.73 \pm 0.025}$ |

Table 31: Misclassification rate (↑) on Citeseer with budget $\Delta = 1$ to 5: GOttack achieves highest performance in 2 out of 5 tasks (RobustGCN model).

| Budget → | 1 | 2 | 3 | 4 | 5 |
|---|---|---|---|---|---|
| FGA | $0.35 \pm 0.113$ | $0.49 \pm 0.095$ | $0.60 \pm 0.029$ | $0.59 \pm 0.048$ | $0.62 \pm 0.057$ |
| SGA | $\mathbf{0.53 \pm 0.000}$ | $\underline{0.59 \pm 0.013}$ | $0.66 \pm 0.013$ | $0.71 \pm 0.013$ | $\underline{0.75 \pm 0.000}$ |
| Nettack | $0.46 \pm 0.040$ | $\mathbf{0.61 \pm 0.071}$ | $\mathbf{0.72 \pm 0.018}$ | $\underline{0.72 \pm 0.038}$ | $0.75 \pm 0.047$ |
| GOttack | $\underline{0.48 \pm 0.070}$ | $0.59 \pm 0.046$ | $\underline{0.72 \pm 0.043}$ | $\mathbf{0.73 \pm 0.020}$ | $\mathbf{0.76 \pm 0.046}$ |

Table 32: Misclassification rate (↑) on Polblogs with budget $\Delta = 1$ to 5: GOttack achieves performance in 0 out of 5 tasks (RobustGCN model).

| Budget → | 1 | 2 | 3 | 4 | 5 |
|---|---|---|---|---|---|
| FGA | $0.31 \pm 0.110$ | $0.32 \pm 0.062$ | $0.41 \pm 0.037$ | $0.50 \pm 0.027$ | $0.48 \pm 0.040$ |
| SGA | $\mathbf{0.46 \pm 0.038}$ | $\mathbf{0.43 \pm 0.025}$ | $\mathbf{0.54 \pm 0.038}$ | $\mathbf{0.61 \pm 0.013}$ | $\mathbf{0.64 \pm 0.063}$ |
| Nettack | $0.38 \pm 0.055$ | $\underline{0.40 \pm 0.048}$ | $0.45 \pm 0.027$ | $\underline{0.52 \pm 0.027}$ | $\underline{0.51 \pm 0.043}$ |
| GOttack | $\underline{0.40 \pm 0.022}$ | $0.40 \pm 0.062$ | $\underline{0.46 \pm 0.046}$ | $0.51 \pm 0.060$ | $0.50 \pm 0.039$ |

Table 33: Misclassification rate (↑) on BlogCatalog with budget $\Delta = 1$ to 5: GOttack achieves performance in 0 out of 5 tasks (RobustGCN model).

| Budget → | 1 | 2 | 3 | 4 | 5 |
|---|---|---|---|---|---|
| FGA | $0.32 \pm 0.027$ | $0.26 \pm 0.046$ | $0.26 \pm 0.033$ | $0.31 \pm 0.019$ | $0.31 \pm 0.088$ |
| SGA | $0.25 \pm 0.025$ | $\mathbf{0.33 \pm 0.025}$ | $\mathbf{0.36 \pm 0.013}$ | $\mathbf{0.36 \pm 0.013}$ | $\mathbf{0.45 \pm 0.050}$ |
| Nettack | $\mathbf{0.35 \pm 0.044}$ | $\underline{0.32 \pm 0.027}$ | $\underline{0.34 \pm 0.029}$ | $\mathbf{0.37 \pm 0.048}$ | $\underline{0.40 \pm 0.044}$ |
| GOttack | $\underline{0.35 \pm 0.050}$ | $0.31 \pm 0.048$ | $0.30 \pm 0.306$ | $0.35 \pm 0.059$ | $0.35 \pm 0.043$ |

### D.2.2 ATTACK RESULTS ON MEDIANGCN

We discuss the performance of the proposed GOttack along with various adversarial attack techniques on the MedianGCN defense model, as shown in Tables 26 to 29. The proposed SGA method achieved the highest performance in 17 out of 20 tasks, while GOttack achieved the second highest performance in 2 tasks.

### D.2.3 ATTACK RESULTS ON ROBUSTGCN

We discuss the performance of the proposed GOttack along with various adversarial attack techniques on RobustGCN, as shown in Tables 30 to 33. The proposed GOttack method achieved the second highest performance in 2 out of 20 tasks, while the competitor attack methods, SGA and Nettack, had the highest performance in 9 tasks.

### D.2.4 ATTACK RESULTS ON GCN-SVD

In this subsection, we discuss the performance of the proposed GOttack along with various adversarial attack techniques on GCN-SVD, as shown in Tables 34 to 37. The proposed GOttack method achieved the highest performance in 2 out of 20 tasks, whereas Nettack and SGA attained the highest performance in 10 and 8 tasks, respectively.

Table 34: Misclassification rate (↑) on Cora with budget $\Delta = 1$ to 5: GOttack achieves performance in 1 out of 5 tasks (GCN-SVD model).

| Budget → | 1 | 2 | 3 | 4 | 5 |
|---|---|---|---|---|---|
| FGA | 0.27 ± 0.010 | 0.23 ± 0.070 | 0.30 ± 0.070 | 0.30 ± 0.040 | 0.27 ± 0.048 |
| SGA | 0.28 ± 0.033 | **0.38 ± 0.038** | **0.39 ± 0.000** | **0.39 ± 0.050** | **0.40 ± 0.100** |
| Nettack | 0.24 ± 0.011 | 0.29 ± 0.050 | 0.31 ± 0.050 | 0.33 ± 0.059 | 0.32 ± 0.031 |
| GOttack | **0.28 ± 0.020** | 0.25 ± 0.060 | 0.30 ± 0.035 | 0.27 ± 0.050 | 0.26 ± 0.031 |

Table 35: Misclassification rate (↑) on Citeseer with budget $\Delta = 1$ to 5: GOttack achieves highest performance in 0 out of 5 tasks (GCN-SVD model).

| Budget → | 1 | 2 | 3 | 4 | 5 |
|---|---|---|---|---|---|
| FGA | 0.22 ± 0.054 | 0.27 ± 0.540 | 0.24 ± 0.031 | 0.22 ± 0.047 | 0.25 ± 0.035 |
| SGA | 0.24 ± 0.013 | 0.25 ± 0.025 | 0.26 ± 0.038 | 0.28 ± 0.100 | **0.38 ± 0.020** |
| Nettack | **0.28 ± 0.040** | **0.30 ± 0.020** | **0.35 ± 0.035** | **0.33 ± 0.023** | 0.33 ± 0.047 |
| GOttack | 0.25 ± 0.030 | 0.27 ± 0.040 | 0.27 ± 0.062 | 0.31 ± 0.065 | 0.37 ± 0.061 |

Table 36: Misclassification rate (↑) on Polblogs with budget $\Delta = 1$ to 5: GOttack achieves performance in 0 out of 5 tasks (GCN-SVD model).

| Budget → | 1 | 2 | 3 | 4 | 5 |
|---|---|---|---|---|---|
| FGA | 0.10 ± 0.010 | 0.16 ± 0.311 | 0.16 ± 0.031 | 0.20 ± 0.054 | 0.25 ± 0.065 |
| SGA | **0.16 ± 0.013** | 0.16 ± 0.038 | 0.15 ± 0.025 | 0.15 ± 0.025 | 0.20 ± 0.025 |
| Nettack | 0.12 ± 0.040 | **0.18 ± 0.031** | **0.22 ± 0.020** | **0.24 ± 0.042** | **0.30 ± 0.093** |
| GOttack | 0.10 ± 0.020 | 0.12 ± 0.020 | 0.12 ± 0.031 | 0.16 ± 0.011 | 0.16 ± 0.011 |

Table 37: Misclassification rate (↑) on BlogCatalog with budget $\Delta = 1$ to 5: GOttack achieves performance in 1 out of 5 tasks (GCN-SVD model).

| Budget → | 1 | 2 | 3 | 4 | 5 |
|---|---|---|---|---|---|
| FGA | 0.17 ± 0.031 | 0.18 ± 0.062 | 0.23 ± 0.031 | 0.19 ± 0.011 | 0.30 ± 0.054 |
| SGA | 0.15 ± 0.050 | **0.24 ± 0.013** | 0.25 ± 0.025 | **0.31 ± 0.063** | 0.35 ± 0.000 |
| Nettack | 0.19 ± 0.023 | 0.23 ± 0.051 | **0.28 ± 0.023** | 0.20 ± 0.023 | **0.37 ± 0.023** |
| GOttack | **0.20 ± 0.002** | 0.22 ± 0.020 | 0.28 ± 0.062 | 0.20 ± 0.024 | 0.35 ± 0.020 |

Table 38: Misclassification rate (↑) on Cora with budget $\Delta = 1$ to 5: GOttack achieves performance in 1 out of 5 tasks (GNNGuard model).

| Budget → | 1 | 2 | 3 | 4 | 5 |
|---|---|---|---|---|---|
| SGA | **0.125 ± 0.024** | **0.200 ± 0.043** | **0.233 ± 0.057** | 0.258 ± 0.062 | **0.316 ± 0.057** |
| Nettack | 0.070 ± 0.050 | 0.138 ± 0.054 | 0.184 ± 0.066 | 0.250 ± 0.065 | 0.278 ± 0.069 |
| PRBCD | 0.074 ± 0.041 | 0.136 ± 0.036 | 0.169 ± 0.046 | 0.200 ± 0.049 | 0.229 ± 0.057 |
| GOttack | 0.116 ± 0.028 | 0.175 ± 0.066 | 0.216 ± 0.057 | **0.258 ± 0.012** | 0.260 ± 0.025 |

Table 39: Misclassification rate (↑) on Cora with budget $\Delta = 1$ to 5: GOttack achieves performance in 4 out of 5 tasks (GCORN model).

| Budget → | 1 | 2 | 3 | 4 | 5 |
|---|---|---|---|---|---|
| SGA | 0.300 ± 0.035 | 0.337 ± 0.017 | 0.400 ± 0.070 | 0.425 ± 0.000 | 0.462 ± 0.053 |
| Nettack | 0.312 ± 0.053 | 0.387 ± 0.088 | 0.475 ± 0.076 | **0.600 ± 0.000** | 0.587 ± 0.017 |
| GOttack | **0.350 ± 0.000** | **0.462 ± 0.017** | **0.537 ± 0.053** | 0.600 ± 0.070 | 0.587 ± 0.017 |

Table 40: Misclassification rate (↑) on Citeseer with budget $\Delta = 1$ to 5: GOttack achieves performance in 3 out of 5 tasks (GCORN model).

| Budget → | 1 | 2 | 3 | 4 | 5 |
|---|---|---|---|---|---|
| SGA | 0.266 ± 0.014 | 0.291 ± 0.014 | 0.366 ± 0.028 | 0.408 ± 0.094 | 0.458 ± 0.057 |
| Nettack | **0.391 ± 0.076** | 0.508 ± 0.052 | **0.608 ± 0.112** | 0.658 ± 0.080 | 0.691 ± 0.052 |
| PRBCD | 0.350 ± 0.024 | 0.508 ± 0.028 | 0.575 ± 0.090 | 0.600 ± 0.086 | 0.583 ± 0.057 |
| GOttack | 0.325 ± 0.017 | **0.550 ± 0.053** | 0.575 ± 0.088 | **0.675 ± 0.034** | **0.696 ± 0.088** |

Table 41: Misclassification rate (↑) on Cora with budget $\Delta = 1$ to 5: GOttack achieves performance in 0 out of 5 tasks (GARNET model).

| Budget → | 1 | 2 | 3 | 4 | 5 |
|---|---|---|---|---|---|
| SGA | 0.183 ± 0.011 | 0.133 ± 0.031 | 0.199 ± 0.020 | 0.250 ± 0.040 | 0.272 ± 0.050 |
| Nettack | **0.250 ± 0.035** | **0.250 ± 0.035** | **0.333 ± 0.042** | **0.541 ± 0.096** | **0.643 ± 0.045** |
| GOttack | 0.185 ± 0.023 | 0.225 ± 0.020 | 0.291 ± 0.062 | 0.383 ± 0.104 | 0.501 ± 0.146 |

#### D.2.5 ATTACK RESULTS ON GNNGUARD

In this subsection, we discuss the performance of the proposed GOttack along with various adversarial attack techniques on GNNGuard, as shown in Table 38. The proposed GOttack method achieved the highest performance in 1 out of 5 tasks, while SGA achieved the highest performance in 4 tasks.

#### D.2.6 ATTACK RESULTS ON GCORN

In this subsection, we discuss the performance of the proposed GOttack along with various adversarial attack techniques on GCORN, as shown in Tables 39 and 40. The proposed GOttack method achieved the highest performance in 7 out of 10 tasks, while Nettack achieved the highest performance in 4 tasks.

#### D.2.7 ATTACK RESULTS ON GARNET

In this subsection, we discuss the performance of the proposed GOttack along with various adversarial attack techniques on GCORN, as shown in Table 41. The proposed GOttack method secured the second-highest performance across all 5 tasks, while Nettack achieved the highest performance.

## E ADDITIONAL PROOF OF GOTTACK'S IMPACT

**Node's features.** Another interesting property of the proposed attacks can be seen in Table 45 and Table 53, in which we observe the change in the target node characteristics after adding or removing

Table 42: Misclassification rate ($\uparrow$) on Cora (GCN backbone) with budget $\epsilon$.

| $\epsilon \rightarrow$ | 0.10 | 0.30 | 0.50 |
|---|---|---|---|
| SGA | $0.087 \pm 0.01$ | $0.250 \pm 0.07$ | $0.325 \pm 0.10$ |
| Nettack | $\underline{0.125 \pm 0.00}$ | $\underline{0.237 \pm 0.05}$ | $\mathbf{0.412 \pm 0.08}$ |
| PRBCD | $\mathbf{0.137 \pm 0.01}$ | $\mathbf{0.262 \pm 0.08}$ | $0.337 \pm 0.12$ |
| GOttack | $0.112 \pm 0.05$ | $0.225 \pm 0.03$ | $\underline{0.387 \pm 0.08}$ |

Table 43: Misclassification rate ($\uparrow$) on Citeseer (GCN backbone) with budget $\epsilon$.

| $\epsilon \rightarrow$ | 0.10 | 0.30 | 0.50 |
|---|---|---|---|
| SGA | $\underline{0.058 \pm 0.02}$ | $0.200 \pm 0.03$ | $0.450 \pm 0.04$ |
| Nettack | $0.037 \pm 0.01$ | $\underline{0.187 \pm 0.05}$ | $\underline{0.625 \pm 0.17}$ |
| PRBCD | $0.033 \pm 0.01$ | $\underline{0.216 \pm 0.04}$ | $0.500 \pm 0.14$ |
| GOttack | $\mathbf{0.062 \pm 0.01}$ | $\mathbf{0.237 \pm 0.12}$ | $\mathbf{0.652 \pm 0.37}$ |

Table 44: Misclassification rate ($\uparrow$) on Pubmed (GCN backbone) with budget $\epsilon$.

| $\epsilon \rightarrow$ | 0.10 | 0.30 | 0.50 |
|---|---|---|---|
| SGA | $\underline{0.262 \pm 0.01}$ | $0.337 \pm 0.01$ | $0.475 \pm 0.01$ |
| Nettack | $0.262 \pm 0.03$ | $\underline{0.525 \pm 0.02}$ | $\mathbf{0.726 \pm 0.01}$ |
| PRBCD | $0.250 \pm 0.00$ | $0.425 \pm 0.07$ | $0.537 \pm 0.05$ |
| GOttack | $\mathbf{0.275 \pm 0.03}$ | $\mathbf{0.537 \pm 0.01}$ | $\underline{0.700 \pm 0.10}$ |

an edge between different orbit types. More precisely, we consider degree centrality, closeness centrality, betweenness centrality, and clustering coefficient as node feature metrics. The results show that adding/removing an edge between the target node and the connecting node with frequent orbit type causes a significant shift in node feature, especially the betweenness centrality of the target node.

Table 45: Changes in the node attribute of the target node, averaged over all target nodes (CORA dataset). A positive value indicates an increase after the attack compared to before.

| Attack Orbit Type | Degree Centrality | Closeness Centrality | Betweenness Centrality | Clustering Coefficient |
|---|---|---|---|---|
| GOttack (1518) | +15.18 | +6.50 | +20.32 | -3.46 |
| 1922 | +15.19 | +17.01 | +25.54 | +1.35 |
| 1519 | +15.21 | +4.58 | +23.60 | -3.43 |
| 1819 | +15.22 | +9.34 | +23.61 | -3.42 |

**GOttack with PRBCD.** We integrated GOttack with PRBCD and GO-PRBCD and evaluated it on Cora and Citeseer. The results (ref. Table 46 and Table 47 ) provided that GO-PRBCD performs the best for budget 3 but not as well as PRBCD or GOttack. We attribute this to the block sampling strategy that scales up PRBCD. In contrast, GOtack presents a non-sampling-based, principled approach to identifying the best candidate nodes.

**Shortest path scores.** In this experiment, we aim to observe the change in shortest path scores before and after attacks, calculating the minimum traveling cost from the source node to the destination node. Table 48 presents the results for different orbit types and node labels in the Cora dataset. Here, the notation **LS** denotes the score when connecting edges between nodes with the same label, while **LD** denotes the score when connecting edges between nodes with different labels. It is worth noting that connecting edges between different labels results in higher or equal changes in shortest path scores compared to connections between nodes with the same label. The results also indicate that connecting nodes with different or the same labels belonging to the 1518 orbit types results in higher changes compared to other orbit types.

**Nodes' label in two hops neighbors changes.**
As we know, GNN classification depends not only on the node itself but also on its neighbors. Given two nodes in the network, if they belong to the same class, their embeddings will exhibit high similarity. Considering this, we investigate the number of nodes that share the same label (LS) as the target node ($v$) and the number of nodes with different labels (LD) before and after applying the attack.

Table 49: Edge betweenness of newly added edges between $1518$ nodes and target nodes.

| Newly added edge to | Edge betweeness |
|---|---|
| Misclassified nodes | $\mathbf{0.44 \pm 0.18}$ |
| Correctly classified nodes | $0.22 \pm 0.14$ |

Table 46: Misclassification rate (↑) on Cora: GOttack achieves performance in 0 out of 5 tasks (GCN model).

Table 47: Misclassification rate (↑) on Citeseer: GOttack achieves performance in 5 out of 5 tasks (GCN model).

| Budget → | 1 | 2 | 3 | 4 | 5 |
|---|---|---|---|---|---|
| PRBCD | $0.41 \pm 0.060$ | $\mathbf{0.61 \pm 0.072}$ | $\underline{0.64 \pm 0.038}$ | $\mathbf{0.73 \pm 0.014}$ | $\mathbf{0.76 \pm 0.025}$ |
| GO-PRBCD | $\mathbf{0.44 \pm 0.017}$ | $0.52 \pm 0.035$ | $\mathbf{0.67 \pm 0.010}$ | $0.70 \pm 0.035$ | $0.70 \pm 0.000$ |
| GOttack | $\underline{0.41 \pm 0.052}$ | $\underline{0.54 \pm 0.049}$ | $0.62 \pm 0.033$ | $0.66 \pm 0.042$ | $\underline{0.71 \pm 0.052}$ |

| Budget → | 1 | 2 | 3 | 4 | 5 |
|---|---|---|---|---|---|
| PRBCD | $\underline{0.46 \pm 0.044}$ | $0.57 \pm 0.014$ | $\underline{0.68 \pm 0.038}$ | $0.70 \pm 0.000$ | $\underline{0.73 \pm 0.014}$ |
| GO-PRBCD | $0.38 \pm 0.035$ | $\underline{0.58 \pm 0.100}$ | $0.59 \pm 0.017$ | $0.65 \pm 0.035$ | $0.70 \pm 0.350$ |
| GOttack | $\mathbf{0.46 \pm 0.034}$ | $\mathbf{0.63 \pm 0.037}$ | $\mathbf{0.72 \pm 0.054}$ | $\mathbf{0.76 \pm 0.063}$ | $\mathbf{0.78 \pm 0.042}$ |

Table 48: Shortest path length changes of the target node to the candidate node, with seven rows each representing a different label node group. These are mean values calculated over target test nodes. LS denotes the same label with the target node, and LD denotes a different label with the target node. A negative value indicates a decrease after the attack compared to before.

| | 1518 | | 1922 | | 1519 | | 1819 | |
|---|---|---|---|---|---|---|---|---|
| | LS | LD | LS | LD | LS | LD | LS | LD |
| Label-0 | -0.06 | -0.08 | -0.04 | -0.01 | -0.00 | -0.00 | -0.01 | -0.00 |
| Label-1 | -0.01 | -0.01 | -0.07 | -0.03 | -0.01 | -0.01 | -0.00 | -0.00 |
| Label-2 | -0.02 | -0.02 | -0.03 | -0.02 | -0.05 | -0.03 | -0.03 | -0.02 |
| Label-3 | -0.05 | -0.08 | -0.05 | -0.04 | -0.06 | -0.04 | -0.07 | -0.04 |
| Label-4 | -0.00 | -0.01 | -0.01 | -0.01 | -0.04 | -0.02 | -0.00 | -0.01 |
| Label-5 | -0.00 | 0.0 | -0.02 | -0.00 | -0.05 | -0.03 | -0.02 | -0.01 |
| Label-6 | -0.03 | -0.02 | -0.02 | -0.01 | -0.03 | -0.01 | -0.01 | -0.01 |

It is noteworthy that when we apply an attack on two convolution layers of GNN variants, the embeddings of nodes in the two-hop neighborhood significantly contribute to updating the embedding of the target node. Table 50 shows the changes in the two-hop neighbors of target nodes for the Cora dataset. From the results, we can observe that the target node is connected to LD nodes, which belong to 1518 orbit types. The neighbor change is significant, with an average increase of $+3.36$ for the GOttack method. Furthermore, the results of the same experiment with the same settings conducted on the Citeseer dataset are shown in the Appendix, Table 54.

**Important edge.** After adding a new edge to the target node, the embedding of the target nodes is updated with information aggregated through the new edge. The more important the edge in the two-hop neighbor of the target node, the more noise is added to the target node that eventually causes misclassification.

The importance of edge can be assessed based on edge betweenness centrality. We conducted 1518 GOttack on GCN, using the Cora dataset once to get the list of the target node and corresponding connecting nodes chosen by our algorithm. We separate them into two classes, one is the list of target nodes that are miss-classified and the other is the list of correct-classified. Table 49 shows that the average edge betweenness of edges added to miss-classified nodes is $0.44$, which is twice compared to the average edge betweenness of edges added to correct-classified nodes, $0.22$. It proves that our attack harms node classification accuracy by discovering and adding important edges, which connect the target node to a faraway region in the graph and bring the noise to the target nodes' embedding.

### E.1 SUBGRAPH-BASED EXPLANATIONS

GNNs integrate node feature information and graph structure by recursively passing messages along the edges of the graph to update the embedding and make predictions; therefore, this complex integration leads to models that are challenging to explain in terms of their predictions.

In this subsection, we leverage two different renowned explainers (subgraph-based) to see the changes that the GOttack method poses to graphs that lead to misclassification.

**GNNExplainer**. Battaglia et al. (2018), Zhou et al. (2020), and Zhang et al. (2022) summarize the update of the GNN model in three core computations. (i) For every pair of node $(v_i, v_j)$, the message is computed from the representations $\mathbf{h_i^{l-1}}$, $\mathbf{h_j^{l-1}}$ of each node in the previous layer and their relation $r_{ij}$, given by $m_{ij}^l = \mathbf{MSG}(\mathbf{h_i^{l-1}}, \mathbf{h_j^{l-1}}, r_{ij})$. (ii) Secondly, an aggregated message $\mathbf{M}_i$ is computed for each node $v_i$ by aggregating messages from all $v_i$'s neighborhood. (iii) Finally, the representation,

Table 50: Changes in the nodes' labels within two-hop neighbors (CORA dataset) of the target node, with seven rows each representing a different label node group. LS denotes the same label with the target node, and LD denotes a different label with the target node. These are mean values calculated over target test nodes. A positive value indicates an increase after the attack compared to before.

| | 1518 | | 1922 | | 1519 | | 1819 | |
|---|---|---|---|---|---|---|---|---|
| | LS | LD | LS | LD | LS | LD | LS | LD |
| Label-0 | +1.25 | +3.70 | +0.63 | +4.75 | +1.28 | +3.88 | +1.60 | +3.50 |
| Label-1 | +1.27 | +3.58 | +0.50 | +4.00 | +0.58 | +3.73 | +0.48 | +4.05 |
| Label-2 | +1.30 | +2.65 | +2.36 | +3.53 | +0.78 | +3.15 | +0.55 | +3.75 |
| Label-3 | +1.00 | +3.50 | +0.43 | +4.53 | +1.28 | +3.58 | +1.08 | +3.88 |
| Label-4 | +0.52 | +3.20 | +0.68 | +3.80 | +0.50 | +3.50 | +0.65 | +3.60 |
| Label-5 | +0.12 | +3.55 | 0.28 | +4.10 | +0.33 | +3.75 | +0.45 | +3.95 |
| Label-6 | +0.97 | +3.40 | +1.88 | +3.80 | +1.63 | +3.43 | +1.48 | +3.83 |

Table 51: Time Experiment Results in Seconds ($\downarrow$).

| | GOttack | Nettack | FGA | SGAttack | PRBCD | Random |
|---|---|---|---|---|---|---|
| Cora | $48.27 \pm 7.55$ | $59.57 \pm 11.31$ | $55.63 \pm 16.96$ | $\mathbf{32.24 \pm 5.27}$ | $242.37 \pm 6.03$ | $0.030 \pm 0.000$ |
| Citeseer | $42.02 \pm 5.14$ | $46.12 \pm 15.96$ | $42.43 \pm 2.31$ | $41.02 \pm 4.45$ | $210.55 \pm 7.58$ | $0.034 \pm 0.001$ |
| Polblogs | $\mathbf{139.68 \pm 62.21}$ | $175.23 \pm 56.14$ | $165.37 \pm 17.12$ | $164.07 \pm 13.77$ | $210.87 \pm 5.23$ | $0.020 \pm 0.000$ |
| BlogCatalog | $512.04 \pm 9.56$ | $916.91 \pm 129.17$ | $\mathbf{223.02 \pm 15.53}$ | $335.91 \pm 4.66$ | $259.38 \pm 3.58$ | $0.038 \pm 0.014$ |
| Pubmed | $246.96 \pm 62.57$ | $243.78 \pm 2.14$ | $533.55 \pm 108.19$ | $\mathbf{68.23 \pm 7.58}$ | $240.23 \pm 4.65$ | $0.040 \pm 0.017$ |

also known as embedding, $\mathbf{h_i^l}$ of node $v_i$ at layer $l$ is calculated from the embedding of the node $v_i$ in the previous layer and aggregated message $\mathbf{M}_i$. This demonstrates that the neighbor of all node $v_i$ contributes to formulating the final embedding of $v_i$. Ying et al. defined the subgraph of all neighbors of the node $v_i$ as a computation graph, denoted as $\mathcal{G}_c$ (Ying et al., 2019).

In particular, $v$'s computation graph tells the GNN how to generate $v$'s embedding $z$. The computation graph of node $v$ is crucial, as it fully determines all the information GNN uses to generate prediction $\hat{y}_v$ at node $v$. Among nodes and edges in $\mathcal{G}_c$, we are interested in $\mathcal{G}_s \subseteq \mathcal{G}_c$ that are important for the GNN's prediction on node $v$, in which removing of either a node or an edge in $\mathcal{G}_s$ strongly decrease the probability of prediction $\hat{y}_v$. By solving the optimization of the conditional entropy $H(Y|\mathcal{G} = \mathcal{G}_s \mathcal{X} = \mathcal{X}_s)$, GNNExplainer returns an explanation for the prediction $\hat{y}_v$ as $(\mathcal{G}_s, \mathcal{X}_s^F)$, where $\mathcal{G}_s$ is a small subgraph of the computation graph, $\mathcal{X}_s^F$ is the associated feature of $\mathcal{G}_s$, and $\mathcal{X}_s^F$ is a small subset of node features that are most important for explaining $\hat{y}_v$ (Ying et al., 2019).

Figure 5 is the subgraph of the computation graph most influential for the GNN's prediction on node 1978, denoted as $\hat{y}_{1978}$. Before the attack, the prediction $\hat{y}_{1978}$ made by GNN on node 1978 is strongly affected by edges connecting to $1306, 1241$ and 2381 having the same label and 6 edges connecting to different label nodes. After the attack by adding an edge between the target node and 1518 node 387, there are two out of three edges connecting the node having the same label to 1978 is no longer having a strong impact on $\hat{y}_{1978}$, while there is an increase in the number of edges connecting different label nodes to 9. As a result, GNN is fooled into making a misprediction on 1978 after applying GOttack. In addition, we leverage GNNExplainer to explain another successful attack of GOttack on

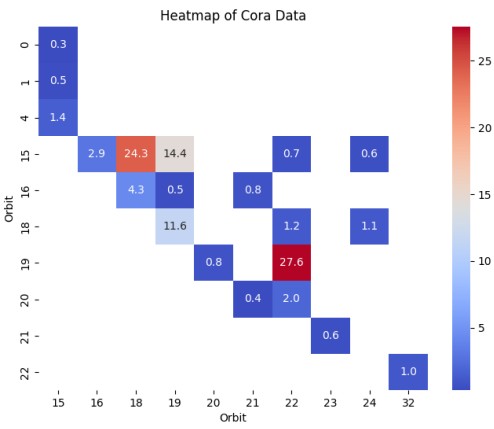

Figure 8: Matrix of co-occurrence percentages for node orbits: visualizing intersections between the first and second orbit.

Table 52: Computation time for orbit discovery across different datasets.

| Dataset | Orbit Discovery Time (in Seconds) |
|---|---|
| Cora | 0.17 |
| Citeseer | 0.10 |
| Polblogs | 18.34 |
| Blogcatalog | 470.60 |
| Pubmed | 2.48 |

Table 53: Changes in the node attribute of the target node, averaged over all target nodes (POLBLOGS dataset). A positive value indicates an increase after the attack compared to before.

| Orbit type | LSame-2 hop | LDiff-2hop | Degree Centrality($10^{-3}$) | Closeness Centrality($10^{-2}$) | Betweenness Centrality($10^{-3}$) | Cluster |
|---|---|---|---|---|---|---|
| GOttack (1518) | $8.30 \pm 1.06$ | $1.58 \pm 0.30$ | $0.82 \pm 0.00$ | $0.35 \pm 0.14$ | $0.11 \pm 0.02$ | $-0.04 \pm 0.03$ |
| 1922 | $8.89 \pm 1.14$ | $1.51 \pm 0.20$ | $0.82 \pm 0.00$ | $0.43 \pm 0.17$ | $0.12 \pm 0.03$ | $-0.04 \pm 0.04$ |
| 1519 | $9.25 \pm 0.83$ | $0.97 \pm 0.46$ | $0.82 \pm 0.00$ | $0.43 \pm 0.17$ | $0.12 \pm 0.03$ | $-0.04 \pm 0.04$ |
| 1819 | $9.64 \pm 1.18$ | $4.29 \pm 0.63$ | $0.95 \pm 0.00$ | $0.47 \pm 0.20$ | $0.12 \pm 0.05$ | $-0.05 \pm 0.04$ |

node 1784, described by Figure 10.

Similarly, the newly added edge is considered an important edge, and there are significant changes in the importance of the remaining edge in the computational subgraph caused by the newly added edge $(1784, 709)$.

**PGExplainer**. PGExplainer (Luo et al., 2020) is a model-agnostic method designed to provide explanations for predictions made by GNNs. It is used to identify a subgraph and subset of node features crucial for a specific prediction. Let us consider a trained GNN model $f_\theta(\mathcal{G})$, where $\mathcal{G} = (\mathcal{V}, \mathcal{E}, \mathcal{X})$ is input graph. It aims to explain the prediction $\hat{y}_v$ for a target node $v \in \mathcal{V}$. It learns an edge mask $\mathbf{M}_e \in [0,1]^{|\mathcal{E}|}$ and a feature mask $\mathbf{M}_x \in [0,1]^{|\mathcal{X}|}$. The masks are optimized to maximize the mutual information between the GNN's predictions $\hat{y}_v$ and the subgraph $\mathcal{G}_s$, expressed as: $\mathbf{MI}(\hat{y}_v, \mathcal{G}_s) = H(\hat{y}_v) - H(\hat{y}_v|\mathcal{G} = \mathcal{G}_s)$.

However, optimizing this objective is infeasible due to the exponential number of possible subgraphs (i.e., $2^N$), the problem is relaxed by assuming $\mathcal{G}_s$ is a Gilbert random graph where edge selections are conditionally independent. The probability of an edge $e_{ij}$ being selected is modeled using a Bernoulli distribution: $P(\mathcal{G}) = \prod_{(i,j) \in \mathcal{E}} P(e_{ij})$, where $e_{ij} \in \mathcal{V} \times \mathcal{V}$ is a binary variable indicating whether the edge is selected, with $e_{ij} = 1$ if the edge $(i, j)$ is selected, and 0 otherwise. The probability $P(e_{ij} = 1) = \theta_{ij}$ is the probability that edge $(i, j)$ exists in $\mathcal{G}$. With this relaxation, the objective can be rewritten as (Luo et al., 2020): $\min_{\mathcal{G}_s} H(\hat{y}_v|\mathcal{G} = \mathcal{G}_s) = \min_{\mathcal{G}_s} \mathbb{E}_{\mathcal{G}_s}[H(\hat{y}_v|\mathcal{G} =$

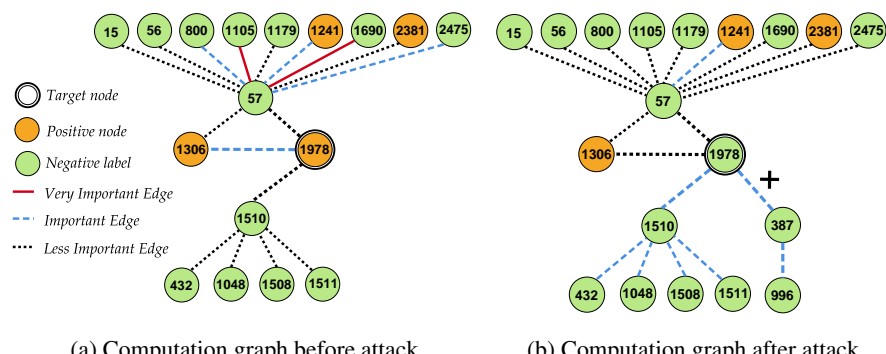

(a) Computation graph before attack  (b) Computation graph after attack

Figure 9: The computation graph for the targeted node 1978 from the CORA datasets, as identified by PGExplainer (Luo et al., 2020) . The edge $(1978, 387)$ is added during the successful attack. Edge importances change considerably after the attack, and the negative class gains importance due to the newly added nodes.

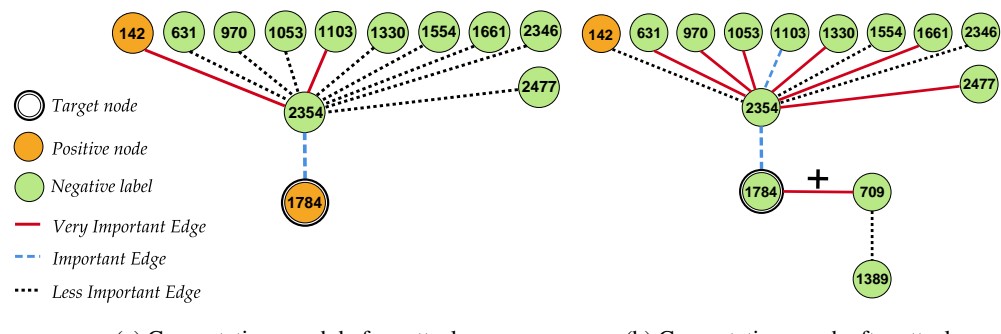

(a) Computation graph before attack          (b) Computation graph after attack

Figure 10: The computation graph for the targeted node 1784 from the CORA datasets, as identified by GNNExplainer (Luo et al., 2020). The edge (1784, 709) is added during the successful attack. Edge importances change considerably after the attack, and the negative class gains importance due to the newly added nodes.

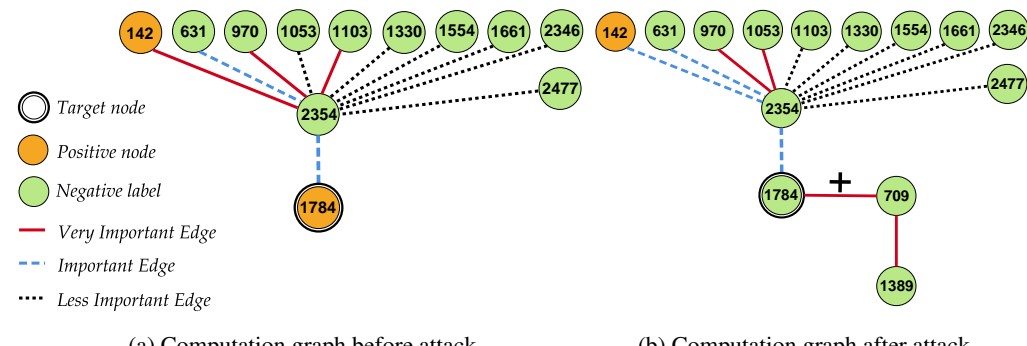

(a) Computation graph before attack          (b) Computation graph after attack

Figure 11: The computation graph for the targeted node 1784 from the CORA datasets, as identified by PGExplainer (Luo et al., 2020). The edge (1784, 709) is added during the successful attack. Edge importances change considerably after the attack, and the negative class gains importance due to the newly added nodes.

Table 54: Changes in the nodes' labels within two-hop neighbors (CITESEER dataset) of the target node, with six rows each representing a different label node group. LS denotes the same label with the target node, and LD denotes a different label with the target node. These are mean values calculated over target test nodes. A positive value indicates an increase after the attack compared to before.

| | 1518 | | 1922 | | 1519 | | 1819 | |
|---|---|---|---|---|---|---|---|---|
| | LS | LD | LS | LD | LS | LD | LS | LD |
| Label-0 | +0.45 | +3.78 | +0.95 | +4.50 | +0.15 | +3.43 | +0.05 | +4.15 |
| Label-1 | +0.58 | +3.58 | +2.50 | +3.35 | +0.33 | +3.30 | +1.53 | +3.08 |
| Label-2 | +2.00 | +3.03 | +1.48 | +4.18 | +2.80 | +2.28 | +0.63 | +3.80 |
| Label-3 | +0.15 | +3.73 | +0.38 | +4.18 | +0.48 | +3.23 | +0.65 | +3.50 |
| Label-4 | +2.55 | +2.95 | +1.70 | +3.30 | +1.13 | +3.28 | +1.78 | +2.88 |
| Label-5 | +1.28 | +3.12 | 0.48 | +4.18 | +1.65 | +3.38 | +0.65 | +3.93 |

Table 55: The descriptions of symbols.

| Symbol | Descriptions |
|---|---|
| $\mathcal{G}$ | Graph representation |
| $\mathcal{V}$ | Sets of vertices in $\mathcal{G}$ |
| $\mathcal{E}$ | Sets of edges in $\mathcal{G}$ |
| $|\mathcal{E}|$ | Number of edges |
| $|\mathcal{V}|$ | Number of nodes |
| $\mathcal{A}$ | Adjacency matrix of $\mathcal{G}$ |
| $\mathcal{X}$ | Node features |
| $f_\theta$ | Graph neural network model |
| $v$ | Target node |
| $\mathcal{V}_T$ | Set of target nodes |
| $\mathcal{N}(v)$ | Set of adjacency nodes of $v$ |
| $h(\cdot)$ | Graph homophily ratio |
| $\mathcal{G}'$ | Perturbed graph |
| $\mathcal{G}_c$ | Computation graph |
| $\mathcal{G}_s$ | Subgraph of computation graph |
| $\mathcal{A}'$ | Perturbed adjacency matrix |
| $\hat{y}_v$ | Predicted class |
| $\Delta$ | Attack budget |
| $\mathcal{G}_{gp}$ | Graphlet |
| $\mathrm{Aut}(\mathcal{G}_{gp})$ | Automorphisms of a graphlet |
| $GOV$ | Graphlets Orbit Vector |
| $Orb_{max}$ | Largest orbit count value |
| $Orb_{sec}$ | Second-largest count value |

$\mathcal{G}_s)] \approx \min_\Theta \mathbb{E}_{\mathcal{G}_s \sim q(\Theta)}[H(\hat{y}_v|G = \mathcal{G}_s)]$, where $q(\Theta)$ is the distribution of the explanatory graph parameterized by $\theta$'s.

Figure 9 shows the subgraph of the computation graph with the most influential edge connections. Before the attack, the prediction $\hat{y}_{1978}$ made by the GNN on node 1978 is strongly influenced by edges connecting to node 1306. However, after the attack, which involves adding an edge between the target node and node 387, the strong connection to node 1306 is no longer present. Instead, there is an increase in the number of edges connecting to different label nodes such as node 1510. Consequently, the GNN is misled into making an incorrect prediction for node 1978 after the application of GOttack. In addition, we leverage PGExplainer to explain another successful attack of GOttack on node 1784, described by Figure 11. Similarly, the newly added edge is considered an important edge, and there are significant changes in the importance of the remaining edge in the computational subgraph caused by the newly added edge $(1784, 709)$.

Table 56: Summary of the Literature review.

| Ref. | Article Name | Attack Type | Perturbation | Evasion/ Poisoning | Domain | Model | Baseline | Metrics | Dataset |
|---|---|---|---|---|---|---|---|---|---|
| Zügner et al. (2018) | Nettack | Target Attack | Structure Feature | Both | Node Classif. | GCN CLN DeepWalk | Random FGSM | Accuracy Classif. Margin | Cora-ML Citeseer Polblogs |
| Chen et al. (2018) | FGA | Target Attack | Structure | Both | Node Classif. | GCN Grarep DeepWalk Node2vec Line GraphGAN | Random DICE Nettack | Success Rate AML | Cora-ML Citeseer Polblogs |
| Zügner & Günnemann (2019) | Mettack | Global Attack | Structure Feature | Poisoning | Node Classif. | GCN CLN DeepWalk | DICE Nettack First-order attack | Accuracy Misclassif. Rate | Cora-ML Pubmed Citeseer Polblogs |
| Dai et al. (2018) | RL-S2V | Target Attack | Structure | Evasion | Node Classif. | GNNs | Rnd. sampling Genetic algs. | Accuracy | Citeseer Finance Pubmed Cora |
| Bojchevski & Günnemann (2019) | Node Embedding | Global Attack | Structure | Poisoning | Node Embedding | DW SVD DW SGNS Node2vec Spect. Embd Label Prop. GCN | Unknown | Accuracy Classif. Margin Loss | Cora Citeseer Polblogs |
| Xu et al. (2019a) | PGD, Min-max | Global Attack | Structure | Both | Node Classif. | GCN | DICE Greedy Meta-self | Misclassif. Rate | Cora Citeseer |
| Waniek et al. (2016) | DICE | Global Attack | Structure | Both | Node Classif. | GCN | DICE ROAM heuristic | Concealment | TerroristNet Facebook Twitter Google+ ScaleFree SmallWorld RandomGraph |
| Wu et al. (2019) | IG-Attack | Target Attack | Structure Feature | Both | Node Classif. | GCN | JSMA IG-JSMA Nettack FGSM | Classif. Margin Accuracy | Cora Citeseer Polblogs |
| Sun et al. (2020) | NIPA | Global Attack | Structure | Poisoning | Node Classif. | GCN | Random FGA Preferential attack | Accuracy Graph Statistics | Cora-ML Pubmed Citeseer |
| Li et al. (2023) | SGAttack | Target Attack | Structure | Poisoning | Node Classif. | GCN GAT SGC GraphSAGE ClusterGCN | Random DICE GradArgmaxNettack | Accuracy Classif. Margin | Citeseer Cora Pubmed Reddit |
| Ma et al. (2020) | GC-RWCS | Target Attack | Structure | Evasion | Node Classif. | GCN JKNetConcat JKNetMaxpool | Random Degree Pagerank Betweenness RWCS GC-RWCS | Accuracy Loss | Citeseer Cora Pubmed |
| Chang et al. (2022) | GF-Attack | Global Attack | Feature | Evasion | Vertex Classif. | GCN SGC Cheby DW LINE | Random Degree RL-S2V A_class GF-Attack | Accuracy Execution Time | Cora Citeseer Pubmed |
| Tao et al. (2021) | G-NIA | Target Attack | Structure Feature | Evasion | Node Classif. | GCN GAT APPNP | Radnom MostAttr. PrefEdge. NIPA AFGSM G-NIA | Misclassif. Rate | Reddit Citeseer ogbn-products |
| Sun et al. (2018) | OPT-Attack | Target Attack | Structure | Poisoning | Node Embedding | GAE DeepWalk Node2vec LINE | Degree sum Shortest path Random PageRank | AP Similarity Score | Cora Citeseer Facebook |
| Hussain et al. (2021) | STRUCtack | Global Attack | Structure | Poisoning | Node Classif. | GCN | Random DICE Mettack PGD MinMax | Accuracy | Cora Citeseer Pubmed Cora-ML Polblogs |
| Wu et al. (2023) | WT-AWP | Unknown | Feature | Poisoning Evasion | Node Classif. Graph Classif. | GCN GAT PPNP | DICE PGD Mettack | Accuracy Loss | Cora Citeseer Polblogs |
| Zou et al. (2021) | TDGIA | Unknown | Feature | Evasion | Node Classif. | GCN | FGSM AFGSMSPEIT | Accuracy | KDD-CUP ogbn-arxiv Reddit |
| Tao et al. (2023) | CANA | Unknown | Feature | Unknown | Unknown | COPOD PCA HBOS IForestAE | PGD TDGIA G-NIA | Accuracy Misclassif. Rate | ogbn-products redditogbn-arxiv |
| Zhao et al. (2024) | HGAttack | Target Attack | Unknown | Evasion | Node Embedding | GCN | FGA | Macro F1 Micro F1 | ACM DBLP IMDB |
| Jin et al. (2023) | PEH | Unknown | Structure | Unknown | Node Classif. Node Clustering | GCN GAT DGI RGCN | Nettack Mettack Random PGD | Accuracy Attention Score | Cora Citeseer Polblogs |
| **Proposed Approach** | **GOttack** | **Target Attack** | **Structure** | **Poisoning** | **Node Classif.** | **GCN GIN GSAGE RGCN GCN-Jaccard GCN-SVD MedianGCN GNNGuard GARNET GCORN** | **Random Nettack FGA SGA PRBCD** | **Misclassif. Rate** | **Cora Citeseer Polblogs Pubmed BlogCatalog** |