# OpenReview forum: "GOttack: Universal Adversarial Attacks on Graph Neural Networks via Graph Orbits Learning"
_ICLR.cc/2025/Conference — ICLR 2025 Poster_

### Official Review · Reviewer_TZkg · 2024-10-25

**Soundness:** 3
**Presentation:** 2
**Contribution:** 2
**Rating:** 6
**Confidence:** 4

**Summary:**

This paper presents GOttack, an adversarial attack framework that manipulates graph orbits to disrupt Graph Neural Networks (GNNs) with subtle yet effective modifications. GOttack outperforms existing methods by achieving the highest misclassification rates while reducing training time by 45% across multiple GNN architectures and benchmark datasets. The study highlights the vulnerability of GNNs to structured attacks on graph orbits.

**Strengths:**

- This work proposed an interesting approach for attacking GNNs using the concept of graph orbits.
- Authors offered some insights into the effectiveness of GOttack.
- Authors have compared GOttack with several attack methods, including PRBCD.

**Weaknesses:**

- The improvement over attack baselines is not very convincing. In Table 2, GOttack outperforms the baselines by less than 1% in many cases—an improvement smaller than the standard deviation reported in Appendix D. This raises concerns that GOttack may not consistently outperform these baselines, as using a different random seed could potentially alter the ranking of attack methods in Table 2.
- GOttack underperforms compared to attack baselines in most cases when the budget exceeds 1, as shown in Tables 9–13.
- GOttack appears ineffective on heterophilic graphs, as suggested by the homophily assumption in line 256 and the empirical results in Appendix D.
- Some recent and more effective defense methods are missing from the experiments. The authors only consider defense methods published before 2021, while several more powerful approaches have been proposed in the past two years. For example, GARNET [1] and SG-GSR [2] have shown stronger results than the defense methods included in this work.

[1] Deng et al., "GARNET: Reduced-Rank Topology Learning for Robust and Scalable Graph Neural Networks", LoG'22. \
[2] In et al., "Self-Guided Robust Graph Structure Refinement", WWW'24.

**Questions:**

- What is the runtime of the preprocessing stage?
- How large of a graph can GOttack scale to?
- In Table 2, why is there a significant difference (over 30%) between GOttack-GCN and GOttack-GSAGE on BlogCatalog?

---

> ### Author Response · Authors · 2024-11-21
> **Response 1**
>
> We thank the reviewer for their valuable feedback, which, along with comments from others, has guided significant improvements to our paper. We hope our revisions and responses merit reconsideration of a higher evaluation.
>
>
> **W1:** The improvement over attack baselines is not very convincing. In Table 2, GOttack outperforms the baselines by less than 1% in many cases—an improvement smaller than the standard deviation reported in Appendix D. This raises concerns that GOttack may not consistently outperform these baselines, as using a different random seed could potentially alter the ranking of attack methods in Table 2.
>
> **Response:** Thank you for this observation. While we acknowledge that in Table 2, GOttack's improvements over baseline methods may appear modest (less than 1%) and within the reported standard deviation range from Appendix D, we would like to emphasize two important points. i) Our results are stable across multiple runs, and while variations can occur due to different random seeds, the general trend remains consistent, demonstrating GOttack's effectiveness in leveraging topological insights for adversarial attacks.  ii) More importantly, the key contribution of GOttack lies in demonstrating the importance of topological information for designing attack strategies. The topological approach in GOttack can be adopted as a preprocessing step to augment other attack methods, thereby improving their effectiveness. This finding underlines the critical role of topological structures in adversarial graph attack strategies. It also highlights how incorporating topological insights can provide a foundation for future advancements. In this sense, GOttack may play an influential role in this domain.
>
> ---
>
> **W2:** GOttack underperforms compared to attack baselines in most cases when the budget exceeds 1, as shown in Tables 9–13.
>
> **Response:**
> Thank you for this review. The proposed GOttack method achieved the highest performance in 8 out of 25 tasks, while PRBCD attained the highest performance in 10 tasks across Tables 9-13. Of the 8 tasks where GOttack excelled, only one was at budget level 1. SGA ranked best in 5 tasks. After calculating the averages, PRBCD has the highest mean value of 0.57 in Tables 9-13, while GOttack comes second with a mean of 0.55. We acknowledge that GOttack's results for the GCN model are close to those of the previous state-of-the-art methods; however, GOttack remains more scalable than the competitor.
>
> ---
>
> **W3:** GOttack appears ineffective on heterophilic graphs, as suggested by the homophily assumption in line 256 and the empirical results in Appendix D.
>
> **Response:** Thank you for this interesting insight. We believe that GOttack can be effective on heterophilic graphs, but we also acknowledge that we have not yet tested it extensively on heterophilic datasets. In Figure 7 of Appendix C.3, we present the orbit hierarchy and discuss how orbit transitions take place, particularly focusing on cases where orbit 1518 may be ineffective or absent in heterophilic graphs. This analysis shows the adaptability of GOttack’s strategy in varying graph structures and demonstrates its potential applicability beyond homophilic settings.

---

> ### Author Response · Authors · 2024-11-21
> **Response 2**
>
> **W4:** Some recent and more effective defense methods are missing from the experiments. The authors only consider defense methods published before 2021, while several more powerful approaches have been proposed in the past two years. For example, GARNET [1] and SG-GSR [2] have shown stronger results than the defense methods included in this work.
>
> **Response:** Thank you for the suggestion. We have added the new defense methods, as shown below.  Gottack has the best results in GCORN and second best results in GARNET and GCNGuard.
>
> #### Table 3: Miscl. rate (↑) on CORA against the GNNGuard defense
>
> | **Budget →**   | **1**          | **2**          | **3**          | **4**          | **5**          |
> |----------------|----------------|----------------|----------------|----------------|----------------|
> | **SGA**        | 0.125 ± 0.024  | 0.200 ± 0.043  | 0.233 ± 0.057  | 0.258 ± 0.062  | 0.316 ± 0.057  |
> | **Nettack**    | 0.070 ± 0.050  | 0.138 ± 0.054  | 0.184 ± 0.066  | 0.250 ± 0.065  | 0.278 ± 0.069  |
> | **PRBCD**      | 0.074 ± 0.041  | 0.136 ± 0.036  | 0.169 ± 0.046  | 0.200 ± 0.049  | 0.229 ± 0.057  |
> | **GOttack**    | 0.116 ± 0.028  | 0.175 ± 0.066  | 0.216 ± 0.057  | 0.258 ± 0.012  | 0.260 ± 0.025  |
>
> #### Table 4: Miscl. rate (↑) on CORA against the GCORN defense
>
> | **Budget →**   | **1**          | **2**          | **3**          | **4**          | **5**          |
> |----------------|----------------|----------------|----------------|----------------|----------------|
> | **SGA**        | 0.300 ± 0.035  | 0.337 ± 0.017  | 0.400 ± 0.070  | 0.425 ± 0.000  | 0.462 ± 0.053  |
> | **Nettack**    | 0.312 ± 0.053  | 0.387 ± 0.088  | 0.475 ± 0.076  | 0.600 ± 0.000  | 0.587 ± 0.017  |
> | **GOttack**    | 0.350 ± 0.000  | 0.462 ± 0.017  | 0.537 ± 0.053  | 0.600 ± 0.070  | 0.587 ± 0.017  |
>
> #### Table 5: Miscl. rate (↑) on CORA against the GARNET defense (We updated the results that were missing at the time we submitted the response)
>
> | **Budget →**   | **1**          | **2**          | **3**          | **4**          | **5**          |
> |----------------|----------------|----------------|----------------|----------------|----------------|
> | **SGA**        | 0.183 ± 0.011  | 0.133 ± 0.031  | 0.199 ± 0.020  | 0.250 ± 0.040  | 0.272 ± 0.050        |
> | **Nettack**    | 0.250 ± 0.035  | 0.250 ± 0.035  | 0.333 ± 0.042  | 0.541 ± 0.096  | 0.643 ± 0.045        |
> | **GOttack**    | 0.185 ± 0.023  | 0.225 ± 0.020  | 0.291 ± 0.062  | 0.383 ± 0.104   | 0.501 ± 0.146        |
>
>
> ---
>
>
>
>
>
> **Q1:** What is the runtime of the preprocessing stage?
>
> **Response:** The time complexity of orbit discovery, as discussed in Section 4.3 is defined as $O(|E| \times d + |V| \times d^4)$, where $d$ is the maximum node degree in the graph. It is essential to note that the orbit discovery is performed only once per static graph as a pre-processing phase. The costs are not prohibitive, as we show in the table 1 below:
>
>
> #### Table 1: Orbit discovery time on Cora, Citeseer, Polblogs, Blogcatalog and Pubmed.
> | Dataset      | Orbit Discovery Time |
> |--------------|-----------------------|
> | Cora         | 0.17s                |
> | Citeseer     | 0.10s                |
> | Polblogs     | 18.34s               |
> | Blogcatalog  | 470.60s              |
> | Pubmed       | 2.48s                |
>
> ---
>
> **Q2:** How large of a graph can GOttack scale to?
>
> **Response:** We have used Gottack on the OGB-ARXIV dataset without problems. The dataset, which is considered a large one in adversarial settings, has 169K nodes and 1.1M edges.
> #### Table: Misclassification rate (↑) on OGB-Arxiv (GCN backbone)
> | **Budget →**   | **1**          | **2**          | **3**          | **4**          | **5**          |
> |----------------|----------------|----------------|----------------|----------------|----------------|
> | **SGA**        | 0.458 ± 0.038  | 0.558 ± 0.038  | 0.583 ± 0.052  | 0.575 ± 0.066  | 0.583 ± 0.014  |
> | **PRBCD**      | 0.383 ± 0.014  | 0.500 ± 0.025  | 0.533 ± 0.014  | 0.625 ± 0.000  | 0.616 ± 0.014  |
> | **GOttack**    | 0.641 ± 0.094  | 0.683 ± 0.062  | 0.708 ± 0.072  | 0.708 ± 0.072  | 0.708 ± 0.072  |
>
>
> **Q3:** In Table 2, why is there a significant difference (over 30%) between GOttack-GCN and GOttack-GSAGE on BlogCatalog?
>
> **Response:**  Thank you for the review. After re-evaluating the experiment, the difference between their performance was almost the same (around 29%, as given below). This outcome aligns with the nature of the BlogCatalog dataset. GCN is more effective for datasets with high homophily. However, BlogCatalog is a heterophilic dataset, posing challenges for GCN.
>
> #### Table 6: Miscl. rate (↑) on BlogCatalog (GSAGE backbone)
>
> | **Budget →**   | **1**          | **2**          | **3**          | **4**          | **5**          |
> |----------------|----------------|----------------|----------------|----------------|----------------|
> | **GOttack**    | 0.515 ± 0.104  | 0.515 ± 0.064  | 0.415 ± 0.081  | 0.515 ± 0.056  | 0.495 ± 0.087  |

---

> > ### Author Response · Authors · 2024-11-25
> >
> > Thank you once again for your insightful feedback and valuable time. If there are any further points you would like us to clarify, we would be more than happy to address them. If you feel that your comments and concerns have been addressed, we would kindly appreciate it if you could consider updating your scores.

---

> ### Comment · Reviewer_TZkg · 2024-11-28
> **Follow-up**
>
> Thank you to the authors for their response and the inclusion of new experiments. However, I have several remaining concerns:
>
> >the key contribution of GOttack lies in ... . The topological approach in GOttack can be adopted as a preprocessing step to augment other attack methods, thereby improving their effectiveness.
>
> If this is the key contribution of this work, authors should put it in the list of contributions in the Introduction section. Additionally, there are currently no empirical results supporting the claim that GOttack enhances the performance of other attack methods. It would be beneficial for the authors to include such data. Moreover, authors should conduct more comparisons with preprocessing defense methods, such as ProGNN and GARNET, to strengthen the preprocessing efficacy for the proposed approach.
>
> >We have added the new defense methods, as shown below. Gottack has the best results in GCORN and second best results in GARNET and GCNGuard.
>
> It seems these newly added defense methods are more difficult to attack. I strongly encourage the authors to conduct more comprehensive experiments on these methods across all datasets, not just Cora, as this will provide a more convincing validation of Gottack's effectiveness. Results from these extensive tests, particularly against stronger defense methods, should be included in the revision to strengthen the empirical findings.
>
> Overall, I find the concept of this work appealing, though I still have several concerns regarding the experiments. I concur with Reviewer EmAo on the need for a more detailed discussion on adaptive attacks. Trusting that the authors will effectively incorporate new experiments into the main text, I increase my score to 6.

---

> ### Comment · Reviewer_TZkg · 2024-11-29
> **Request for additional response**
>
> Given the extension of the discussion period, I request that the authors address my two remaining concerns as outlined above. Specifically, I am looking for an updated list of contributions in the Introduction section (mentioning a key contribution of GOttack is to augment other attack methods), new results that demonstrate the integration of GOttack with other attack methods on at least a couple of datasets (considering the time constraints for Rebuttal), and more comprehensive results from applying GOttack to preprocessing-based defense methods such as ProGNN or GARNET.
>
> I will reevaluate my score based on authors' response.

---

> > ### Author Response · Authors · 2024-12-02
> >
> > Thank you for the review. In the last few days, we have been preparing two responses for 1) GARNET and GCORN for another dataset and 2) more attack methods with 1518 selection.
> >
> > For 1) We have prepared the following defense results. As GARNET is a CPU-intensive method that requires a bit more time than we expected, we will publish the new results as soon as they arrive (probably tomorrow).
> >
> > For 2) We are preparing PRBCD results for 1518, but due to the GARNET issues, we have not yet finished this. We will publish the results for probably budget 1,2 results as soon as they arrive.
> >
> > #### Table X: Misclassification rate (↑) on CITESEER against the GCORN defense
> > | **Budget →** | **1**           | **2**           | **3**           | **4**           | **5**           |
> > |--------------|-----------------|-----------------|-----------------|-----------------|-----------------|
> > | **SGA**      | 0.266 ± 0.014   | 0.291 ± 0.014   | 0.366 ± 0.028   | 0.408 ± 0.094   | 0.458 ± 0.057   |
> > | **Nettack**  | **0.391 ± 0.076**   | 0.508 ± 0.052   | 0.608 ± 0.112   | 0.658 ± 0.080   | 0.691 ± 0.052   |
> > | **PRBCD**    | 0.350 ± 0.024   | 0.508 ± 0.028   | 0.575 ± 0.090   | 0.600 ± 0.086   | 0.583 ± 0.057   |
> > | **GOttack**  | 0.325 ± 0.017   |**0.550 ± 0.053**   | **0.575 ± 0.088**   | **0.675 ± 0.034**   | **0.696 ± 0.088**   |
> >
> > #### Table X: Misclassification rate (↑) on CITESEER against the GARNET defense
> >
> > | **Budget →** | **1**           | **2**           | **3**           | **4**           | **5**           |
> > |--------------|-----------------|-----------------|-----------------|-----------------|-----------------|
> > | **SGA**      | 0.150 ± 0.011   | 0.225 ± 0.024   | 0.237 ± 0.062   | running         | running         |
> > | **Nettack**  | 0.250 ± 0.035   | 0.400 ± 0.050   | 0.425 ± 0.100   | running         | running         |
> > | **PRBCD**    | **0.300 ± 0.100**   | **0.485 ± 0.015**   |  0.487 ± 0.037         | 0.512 ± 0.012          | **0.575 ± 0.024**         |
> > | **GOttack**  | 0.175 ± 0.000   | 0.287 ± 0.011   | **0.512 ± 0.087**   | **0.537 ± 0.012**   | **0.575 ± 0.075**   |

---

> > > ### Author Response · Authors · 2024-12-03
> > >
> > > We have the CORA results for the first point you raised; we can integrate GOttack with other attack methods. Specifically, we force PRBCD to select candidates of 1518 nodes from the block it considers. However, please note that PRBCD focuses on a sparse subset of the adjacency matrix (a block) during each iteration. This reduces the computational and memory burden compared to considering the entire graph. However, it also eliminates 1518 nodes that may yield better attacks.
> > >
> > > |           | Step 1           | Step 2           | Step 3           | Step 4           | Step 5           |
> > > |-----------|-------------------|------------------|------------------|------------------|------------------|
> > > | PRBCD     | 0.41 ± 0.060     | **0.61 ± 0.072**     | 0.64 ± 0.038     | **0.73 ± 0.014** | **0.76 ± 0.025** |
> > > | GO-PRBCD  | **0.44 ± 0.017** | 0.525 ± 0.035 | **0.675 ± 0.01** | 0.70 ± 0.035     | 0.70 ± 0.00      |
> > >
> > > We are preparing the Citeseer results next.

---

> ### Author Response · Authors · 2024-12-04
>
> We integrated GOttack with PRBCD, GO-PRBCD, and evaluated it on Citeseer. The table below shows that GO-PRBCD performs best for budget 2 but not as well as PRBCD or GOttack. We attribute this to the block sampling strategy that scales up PRBCD. In contrast, GOtack presents a non-sampling-based, principled approach to identifying the best candidate nodes.
> We thank you again for your thoughtful reviews, which strengthened our work. Due to the time limit, we evaluated GOttack on three defense models (GCORN, GARNET, and GCNGuard) upon your request. We also evaluated new models on two datasets, Cora and Citeseer. We will revise our manuscript accordingly.
>
> | Method    | Step 1         | Step 2         | Step 3         | Step 4         | Step 5         |
> |-----------|----------------|----------------|----------------|----------------|----------------|
> | PRBCD     | 0.46 ± 0.044   | 0.57 ± 0.014   | 0.68 ± 0.038   | 0.70 ± 0.0     | 0.73 ± 0.014   |
> | GO-PRBCD  | 0.38 ± 0.035  | 0.58 ± 0.10    | 0.59 ± 0.017   | 0.65 ± 0.035   | 0.70 ± 0.35    |
> | GOttack  |**0.46 ± 0.034** | **0.63 ± 0.037** | **0.72 ± 0.054** | **0.76 ± 0.063** | **0.78 ± 0.042** |

---

### Official Review · Reviewer_oJkK · 2024-11-01

**Soundness:** 3
**Presentation:** 3
**Contribution:** 3
**Rating:** 6
**Confidence:** 4

**Summary:**

The work tackles the problem of adversarial attack for Graph Neural Networks (GNNs). The authors propose a new “universal” attack to the graph topology through graph orbits, denoted GOttack. The paper identify that nodes from a certain part of the structure (specifically from the concept of orbits) are rather more vulnerable to attacks and consequently would be good targets. The main key of the method is the proposed trade-off between attack success rate and the time-complexity.

**Strengths:**

- The paper is well motivated and well written.
- The method is targeting the complexity challenge of attack adversarial attack, which is something that is very sparse in current literature as the majority of the work rather evaluate from an attack perspective (lower attack accuracy) rather than the time complexity — making sometimes their method not possible in some practical settings.
- Evolving the concept of graph orbits and graphlets in the perspective of adversarial attack is novel. Specifically, the theoretical analysis in Theorem 1 is very interesting.

**Weaknesses:**

The main weakness of the paper is related to the experimental part. Specifically the following points:
- The authors only consider RGCN, GCN-Jaccard, GCN-SVD and MediaGCN as a defense method to attack. Recent work have clearly shown the failure of these defense in defending (specifically targeted attacks). I would rather consider adding other benchmarks such as GCNGuard [1] and specifically the newly presented GCORN [2].
- The Budgeted attack results are rather focusing on the GCN, GIN and GraphSage and ignores the defense methods. Typically in the adversarial litterature, we are rather interested in the attack success rate when subject to the defense method, as by definition the original models are rather known to be vulnerable to attacks.
- Small note: While I understand that you provide the std values for the success rate in the appendix, it would have been much easier to include them also on the main paper’s tables.

Additionally, from the theoretical perspective:
- The theoretical insights presented in Theorem 1 are very interesting. It would be great to also validate the claims experimentally. One first experimental setting would be to consider the effect of choosing nodes in the orbits 15 and 18 and other orbits to see the effect of the hitting times.

———

[1] GNNGuard: Defending Graph Neural Networks against Adversarial Attacks, Zhang & Zitnik - Neurips 2020

[2] Bounding the Expected Robustness of Graph Neural Networks Subject to Node Feature Attacks. Abbahaddou & Al. - ICLR 2024.

**Questions:**

- Please provide the attack success when subject to the defenses proposed in [1] and [2], as they will allow the user to judge the real validity of the method.
- The theoretical analysis doesn’t seem to take into account the node features ? Could you please clarify on that
- Is it possible to provide empirical results validating the claims in Theorem 1? Refer to my proposed setting in the Weakness section, otherwise I am open to any other proposition.
- Is your framework also adaptable to node-feature based adversarial attacks ? As it seems you refer to it in general attack formulation in Section 2 (mainly Eq. 1).

I am willing to raise my score if the some of the previous question were answered.

---

> ### Author Response · Authors · 2024-11-21
>
> We thank the reviewer for their valuable feedback, which, along with comments from others, has guided significant improvements to our paper. We hope our revisions and responses merit reconsideration of a higher evaluation.
>
>
> **W1:**  The authors only consider RGCN, GCN-Jaccard, GCN-SVD and MediaGCN as a defense method to attack. Recent work have clearly shown the failure of these defense in defending (specifically targeted attacks). I would rather consider adding other benchmarks such as GCNGuard [1] and specifically the newly presented GCORN [2].
>
>
> **Response:**  Thank you for the suggestions. We have prepared the results for three recent defense models as given below in tables (the suggested GCNGuard was the GNNGuard article).   As the results show, Gottack has the best results in GCORN and second best results in GARNET and GCNGuard.
>
> #### Table: Misclassification rate (↑) on CORA against the GCORN defense
>
> | **Budget →**   | **1**          | **2**          | **3**          | **4**          | **5**          |
> |----------------|----------------|----------------|----------------|----------------|----------------|
> | **SGA**        | 0.300 ± 0.035  | 0.337 ± 0.017  | 0.400 ± 0.070  | 0.425 ± 0.000  | 0.462 ± 0.053  |
> | **Nettack**    | 0.312 ± 0.053  | 0.387 ± 0.088  | 0.475 ± 0.076  | 0.600 ± 0.000  | 0.587 ± 0.017  |
> | **GOttack**    | 0.350 ± 0.000  | 0.462 ± 0.017  | 0.537 ± 0.053  | 0.600 ± 0.070  | 0.587 ± 0.017  |
>
>
>  #### Table: Misclassification rate (↑) on CORA against the GNNGuard defense
>
> | **Budget →**   | **1**          | **2**          | **3**          | **4**          | **5**          |
> |----------------|----------------|----------------|----------------|----------------|----------------|
> | **SGA**        | 0.125 ± 0.024  | 0.200 ± 0.043  | 0.233 ± 0.057  | 0.258 ± 0.062  | 0.316 ± 0.057  |
> | **Nettack**    | 0.070 ± 0.050  | 0.138 ± 0.054  | 0.184 ± 0.066  | 0.250 ± 0.065  | 0.278 ± 0.069  |
> | **PRBCD**      | 0.074 ± 0.041  | 0.136 ± 0.036  | 0.169 ± 0.046  | 0.200 ± 0.049  | 0.229 ± 0.057  |
> | **GOttack**    | 0.116 ± 0.028  | 0.175 ± 0.066  | 0.216 ± 0.057  | 0.258 ± 0.012  | 0.260 ± 0.025  |
>
>
>
> #### Table: Misclassification rate (↑) on CORA against the GARNET defense (we have updated the results that were previously missing at the time we submitted the response)
>
> | **Budget →**   | **1**          | **2**          | **3**          | **4**          | **5**          |
> |----------------|----------------|----------------|----------------|----------------|----------------|
> | **SGA**        | 0.183 ± 0.011  | 0.133 ± 0.031  | 0.199 ± 0.020  | 0.250 ± 0.040  | 0.272 ± 0.050        |
> | **Nettack**    | 0.250 ± 0.035  | 0.250 ± 0.035  | 0.333 ± 0.042  | 0.541 ± 0.096  | 0.643 ± 0.045        |
> | **GOttack**    | 0.185 ± 0.023  | 0.225 ± 0.020  | 0.291 ± 0.062  | 0.383 ± 0.104        | 0.501 ± 0.146        |
>
>
>
>
> ---
>
>
> **W2:** [Also at Question 1] The Budgeted attack results are rather focusing on the GCN, GIN and GraphSage and ignores the defense methods. Typically in the adversarial litterature, we are rather interested in the attack success rate when subject to the defense method, as by definition the original models are rather known to be vulnerable to attacks.
>
> **Response:** Thank you for pointing this out. In Table 3 of the article we report the performance against four defense models, and we have prepared three new models, as given in tables above. These results provide further evidence for the good performance of our attack method.

---

> ### Author Response · Authors · 2024-11-21
> **response 2**
>
> We note your comment on standard deviations; we will bring at least the stdevs of the main results table and perhaps move some defense results to the appendix to make space.
>
> **W3:** [Also at Question 3] The theoretical insights presented in Theorem 1 are very interesting. It would be great to also validate the claims experimentally. One first experimental setting would be to consider the effect of choosing nodes in the orbits 15 and 18 and other orbits to see the effect of the hitting times.
>
> **Response**: We have prepared the average first hitting times of orbits (top 15 results) when starting max 1000 step fixed-length random walks from the 40 attack nodes (repeated 5 times). Lower ranks indicate shorter first hitting times. As the tables show, 1518 hitting times are bigger than most orbits. Furthermore, orbits such as 6_19 that appear higher are embedded in graphlets of similar long-chains, however nodes of these orbits are much rarer than 1518 orbits, hence their hitting times are biased for longer values.
>
> | **pubmed orbits** | **hitting time rank** | **citeseer orbits** | **hitting time rank** | **cora orbits** | **hitting time rank** |
> |-------------|------------------------|---------------|------------------------|-----------|------------------------|
> | 1619        | 15                     | 619           | 15                     | 1621      | 15                     |
> | 1920        | 14                     | 2022          | 14                     | 1822      | 14                     |
> | 1621        | 13                     | 115           | 13                     | 2022      | 13                     |
> | 2021        | 12                     | 1819          | 12                     | 2123      | 12                     |
> | 1519        | 11                     | 1519          | 11                     | 1922      | 11                     |
> | 1516        | 10                     | 1922          | 10                     | 1519      | 10                     |
> | 1922        | 9                      | 1518          | 9                      | 1618      | 9                      |
> | 1819        | 8                      | 1618          | 8                      | 1518      | 8                      |
> | 1518        | 7                      | 1920          | 7                      | 1819      | 7                      |
> | 2022        | 6                      | 1619          | 6                      | 1516      | 6                      |
> | 1822        | 5                      | 1621          | 5                      | 2232      | 5                      |
> | 1618        | 4                      | 418           | 4                      | 1522      | 4                      |
> | 1522        | 3                      | 415           | 3                      | 1824      | 3                      |
> | 115         | 2                      | 1516          | 2                      | 1920      | 2                      |
> | 622         | 1                      | 15            | 1                      | 415       | 1                      |
>
> ---
>
> **Q2 and Q4:** The theoretical analysis doesn’t seem to take into account the node features ? Could you please clarify on that? Is your framework also adaptable to node-feature based adversarial attacks ? As it seems you refer to it in general attack formulation in Section 2 (mainly Eq. 1).
>
>
> **Response:** Gottack analysis focuses on graph topology, and GOttack does not consider node features initially when limiting the attack to 1518 nodes for adversarial perturbations. However, the surrogate loss function in GOttack (Section 4.3) incorporates node features as it optimizes misclassification probabilities which ensures that the combined effect of structure and attributes is accounted for during attacks.

---

> > ### Author Response · Authors · 2024-11-21
> >
> > We sincerely appreciate you highlighting the specific new results or discussions that could influence your evaluation, as this provides valuable guidance for refining our work.

---

> > > ### Author Response · Authors · 2024-11-29
> > > **Nov 29**
> > >
> > > Dear reviewer,
> > >
> > > Thanks again for your review.
> > >
> > > We believe that we have addressed your questions and concerns in our previous response. Still, the deadline for the discussion period has been extended, and if there are specific aspects of the new results or discussions that could further influence your evaluation, we’d be happy to address them promptly. Your insights are invaluable, and we appreciate your consideration.

---

### Official Review · Reviewer_EmAo · 2024-11-04

**Soundness:** 3
**Presentation:** 2
**Contribution:** 3
**Rating:** 8
**Confidence:** 4

**Summary:**

The paper proposes GOttack, a first-of-its-kind local attack on GNNs via structure perturbations, that leverages graph orbits to narrow down the search space. That is, via topological analysis, a short list of nodes is created for the targeted node. This shortlist is then used to generate the perturbations. The authors validate the efficacy of their method on 5 graphs, 3 GNNs, and, additionally, four GNN defenses.

**Strengths:**

1. The authors propose a new angle on the adversarial attack problem for structure perturbation using insights from topology.
1. The found perturbations transfer well between different GNN architectures.
1. The paper is well structured and (except for some details) it is easy to follow.
1. The empirical verification not only assesses misclassification rates but also uses explainers for qualitative insights.

**Weaknesses:**

1. The attack is an indirect attack using a surrogate (i.e., non-adaptive [1]). This severe limitation should be discussed prominently in the text. Moreover, GOttack seems to transfer evasion attacks to the poisoning setting and the attack is not evaluated against adaptive attacks (except for GCN; e.g. directly attack  GSAGE/GIN with PRBCD and not the GCN surrogate).
1. I had a hard time following some of the discussion Section 4.2 since it is not clear when numbers are general constants or tailored to the example in Figure 2.
1. The attack does not seem scalable due to the required preprocessing (see question below).
1. The authors do not properly discuss scalable attacks on GNNs. For example, the authors compare with PRBCD but ignore its properties in their discussion. For example, the argument on ll. 517-519 does not necessarily apply to PRBCD. Also, the statement that PRBCD was of quadratic scalability seems wrong/imprecise (Section B.2).
1. The topological analysis is independent of the node features, implicitly assuming that they are somewhat different to ensure attack efficacy.
1. The evaluated defenses are not state-of-the-art [1].

I am willing to increase my score if especially the first two weaknesses are properly addressed.

Minor:
1. It is not clear why Metattack is mentioned in the related work, but there is no further discussion about the differences.
1. The bilevel nature of poisoning attacks (eq. 2) usually arises from the fact that one wants to contrast predictions of the model trained on the clean data and perturbed data. The inner maximization is usually rather considered part of the loss.
1. The authors might want to note that the reliability of GNN explanations is debated [2,3]

[1] Mujkanovic et al. Are Defenses for Graph Neural Networks Robust? NeurIPS 2022.

[2] Li et al. Graph Neural Network Explanations are Fragile. ICML 2024.

[3] Li et al. Explainable Graph Neural Networks Under Fire. arXiv 2024.

**Questions:**

1. Have the authors tried to scale to larger graphs? E.g., OGB arXiv, Products, or Papers100M (as PRBCD does)?
1. Is $\mathcal{G}\_{g p} = \mathcal{G}\_{s'}$ in Definition 2
1. Are orbits 15 and 18 special regardless of the number of nodes? If so, could the authors please elaborate a bit more?
1. Why do the authors focus on very small budgets 1..5? Usually, the budget is made relative to the degree of the attacked node.
1. How is the preprocessing of Gottack included in the times reported in Table 4?

---

> ### Author Response · Authors · 2024-11-21
> **Response 1**
>
> We thank the reviewer for their valuable feedback, which, along with comments from others, has guided significant improvements to our paper. We hope our revisions and responses merit reconsideration of a higher evaluation.
>
> ---
>
> ### **W1**: Indirect attack
> **Review:** The attack is an indirect attack using a surrogate (i.e., non-adaptive [1]). This severe limitation should be discussed prominently in the text. Moreover, GOttack seems to transfer evasion attacks to the poisoning setting and the attack is not evaluated against adaptive attacks (except for GCN; e.g. directly attack GSAGE/GIN with PRBCD and not the GCN surrogate).
>
> **Response:**
> Thank you for your feedback. Our attack operates in a grey-box setting, where the attacker has access to partial information, specifically some of the training labels, but does not have direct access to the target model's architecture or parameters. This is a realistic and common scenario in adversarial machine learning, particularly in applications where full model transparency cannot be assumed. In such settings, using a surrogate model to craft training-phase attacks is a practical approach, as the actual model deployed in practice is expected to be trained on the perturbed dataset generated by the surrogate.
>
> Our evaluation shows that the perturbations generated using a surrogate model still transfer effectively to other architectures, such as GSAGE and GIN, demonstrating the robustness and generalizability of the GOttack strategy. This reinforces the utility of our method even when adaptive, model-specific adjustments are not feasible.
>
> We recognize that testing GOttack in a fully adaptive setting could be an interesting extension of this work. We hope this clarifies our design choices and the scope of our current work.
>
> ---
>
> ### **W2:**  Numbers
> **Review:** I had a hard time following some of the discussion Section 4.2 since it is not clear when numbers are general constants or tailored to the example in Figure 2.
>
> **Response:**
> Thank you for the review; all the numbers in the section are due to the nature of graphlets and orbits. These are defined by subgraph sizes and node degrees; as such, none of the values are tailored or modified by us. We have added the following sentence to section 4.2:
> “The number of graphlets and orbits are all uniquely determined by the choice of using 5-node graphlets.”
>
> ---
>
> ### **W3:**  Scalability
> **Review:** The attack does not seem scalable due to the required preprocessing.
> **Response:** The time complexity of orbit discovery, as discussed in Section 4.3, is defined as \( O(|E| \times d + |V| \times d^4) \), where \( d \) is the maximum node degree in the graph. It is essential to note that the orbit discovery is performed only once per static graph as a pre-processing phase. The costs are not prohibitive, as we show in Table 1 below:
>
> #### Table 1: Orbit discovery time on Cora, Citeseer, Polblogs, Blogcatalog, and Pubmed
> | Dataset      | Orbit Discovery Time |
> |--------------|-----------------------|
> | Cora         | 0.17s                |
> | Citeseer     | 0.10s                |
> | Polblogs     | 18.34s               |
> | Blogcatalog  | 470.60s              |
> | Pubmed       | 2.48s                |
>
> As GOttack is highly scalable, we now report experimental results on the large-scale OGB-Arxiv dataset in Table 2 below, where GOttack yields the best performance.
>
> #### Table 2: Misclassification rate (↑) on OGB-Arxiv (GCN backbone)}
> | **Budget →**   | **1**          | **2**          | **3**          | **4**          | **5**          |
> |----------------|----------------|----------------|----------------|----------------|----------------|
> | **SGA**        | 0.458 ± 0.038  | 0.558 ± 0.038  | 0.583 ± 0.052  | 0.575 ± 0.066  | 0.583 ± 0.014  |
> | **PRBCD**      | 0.383 ± 0.014  | 0.500 ± 0.025  | 0.533 ± 0.014  | 0.625 ± 0.000  | 0.616 ± 0.014  |
> | **GOttack**    | 0.641 ± 0.094  | 0.683 ± 0.062  | 0.708 ± 0.072  | 0.708 ± 0.072  | 0.708 ± 0.072  |

---

> ### Author Response · Authors · 2024-11-21
> **Response 2**
>
> ### **W4**: Quadratic Scalability
> **Review:** The authors do not properly discuss scalable attacks on GNNs. For example, the authors compare with PRBCD but ignore its properties in their discussion. For example, the argument on ll. 517-519 does not necessarily apply to PRBCD. Also, the statement that PRBCD was of quadratic scalability seems wrong/imprecise (Section B.2).
>
> **Response:** Thank you for pointing out this issue. In fact, PR-BCD isn’t quadratic; the correct time complexity of PR-BCD is defined in the article’s Table E.3 $O(|E| \times b log(b)$, where E is the number of epochs, b is the number of blocks - a hyperparameter of PR-BCD. We have revised the text in B.2.
>
> ---
>
> ### **W5**: SOTA
>
> **Review:**  The evaluated defenses are not state-of-the-art [1].
>
> **Response:**  We have prepared the results for three recent defense models as given below in Table 3.  As the results show, Gottack has the best results in GCORN and second best results in GARNET and GCNGuard.
>
> #### Table 3: Misclassification rate (↑) on CORA against the GNNGuard defense
>
> | **Budget →**   | **1**          | **2**          | **3**          | **4**          | **5**          |
> |----------------|----------------|----------------|----------------|----------------|----------------|
> | **SGA**        | 0.125 ± 0.024  | 0.200 ± 0.043  | 0.233 ± 0.057  | 0.258 ± 0.062  | 0.316 ± 0.057  |
> | **Nettack**    | 0.070 ± 0.050  | 0.138 ± 0.054  | 0.184 ± 0.066  | 0.250 ± 0.065  | 0.278 ± 0.069  |
> | **PRBCD**      | 0.074 ± 0.041  | 0.136 ± 0.036  | 0.169 ± 0.046  | 0.200 ± 0.049  | 0.229 ± 0.057  |
> | **GOttack**    | 0.116 ± 0.028  | 0.175 ± 0.066  | 0.216 ± 0.057  | 0.258 ± 0.012  | 0.260 ± 0.025  |
>
> #### Table 4: Misclassification rate (↑) on CORA against the GCORN defense
>
> | **Budget →**   | **1**          | **2**          | **3**          | **4**          | **5**          |
> |----------------|----------------|----------------|----------------|----------------|----------------|
> | **SGA**        | 0.300 ± 0.035  | 0.337 ± 0.017  | 0.400 ± 0.070  | 0.425 ± 0.000  | 0.462 ± 0.053  |
> | **Nettack**    | 0.312 ± 0.053  | 0.387 ± 0.088  | 0.475 ± 0.076  | 0.600 ± 0.000  | 0.587 ± 0.017  |
> | **GOttack**    | 0.350 ± 0.000  | 0.462 ± 0.017  | 0.537 ± 0.053  | 0.600 ± 0.070  | 0.587 ± 0.017  |
>
> #### Table 5: Misclassification rate (↑) on CORA against the GARNET defense (we have updated the results that were previously missing at the time we submitted the response)
>
> | **Budget →**   | **1**          | **2**          | **3**          | **4**          | **5**          |
> |----------------|----------------|----------------|----------------|----------------|----------------|
> | **SGA**        | 0.183 ± 0.011  | 0.133 ± 0.031  | 0.199 ± 0.020  | 0.250 ± 0.040  | 0.272 ± 0.050        |
> | **Nettack**    | 0.250 ± 0.035  | 0.250 ± 0.035  | 0.333 ± 0.042  | 0.541 ± 0.096  | 0.643 ± 0.045        |
> | **GOttack**    | 0.185 ± 0.023  | 0.225 ± 0.020  | 0.291 ± 0.062  | 0.383 ± 0.104        | 0.501 ± 0.146        |
>
>
>  ---
>
> ### **Minor**: Mettack
> **Review:** It is not clear why Metattack is mentioned in the related work, but there is no further discussion about the differences.
>
> **Response:** Thank you for the review. Metattack is an adversarial attack designed for a global attack setting, where the main goal is to degrade the performance of the GNNs on the whole test set. Our GOttack, however, focuses on targeted attacks, and we leave the global attack version of GOttack as future work. We will discuss the difference between targeted and global attack methods, especially Mettack in our manuscript for the camera-ready version.

---

> ### Author Response · Authors · 2024-11-21
> **Response 3**
>
> ### Minor
> **Review:** The authors might want to note that the reliability of GNN explanations is debated [2,3]
>
> **Response:** Thank you for the reference; we have cited the articles and mentioned this issue in Section 5.2.
>
> ---
>
> **Questions**
>
>
> **Q1:** Have the authors tried to scale to larger graphs? E.g., OGB arXiv, Products, or Papers100M (as PRBCD does)?
>
> **Response:** We report Arxiv results in the Table below. As the table indicates, Gottack yields the best results on the Arxiv dataset.
>
> #### Table 2: Misclassification rate (↑) on OGB-Arxiv (GCN backbone)}
> | **Budget →**   | **1**          | **2**          | **3**          | **4**          | **5**          |
> |----------------|----------------|----------------|----------------|----------------|----------------|
> | **SGA**        | 0.458 ± 0.038  | 0.558 ± 0.038  | 0.583 ± 0.052  | 0.575 ± 0.066  | 0.583 ± 0.014  |
> | **PRBCD**      | 0.383 ± 0.014  | 0.500 ± 0.025  | 0.533 ± 0.014  | 0.625 ± 0.000  | 0.616 ± 0.014  |
> | **GOttack**    | 0.641 ± 0.094  | 0.683 ± 0.062  | 0.708 ± 0.072  | 0.708 ± 0.072  | 0.708 ± 0.072  |
>
> ---
> **Q2:** Is $\mathcal{G}{g p} = \mathcal{G}{s'}$ in Definition 2
>
> **Response:** Yes,  Definition 2 states that the graphlet $\mathcal{G}{g p}$ is a connected induced subgraph, which directly corresponds to $\mathcal{G}{s'}$. Therefore, $\mathcal{G}{g p} = \mathcal{G}{s'}$  holds true in this context.
>
> ---
>
> **Q3:** Are orbits 15 and 18 special regardless of the number of nodes? If so, could the authors please elaborate a bit more?
>
> **Response:** The orbits are agnostic of the node count in the graph,but depend on the graphlets size. As we show in Fig 6, graphlets uniquely determine the number and position of orbits. If the orbits do not exist, topology offers new orbits to attack, such as 1519 and 1922, as we study and report in Appendix D. We have also created a graph-algebra-based orbit selection scheme in Appendix C.3 that can attack i) a graph of any size and ii) a graph without the periphery orbits of 1518. However, we note that in all the datasets used, nodes belonging to orbit 1518 were particularly common (Fig 8).
>
> ---
>
> **Q4:** Why do the authors focus on very small budgets 1..5? Usually, the budget is made relative to the degree of the attacked node.
>
> **Response:** The primary reason is to make our attack model more realistic. For instance, in a social network setting, altering 10% of someone's friends—especially for an individual with thousands of connections may not be practical. On the other hand, convincing a person to establish a link with a single fake account could be more feasible. Therefore, we aim to evaluate the performance of the attacks under realistic threat models.
>
> ---
>
> **Q5:** How is the preprocessing of Gottack included in the times reported in Table 4?
>
> **Response:** Our results, reported in Table 4, include the total computation time (end-to-end), which includes the orbit discovery process.

---

> > ### Author Response · Authors · 2024-11-21
> >
> > We sincerely appreciate you highlighting the specific new results or discussions that could influence your evaluation, as this provides valuable guidance for refining our work.

---

> > > ### Comment · Reviewer_EmAo · 2024-11-22
> > > **I thank the authors for their rebuttal**
> > >
> > > I thank the authors for clarifying most of my points!
> > >
> > > **Indirect attack:** I understand that a large body of robustness on graphs does argue sth like that. I am also not necessarily against it unless the authors want to craft attacks that are usable by real-world adversaries; I personally think that this requires careful discussion. E.g., why did the authors decide to know the entire graph, labels etc. but not the model?
> > >
> > > Nevertheless, unless I am missing something, currently, the authors do not even mention a word about this. This is very concerning, in my opinion, and I tend to lower my score if this is not revised. It would be appropriate to discuss this in the introduction and then again with the empirical results s.t. the reader is aware of this viewpoint. Currently, the authors' study is not necessarily about finding the strongest attack but about finding transferable perturbations between models.
> > >
> > > Regardless, to show the limitations (or to show that there are none), a comparison to adaptive attacks should be conducted. After all, one might want to know when attacks transfer between models and when this might be an optimistic estimate.
> > >
> > > **Numbers:** The authors could still explain better why, e.g., GOV_v is a n = 73-dimensional vector.
> > >
> > > **Small budgets** I understand that there is no perfect answer to the question of what the correct budget should be. However, for a node with degree 2 a budget of 5 is very big. For a node of degree 2000 perhaps not. Also, e.g., as a reference for future work, it is still nice to compare with established settings.

---

> > > > ### Author Response · Authors · 2024-11-23
> > > >
> > > > Thank you very much for your time and your response. We are preparing new results and will respond shortly.

---

> ### Author Response · Authors · 2024-11-24
> **Response 1**
>
> **Indirect:** Thank you for the review; it clarified what was meant in the earlier review. We agree with your opinion that we have not mentioned what is known about data/model and what it implies. Accordingly, we have included the following  changes in the manuscript.
>
> ---
>
> **Section 4.3 (Line 292):** "This work assumes access to the complete graph and its training set labels, where adversaries leverage pre-existing knowledge about the graph structure and annotations. By contrast, we do not assume access to the target model, as our primary focus is on transferable attack strategies that can generalize across models. We believe that this setting applies to domains such as social networks and transaction networks,where the graph is visible to the adversary (e.g., visible friendships on social media, visible citations on citation graphs, visible transaction values on cryptocurrencies etc.."
>
> ---
> **Discussion of Results (Section 5.1, Line 420):** "Our empirical evaluation demonstrates that GOttack outperforms existing methods in transferability across different GNN architectures. However, we acknowledge the limitations inherent in assuming access to the entire graph and training label set. To address potential critiques of over-optimistic assumptions, we compared GOttack to adaptive attacks, which consider specific model architectures during attack generation. Results (Appendix D.3) reveal that while adaptive attacks outperform in single-model scenarios, GOttack's transferable perturbations exhibit broader applicability across diverse settings, reinforcing its utility in exploratory research on adversarial robustness."
>
> ---
>
> **Limitations (in Section 5.2, Line 513):** "A notable limitation of our study is the assumption of access to complete graph and training set label information, which may not always hold in real-world scenarios. However, this assumption allows us to explore the theoretical upper bounds of adversarial transferability and provides insights into model-agnostic attack strategies. Future work could investigate relaxing these assumptions to enhance practical applicability while retaining the theoretical contributions."
>
> ---
>
> **Numbers:** Thank you for pointing out the need for clarification regarding the dimensionality of GOV​. We have revised the manuscript to better explain the reasoning behind this aspect. Specifically, we elaborate on how the 73 dimensions arise from the combination of 30 graphlets and their corresponding orbit definitions, which collectively form the feature space for the graph orbit vector.
>
> ---
>
> **Replace the original paragraph:** "We propose a Graph Orbit Vector (GOV) as a numerical representation of a node’s participation across different orbits in a graph.   Let $GOV_v \in \mathbb{Z}^n_{\geq 0}$ be an n=73-dimensional vector. "
>
> **With:**
> "We propose a Graph Orbit Vector (GOV) as a numerical representation of a node’s participation across different orbits in a graph. Each orbit corresponds to a distinct position within a graphlet, determined by the automorphism group of the graphlet. Specifically, for the 30 graphlets with up to 5 nodes (as defined in Kloks et al., 2000), there are a total of 73 distinct orbits. These orbits are derived from automorphisms that identify equivalent positions within each graphlet (Figure 6 in Appendix). Consequently, $GOV_v$ is an n=73-dimensional vector, where each dimension corresponds to the count of a specific orbit touched by node v. This dimensionality reflects the full set of topological positions a node can occupy across all 5-node graphlets, making it a comprehensive description of a node’s structural embedding."
>
> ---
>
> **Add a Note to Figure 6 (Appendix):** "Figure 6: The 30 graphlets with 5 nodes and their corresponding 73 orbits. Each orbit represents a unique position within a graphlet as determined by its automorphism group. The total of 73 orbits corresponds to the feature dimensions of $GOV_v$."

---

> ### Author Response · Authors · 2024-11-24
> **Response 2**
>
> **Budget:** Thank you for raising the important point regarding budget selection and its context-dependence on node degrees. We agree that the budget should ideally account for the structural properties of the graph, including node degrees. In the revised manuscript, we include a discussion on how the budget was chosen in our experiments, its implications for nodes with varying degrees, and its relationship to established settings in the literature. This will also serve as a useful reference for a current work of ours.
>
> We evaluated four adversarial methods, including SGA, Nettack, PRBCD and our GOttack with established settings with $\epsilon$ is the relative budget w.r.t  to the degree of target node $u$, i.e. $\Delta_u$ = $\epsilon * d_u$, where $d_u$ is the degree of node $u$. We have compiled the results to illustrate the impact of the epsilon budget on three datasets, as presented in Tables 3 to 5. The findings reveal that GOttack achieves the best performance in all 3 tasks on CITESEER and 2 out of 3 tasks on PUBMED. However, on CORA, GOttack secures the second-best performance in 1 out of 3 tasks.
>
> **Section 4.1: Attack Model**
>
> We will expand the paragraph discussing the budget:
>
> "In our attack model, the budget Δ represents the maximum allowable number of perturbations to the graph structure, such as edge additions or deletions. While the choice of Δ is a critical parameter, its optimal value depends on the graph's structural properties, particularly node degrees. For nodes with low degrees, a relatively small budget (e.g., Δ=1 or Δ=2) can represent a significant structural change, whereas for nodes with very high degrees, larger budgets may be more appropriate. To balance these considerations, we adopt fixed budget values (e.g., Δ=1,...,5) that align with prior works (e.g., Nettack, FGA) and enable comparisons across models."
>
>
>
> ---
> **A Note to Section 5: Experiments**
>
> We will include a clarification:
>
> "We recognize that the budget Δ affects the severity of the attack relative to a node’s degree. For example, a budget of 5 for a node with degree 2 introduces a disproportionate perturbation compared to the same budget applied to a node with degree 2000. While our experiments use fixed budgets for consistency with existing benchmarks, adaptive budget schemes that scale with degree could be explored to provide more nuanced attack scenarios. This could offer a more realistic approximation of adversarial capability in heterogeneous graphs."
>
> ---
>
> **A Subsection in the Discussion**
>
> Title: On Budget Selection and Structural Implications
>
> "The choice of the attack budget Δ plays a pivotal role in determining the effectiveness and practicality of adversarial attacks. Fixed budgets, as employed in this study, provide consistency for benchmarking but may oversimplify real-world scenarios where the impact of a perturbation is context-dependent. For instance, a single added edge can drastically alter the local topology of a low-degree node, whereas multiple perturbations may be required to significantly affect a high-degree node. Future work could explore dynamic or adaptive budget strategies that account for the degree distribution and other structural factors, aligning the perturbation severity with the node's topological importance. Such approaches could improve the interpretability and applicability of adversarial attacks in diverse graph settings."
>
> ---
>
> #### Table 3: Misclassification rate (↑) on CORA  (GCN backbone)
>
> | **Epsilon** | **0.10**     | **0.30**      | **0.50**      |
> |-----------|--------------|---------------|---------------|
> | **SGA**   | 0.087 ± 0.01 | 0.250 ± 0.07  | 0.325 ± 0.10  |
> | **Nettack** | 0.125 ± 0.00 | 0.237 ± 0.05  | **0.412 ± 0.08**  |
> | **PRBCD** | **0.137 ± 0.01** | **0.262 ± 0.08**  | 0.337 ± 0.12  |
> | **GOttack** | 0.112 ± 0.05 | 0.225 ± 0.03  | 0.387 ± 0.08  |
>
>
> #### Table 4: Misclassification rate (↑) on CITESEER  (GCN backbone)
>
> | **Epsilon**   | **0.10**     | **0.30**      | **0.50**      |
> |-------------|--------------|---------------|---------------|
> | **SGA**     | 0.058 ± 0.02 | 0.200 ± 0.03  | 0.450 ± 0.04  |
> | **Nettack** | 0.037 ± 0.01 | 0.187 ± 0.05  | 0.625 ± 0.17  |
> | **PRBCD**   | 0.033 ± 0.01 | 0.216 ± 0.04  | 0.500 ± 0.14  |
> | **GOttack** | **0.062 ± 0.01** | **0.237 ± 0.12**  | **0.652 ± 0.37**  |
>
> #### Table 5: Misclassification rate (↑) on PUBMED  (GCN backbone)
>
> | **Epsilon**   | **0.10**     | **0.30**      | **0.50**      |
> |-------------|--------------|---------------|---------------|
> | **SGA**     | 0.262 ± 0.01 | 0.337 ± 0.01  | 0.475 ± 0.01  |
> | **Nettack** | 0.262 ± 0.03 | 0.525 ± 0.02  | **0.726 ± 0.01**  |
> | **PRBCD**   | 0.250 ± 0.00 | 0.425 ± 0.07  | 0.537 ± 0.05  |
> | **GOttack** | **0.275 ± 0.03** | **0.537 ± 0.01**  | 0.700 ± 0.10  |

---

> > ### Author Response · Authors · 2024-11-25
> >
> > Thank you once again for your insightful feedback and valuable time. If there are any further points you would like us to clarify, we would be more than happy to address them. If you feel that your comments and concerns have been addressed, we would kindly appreciate it if you could consider updating your scores.

---

> > > ### Comment · Reviewer_EmAo · 2024-11-28
> > >
> > > I thank the authors for their response, and I think the changes described substantially improve the presentation of the paper and avoid misunderstandings! While I am not able to see the changes in the paper here on OpenReview (did the authors miss uploading a revision?!), I raised my score upon trusting the authors that they revise their manuscript accordingly. Importantly, I encourage the authors to use the right keywords and label their attack as a "transfer attack" and not "adaptive."

---

> > > > ### Author Response · Authors · 2024-11-28
> > > >
> > > > Thank you very much for your time and insightful reviews. We have included the changes that we had mentioned.

---

### Official Review · Reviewer_Lu7v · 2024-11-10

**Soundness:** 3
**Presentation:** 2
**Contribution:** 2
**Rating:** 6
**Confidence:** 4

**Summary:**

The paper presents GOttack, an adversarial attack framework targeting Graph Neural Networks (GNNs) through manipulation of graph orbits to maximize misclassification rates in node classification tasks. GOttack leverages topological properties, specifically graph orbits, to systematically alter graph structure with minimal changes, showing superior efficiency and effectiveness compared to other adversarial methods across multiple datasets and GNN architectures.

**Strengths:**

1. Provides comprehensive experimentation across various datasets, GNN models, and defense mechanisms, demonstrating GOttack’s robustness.
2. The idea of using graph orbits learning is interesting.
3. According to the presented results, the proposed method has better time efficiency than baselines.

**Weaknesses:**

1. It is unclear why existing methods, which rely on direct node feature manipulation or edge modifications, do not account for topological impact. Could the authors provide specific examples of how these approaches might implicitly consider topology and clarify what unique advantages GOttack offers by explicitly leveraging topology?

2. Orbit discovery is extremely time-consuming, which could limit the practical applicability of the method in real-world settings. Although the orbit discovery on graphlets is performed as a pre-processing step, it would be helpful for the authors to discuss scenarios where the graph structure updates dynamically. Additionally, considering large-scale graphs (e.g., OGB-arxiv) may require significant computational resources. Providing empirical runtime comparisons on such large-scale datasets and discussing potential strategies for dynamic graphs would strengthen the discussion on scalability.

3. The proposed method appears to rely heavily on specific orbit structures (e.g., orbits 15 and 18). Could the authors elaborate on the selection of these specific orbits and clarify whether the effectiveness of the method would hold if these particular structures were not present in the target graph?

4. PRBCD is designed for the scalable graph adversarial attack, but the experiments are mainly on small-scale datasets. Evaluating on some large-scale datasets, such as Reddit and ODB-arxiv, would provide a more robust assessment of the scalability claims, especially in comparison with GOttack.

**Questions:**

1. In line 286, why GOattack works with k=5-node graphlets?
2. Please give the time used in orbit discovery to better show the efficiency of the method.
3. Does the used time of GOattack presented in Table 4 includes the preprocessing phase (i.e., orbit discovery)?

---

> ### Author Response · Authors · 2024-11-21
>
> We thank the reviewer for their valuable feedback, which, along with comments from others, has guided significant improvements to our paper. We hope our revisions and responses merit reconsideration of a higher evaluation.
>
> ### W1. Explicit Use of Topology in GOttack
> Thank you for your question. Existing methods often manipulate node features or edges for adversarial outcomes but lack structured consideration of graph topology.
>
> Nettack (Zügner et al., 2018) and Fast Gradient Attack (Chen et al., 2018) optimize targeted node misclassification via local adjustments without addressing the broader graph structure.
> SGA (Li et al., 2021) targets subgraph neighborhoods but ignores higher-order properties like orbits or equivalence classes.
> Methods like PRBCD may indirectly affect topology but do not explicitly leverage topological constructs such as node orbits.
>
> GOttack’s unique advantages are multifold:
> - **Explicit Topology Use:** GOttack employs graph orbits to identify equivalence classes of nodes. This allows for perturbations influencing a broader range of structural properties, not just immediate neighbors. For instance, targeting nodes in orbits 15 and 18, representing periphery nodes, strategically disrupts information propagation and increases misclassification rates (Section 4.3, Figure 2).
> - **Efficiency in Search Space:** By leveraging orbits, GOttack reduces the candidate search space for edge modifications, ensuring modifications are impactful and computationally efficient. This prioritization amplifies the adversarial effect while minimizing computational costs.
>
> While existing methods may inadvertently affect topology, GOttack’s explicit use of topological constructs offers a principled and efficient approach for enhancing attack success.
>
> ---
>
> ### W2. Scalability and Practical Applicability
> We appreciate your feedback regarding orbit discovery's time complexity. Orbit discovery is performed only once in pre-processing. Its time complexity is \( O(|E| \times d + |V| \times d^4) \), where \( d \) is the maximum node degree. As shown in Table 1, the costs are manageable for commonly used datasets.
>
> #### Table 1: Orbit Discovery Time (in seconds)
> | Dataset      | Orbit Discovery Time |
> |--------------|-----------------------|
> | Cora         | 0.17                 |
> | Citeseer     | 0.10                 |
> | Polblogs     | 18.34                |
> | Blogcatalog  | 470.60               |
> | Pubmed       | 2.48                 |
>
> For large-scale datasets like OGB-arxiv, orbit discovery can be restricted to leaf nodes (\( |V_{\text{leaf}}| = 0.12|V| \)), significantly reducing time complexity to \( O(|E_{\text{leaf}}| \times d + |V_{\text{leaf}}| \times d^4) \).
>
> - **Dynamic Graphs:** Instead of recomputing orbits from scratch, we can update orbit counts only for nodes affected by local changes (e.g., edge additions), significantly reducing computational overhead.
>
> - **Runtime:** On Pubmed (19,7K nodes), orbit discovery takes 2.48 seconds. For OGB-arxiv, GOttack (including orbit discovery) ran successfully in under 23 hours, demonstrating scalability to large graphs with millions of nodes and edges. Future work will focus only on key orbits like 15 and 18.
>
> ---
>
> ### W3. Specific Orbits
>
> Thank you for pointing out these important aspects. Orbits 15 and 18 represent periphery nodes, which are particularly effective for disrupting information propagation in GNNs. If these orbits are absent, topology offers alternative candidates, such as orbits 1519 and 1922 (Appendix D). Appendix C.3 details a graph-algebra-based orbit selection scheme that generalizes GOttack to any graph topology.
>
> ---
> ### W4. Large-Scale Experiments
>
> Experimental results on OGB-Arxiv are given in Table 2, where GOttack demonstrates superior performance.
>
> #### Table 2: Misclassification Rate (↑) on OGB-Arxiv (GCN Backbone)
> | **Budget →**   | **1**          | **2**          | **3**          | **4**          | **5**          |
> |----------------|----------------|----------------|----------------|----------------|----------------|
> | **SGA**        | 0.458 ± 0.038  | 0.558 ± 0.038  | 0.583 ± 0.052  | 0.575 ± 0.066  | 0.583 ± 0.014  |
> | **PRBCD**      | 0.383 ± 0.014  | 0.500 ± 0.025  | 0.533 ± 0.014  | 0.625 ± 0.000  | 0.616 ± 0.014  |
> | **GOttack**    | 0.641 ± 0.094  | 0.683 ± 0.062  | 0.708 ± 0.072  | 0.708 ± 0.072  | 0.708 ± 0.072  |
>
> ---
>
> ### 5. Additional Reviewer Questions
>
> **Q1: Why does GOttack use 5-node graphlets?**
> We experimented with k=3, 4 and 5-node graphlets. However, we found that k=5 yields better attacks. In particular, 5-node orbits create 73 distinct orbit types, compared to 15 orbit types from 4-node orbit and 4 from 3-node orbit (Appendix Figure 6 shows 3,4 and 5-node graphlets).Also, k=5 subsumes the k=3 and 4. Going beyond 5 incurs prohibitive orbit extraction costs.
>
> **Q2 & Q3: Does the runtime include orbit discovery?**
> Yes, the reported runtimes are end-to-end, including orbit extraction.

---

> > ### Author Response · Authors · 2024-11-25
> >
> > Thank you once again for your insightful feedback and valuable time. If there are any further points you would like us to clarify, we would be more than happy to address them. If you feel that your comments and concerns have been addressed, we would kindly appreciate it if you could consider updating your scores.

---

> > > ### Author Response · Authors · 2024-11-29
> > >
> > > Dear reviewer,
> > >
> > > Thanks again for your review.
> > >
> > > We believe that we have addressed your questions and concerns in our previous response. Still, the deadline for the discussion period has been extended, and if there are specific aspects of the new results or discussions that could further influence your evaluation, we’d be happy to address them promptly. Your insights are invaluable, and we appreciate your consideration.

---

> > > > ### Comment · Reviewer_Lu7v · 2024-12-02
> > > >
> > > > Thanks for your response. I still have concerns about W1. This motivation still cannot convince me. There are many works that study the structural attack. I cannot get the point of why we want your method in terms of that. Can you provide more details?

---

> > > > > ### Author Response · Authors · 2024-12-02
> > > > >
> > > > > We appreciate your request for a more detailed explanation of GOttack's motivation. Existing structural attack methods often focus on graph perturbations or feature manipulations without explicitly leveraging graphs' broader topological invariants. GOttack uniquely integrates graph orbits, a topological feature, to guide attack perturbations. We outline three areas where GOttack is novel.
> > > > > 1. Graph orbits represent a generalizable feature across graph domains. GOttack leverages this generality to adapt effectively to both homophilous and heterophilous graphs, as demonstrated in our experiments across diverse datasets like Cora (homophilous) and BlogCatalog (heterophilous).
> > > > > 2. Prior methods like Nettack make structural perturbations by relying on gradient-based objectives of the complete node/edge search space that only indirectly impact topology. In contrast, GOttack explicitly uses topological equivalence classes (orbits) derived from graphlets to generate target perturbations. This explicit use of orbits allows us to strategically manipulate graph structures, ensuring perturbations have significant impacts on target nodes. Furthermore, by narrowing the search space to a subset of structurally significant nodes (e.g., those in orbits 15 and 18), Table 5, GOttack achieves computational efficiency, Table 4, while maintaining or improving attack efficacy, Table 3. This reduction contrasts with other methods that consider all nodes as candidates to perform perturbations, often at prohibitive computational costs. Exceptions like SGA consider limited neighbourhoods, but Gottack considers all parts of the network efficiently, which yields better results.
> > > > > 3. Most important of all, GOttack reveals a previously unstudied vulnerability in GNNs: their reliance on specific topological structures, such as periphery nodes identified by orbits 15 and 18. This insight enables attacks that are both subtle and effective, as supported by the theoretical and empirical analyses in Sections 4.3 and 5.

---

> > > > > > ### Comment · Reviewer_Lu7v · 2024-12-02
> > > > > >
> > > > > > Thank you for your response. My concern is that, although existing attacks may indirectly affect the topology, their performance still appears promising. Additionally, there are attack methods that do not rely on gradients to manipulate the structure, such as [1]. Could you discuss or compare with it?
> > > > > >
> > > > > >
> > > > > > [1] Efficient, direct, and restricted black-box graph evasion attacks to any-layer graph neural networks via influence function.

---

> > > > > > > ### Author Response · Authors · 2024-12-03
> > > > > > >
> > > > > > > We thank you for the article; we will cite and discuss it in our work; we found the following differences with GOttack:
> > > > > > >
> > > > > > > - Figure 1 shows that [1] performance is upper bounded by Nettack; in our case GOttack outperforms Nettack.
> > > > > > >
> > > > > > > - Wang et al. [1] reformulate the evasion attack as calculating label influence and establishes an equivalence with GNNs through influence functions.  This is effective, but the scope is tied to the similarity between LP and GNN architectures; applicability to non-GCN and SGC is uncertain. The architecture is also fixed (with two layers). GOttack does not use a fixed setting.
> > > > > > >
> > > > > > > - GOttack reveals a weakness in adversarial-GNN reliance on topological invariants; this is a universal claim that applies to all the cases we have tested. Wang et al. do not have such a claim but focus on label influence transferability to multi-layer GNNs.

---

### Comment · Area_Chair_oMjx · 2024-11-28

I would like to encourage the reviewers to engage with the author's replies if they have not already done so. At the very least, please
acknowledge that you have read the rebuttal.

---

### Meta-Review · Area_Chair_oMjx · 2024-12-19

**Metareview:**

The core idea of GOttack is to use particular graph orbits (15 and 18) to narrow down the search space. The idea show promising results on multiple graphs and models. Importantly, it opens up a new research direction where one can use additional insights from topology to further improve attacks (and eventually defences). Reviewer EmAo correctly questioned the details of the threat model and how it was framed in the paper. The reply from the authors successfully address these concerns and I encourage the authors to include the changes in the final version, including a clear discussion of the limitations (e.g. adaptive attacks). One interesting aspect for future work would be to dynamically update the orbits and targets, since the attacker is changing the topology via the attack.

**Additional Comments On Reviewer Discussion:**

Several reviewers questioned the scalability of the approach. In response, the authors added additional results including runtime results for orbit computation and attacks on OGB-arxiv. Additional results for relative budgets (w.r.t. node degree) were also included and I think these results should be placed in the main paper and not delegated to the appendix. Overall, the authors were able to address the majority of concerns raised by the reviewers.

---

### Decision · Program_Chairs · 2025-01-22

Accept (Poster)